# Enhancer–promoter interactions and transcription are largely maintained upon acute loss of CTCF, cohesin, WAPL or YY1

**Tsung-Han S. Hsieh**[1,2,3,4,5]**, Claudia Cattoglio**[1,2,3,4,7]**, Elena Slobodyanyuk**[1,2,7]**, Anders S. Hansen** ⓘ [6]**, Xavier Darzacq** ⓘ [1,2,3,5] ✉ **& Robert Tjian** ⓘ [1,2,3,4] ✉

It remains unclear why acute depletion of CTCF (CCCTC-binding factor) and cohesin only marginally affects expression of most genes despite substantially perturbing three-dimensional (3D) genome folding at the level of domains and structural loops. To address this conundrum, we used high-resolution Micro-C and nascent transcript profiling in mouse embryonic stem cells. We find that enhancer–promoter (E–P) interactions are largely insensitive to acute (3-h) depletion of CTCF, cohesin or WAPL. YY1 has been proposed as a structural regulator of E–P loops, but acute YY1 depletion also had minimal effects on E–P loops, transcription and 3D genome folding. Strikingly, live-cell, single-molecule imaging revealed that cohesin depletion reduced transcription factor (TF) binding to chromatin. Thus, although CTCF, cohesin, WAPL or YY1 is not required for the short-term maintenance of most E–P interactions and gene expression, our results suggest that cohesin may facilitate TFs to search for and bind their targets more efficiently.

High-throughput chromosomal conformation capture (Hi-C)-based assays have transformed our understanding of 3D genome folding[1,2]. Based on such studies, we can distinguish at least three levels of 3D genome folding. First, the genome is segregated into A and B compartments, which largely correspond to active and inactive chromatin segments, respectively, and appear as a plaid-like pattern in Hi-C contact maps[3]. Second, the proteins CTCF and cohesin help fold the genome into topologically associating domains (TADs)[4,5] and structural chromatin loops[6], probably through DNA loop extrusion[7,8]. Third, at a much finer scale, transcriptional elements engage in long-range chromatin interactions such as E–P and promoter–promoter (P–P) interactions to form local domains[9–11].

Elegant experiments combining acute protein depletion of CTCF, cohesin and cohesin-regulatory proteins with Hi-C or imaging approaches have revealed the role of CTCF and cohesin in regulating the first two levels: TADs and compartments[12–16]. However, Hi-C is ineffective for capturing the third level of 3D genome folding: the fine-scale transcriptionally important E–P/P–P interactions[9,17,18]. Our understanding of the role of CTCF and cohesin in regulating gene expression has mainly come from genetic experiments focusing on a few developmental loci[19–21]. Thus, it remained unclear whether, when, where and how CTCF/cohesin regulates E–P/P–P interactions and gene expression.

We recently reported that Micro-C can effectively resolve ultra-fine 3D genome folding at nucleosome resolution[22,23], including E–P/P–P interactions[9,17]. In the present study, we used Micro-C, chromatin immunoprecipitation sequencing (ChIP-seq), total RNA-sequencing (RNA-seq) and nascent RNA-seq[24] to systematically investigate how

[1]Department of Molecular and Cell Biology, University of California, Berkeley, Berkeley, CA, USA. [2]Li Ka Shing Center for Biomedical and Health Sciences, University of California, Berkeley, Berkeley, CA, USA. [3]CIRM Center of Excellence, University of California, Berkeley, Berkeley, CA, USA. [4]Howard Hughes Medical Institute, University of California, Berkeley, Berkeley, CA, USA. [5]Center for Computational Biology, University of California, Berkeley, Berkeley, CA, USA. [6]Department of Biological Engineering, Massachussets Institute of Technology, Cambridge, MA, USA. [7]These authors contributed equally: Claudia Cattoglio, Elena Slobodyanyuk. ✉e-mail: darzacq@berkeley.edu; jmlim@berkeley.edu

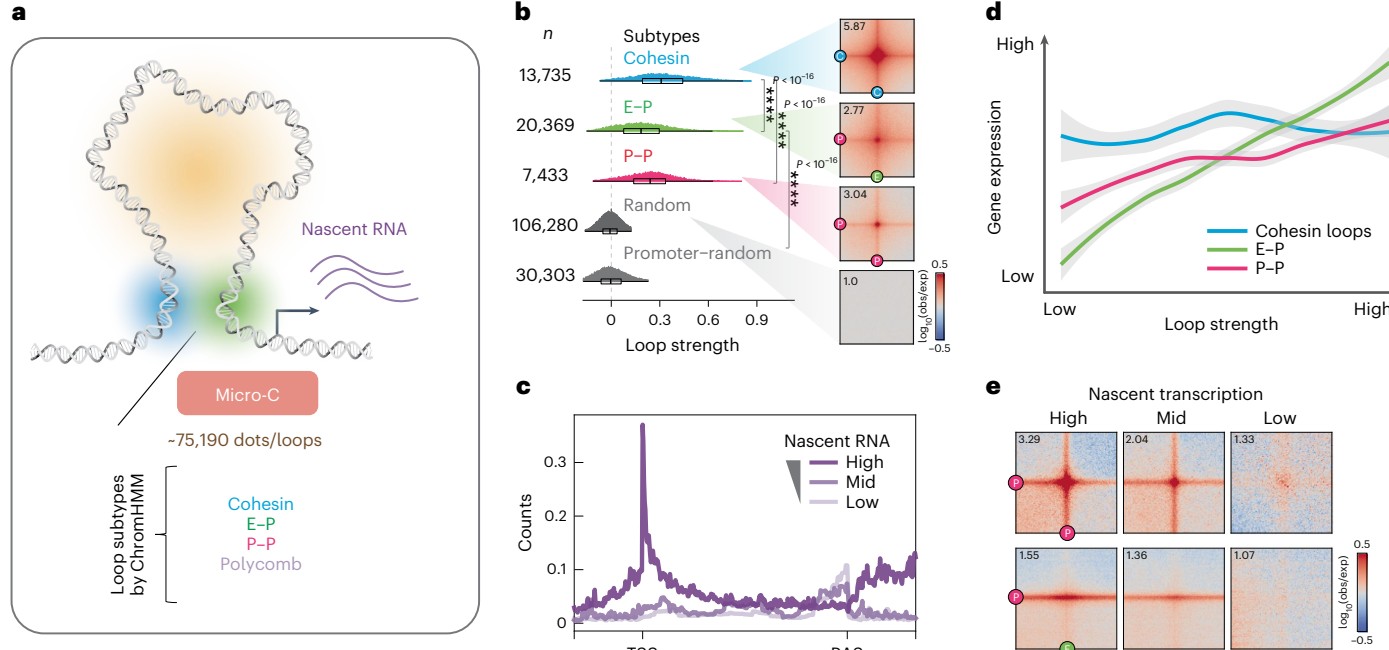

**Fig. 1 | Genome-wide identification of transcription-linked chromatin loops. a**, Micro-C identified >75,190 chromatin dots/loops, subclassified into four primary types (Mustache loop caller[26]; see Methods and Supplementary Note). **b**, Probability distribution of loop strength for cohesin, E–P, P–P and random loops. Chromatin loop numbers are shown on the left. The box plot indicates the quartiles for the loop strength score distribution (min. = lower end of line, Q1 = lower bound of box, Q2 = line in box, Q3 = higher bound of box and max. = higher end of line). Genome-wide averaged contact signals (aggregate peak analysis (APA)) are plotted on the right. The contact map was normalized by matrix balancing and distance (Obs/Exp), with positive enrichment in red and negative signal in blue, shown as the diverging color map with the gradient of normalized contact enrichment in $\log_{10}$. The ratio of contact enrichment for the center pixels is annotated within each plot. This color scheme and normalization method are used for normalized matrices throughout the manuscript unless otherwise mentioned. Loop anchors are annotated as 'C' for CTCF/cohesin, 'P' for promoter and 'E' for enhancer. Asterisks denote a $P < 10^{-16}$ using two-sided

Wilcoxon's signed-rank test. The data are presented in the same format and color scheme throughout the manuscript unless otherwise indicated ($n = 37$ biological replicates)[9]. **c**, Genome-wide averaged transcript counts for nascent transcript profiling. Genes are grouped into high, medium and low expression levels based on nascent RNA-seq data (gene body) and rescaled to the same length from TSS (transcription start site) to poly(adenylation) cleavage site (PAS) or TES (transcription end site) on the $x$ axis. **d**, Rank-ordered distribution of loop strength against gene expression for cohesin, E–P and P–P loops. Gene expression levels for the corresponding chromatin loop were calculated by averaging the genes with TSSs located ±5 kb around the loop anchors. Loop strength was obtained from the same analysis shown in **b**. The distribution for each loop type was fitted and smoothed by LOESS (locally estimated scatterplot smoothing) regression. Error bands indicate fitted curve ± s.e.m. with 95% confidence interval (CI). **e**, APAs are plotted by paired E–P/P–P loops and sorted by the level of nascent transcription into high, mid and low levels.

acutely depleting CTCF, RAD21 (cohesin subunit), WAPL (cohesin unloader) or YY1 (a putative structural protein[25]) affects gene regulatory chromatin interactions and transcription in mouse embryonic stem cells (mESCs). Finally, focusing on the dynamics of YY1 uncovered an unexpected role for cohesin in facilitating TF binding.

## Results

### Genome-wide identification of transcription-linked chromatin loops

Our previous study used Micro-C to reveal that fine-scale 3D genome structure correlates well with transcriptional activity, forming 'dots' or 'loops' (see Methods for terminology) at E–P and P–P intersections[9]. In the present study, we identified over 75,000 statistically significant loops in mESCs using the newly developed loop caller Mustache[26] (Fig. 1a) or Chromosight[27] (Extended Data Fig. 1a), approximately 2.5× more than in our previous report[9,26] and about 4× more than Hi-C[26,28] (Extended Data Fig. 1b). Through analysis of local chromatin state at loop anchors (Extended Data Fig. 1c,d), we subclassified these loops into cohesin loops (~13,735), E–P loops (~20,369), P–P loops (~7,433) and polycomb-associated contacts (~700) (Fig. 1a,b), with a median size of ~160 kb for cohesin loops and ~100 kb for E–P/P–P loops (Extended Data Fig. 1e).

We profiled nascent transcription by mammalian native elongating transcript sequencing (mNET-seq)[24] in mESCs to better understand

the relationship between active transcription and chromatin loops (Fig. 1c and Extended Data Fig. 1f). Newly transcribed RNAs generally have a higher correlation with E–P contacts than with compartments and TADs (Extended Data Fig. 1g). Specifically, the strength of E–P/P–P loops positively correlates with the level of gene expression, whereas cohesin loops show no such correlation (Fig. 1d,e). Thus, by coupling Micro-C with nascent RNA-seq, we can more precisely delineate which chromatin loops are associated with active transcription in a cell type of interest.

We note that Micro-C assay is superior to Hi-C at detecting E–P/P–P contacts (Extended Data Fig. 1h)[26,27], as illustrated by the region around the *Klf2* gene (Fig. 2a and Extended Data Fig. 1i), providing a less biased method for studying genome organization relevant to transcription regulation[29] (see Supplementary Note).

### E–P/P–P loops can cross TAD boundaries

TAD boundaries formed by CTCF and cohesin are thought to regulate E–P/P–P interactions in two ways: by increasing interactions inside the TAD and by blocking interactions across TADs[2]. Nevertheless, it remains debatable whether TAD boundaries can absolutely prevent an enhancer from interacting with and activating a gene in another TAD[30–34]. Our genome-wide analysis uncovered that, although loop interactions largely decay across distance (Extended Data Fig. 1j), ~22.5% of E–P and ~33.2% of P–P loops that cross TAD boundaries retain a comparable level

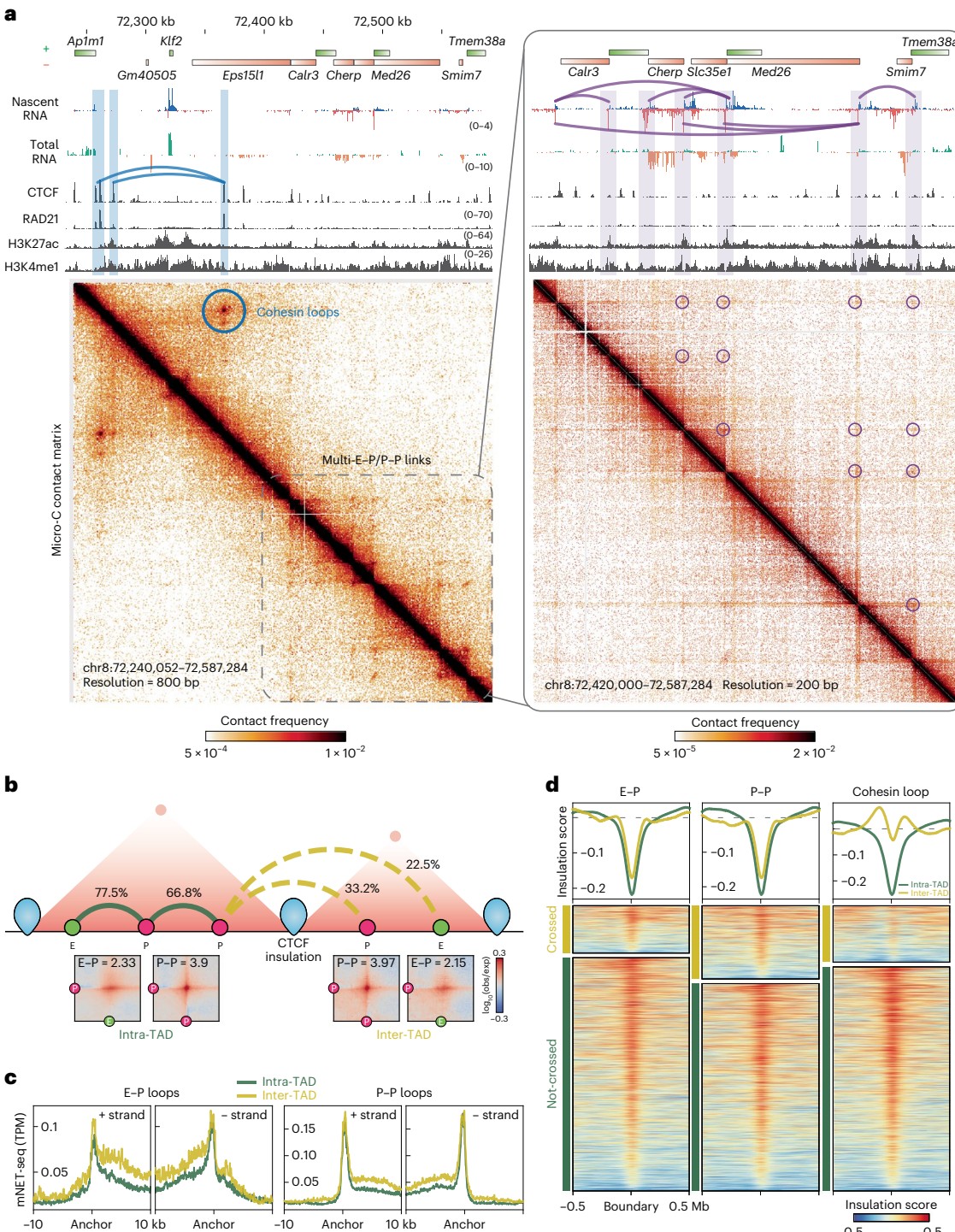

**Fig. 2 | E–P/P–P loops can cross TAD boundaries. a**, Snapshots of Micro-C maps of an ~300-kb region plotted with 800-bp resolution (left) and an ~150-kb region plotted with 200-bp resolution (zoomed-in, right). Micro-C data are reanalyzed from our previous study[9]. The standard heatmap shows the gradient of contact intensity for a given pair of bins. This color scheme is used for Micro-C maps throughout the manuscript. Contact maps are annotated with gene boxes and 1D chromatin tracks show the signal enrichment in the same region. Features such as cohesin loops (blue arched lines and circles) and E–P/P–P loops (purple arched lines and circles) enriched at stripe intersections are highlighted. The CTCF and cohesin ChIP-seq peaks show strong contact signals between the *Ap1m1* and *Eps15l1* genes (blue arched lines and circle), which insulate the *Klf2* gene from communicating with regions outside the loop domain. However, multiple weak interactions within the downstream 150-kb region around the *Med26* gene still occur without apparent cohesin residency at their anchors (purple arched lines

and circles), and these contacts sharply correlate with nascent transcription signals at promoters and enhancers. **b**, Schematic (top) showing two adjacent TADs insulated by CTCF boundaries and E–P/P–P interactions either within a TAD (intra-TAD, solid arched line) or across TADs (inter-TAD, dashed arched line). E–P/P–P contact intensity was quantified with the Micro-C data at 2-kb or 4-kb resolution. TADs called by Cooltools and Arrowhead returned similar results for the ratio of boundary-crossing E–P/P–P (see Methods). APA (bottom) is plotted for paired E–P/P–P that either cross (inter-TAD) or do not cross (intra-TAD) a TAD boundary. **c**, Nascent transcription (± strand) at the loop anchors of intra- (green) or inter-TAD (yellow) E–P/P–P loops. TPM, transcripts per million. **d**, Heatmap and histogram profile of insulation scores at 20-kb resolutions spanning the 1-Mb window for intra- (green) or inter-TAD (yellow) E–P/P–P loops. Color map shows strong insulation in red and weak insulation in blue in log₁₀.

of contact intensity to equidistant loops within a TAD (Fig. 2b). Genes located at the anchors of these inter-TAD loops also show similar or even higher expression levels in nascent or total RNA analysis (Fig. 2c).

We postulated two possibilities that could lead to this observation: the TAD boundaries that are crossed by E–P/P–P loops have either lower CTCF/cohesin occupancy or weaker insulation propensity. We first split the TAD boundaries into two groups: 'crossed' or 'not crossed' by loops. Strikingly, CTCF and RAD21 occupancy at the boundaries is almost the same regardless of whether the boundaries are crossed by loops (Extended Data Fig. 1k). The TAD boundaries crossed by either E–P or P–P loops show only slightly weaker insulation strength than the noncrossed boundaries (Fig. 2d). In contrast, the boundaries that insulate the cohesin loops are substantially stronger than those that allow their crossing (Fig. 2d). Together, these results indicate that TAD boundaries are much more effective at insulating cohesin loops than insulating E–P/P–P loops and that strong E–P interactions can overcome structural barriers[18,35–37].

## Acute depletion of CTCF, cohesin or WAPL alter CTCF and cohesin binding on chromatin

To test whether active loop extrusion is essential for maintaining various types of chromatin loops and transcription, we endogenously and homozygously tagged each of the three primary loop extrusion factors (CTCF, RAD21 or WAPL) with an auxin-inducible degron (AID) by clustered regularly repeating interspaced short palindromic repeats (CRISPR)–Cas9-mediated genome editing in mESC lines expressing the F-box protein OsTir1 (Fig. 3a and Extended Data Fig. 2a)[38]. Despite CTCF-AID and RAD21–AID cell lines showing some basal degradation (Extended Data Fig. 2b,c), we found no substantial change in their chromatin association, 3D genome organization or transcriptome compared with wild-type cells (see Supplementary Note). Previous studies employing acute CTCF/cohesin depletion used prolonged degradation (6–48 h (refs. [12,13,39]), which may confound the primary molecular response with potential secondary effects[40]. To minimize indirect effects, we used a shorter degradation time and achieved almost-complete degradation of AID-tagged proteins after 3 h of iodoacetamide (IAA) treatment, confirmed by western blotting (Fig. 3b and Extended Data Fig. 2d) and biochemical fractionation experiments (Extended Data Fig. 2e,f, red box).

We then asked how the loss of each loop extrusion factor affects the binding of the remaining factors. We obtained high-quality and high-reproducibility ChIP-seq data for CTCF, RAD21, SMC1A and SMC3 in the AID-tagged lines treated with either ethanol (untreated (UT)) or IAA to degrade the tagged protein (Extended Data Fig. 3a–d). Consistent with previous studies[12,41], both CTCF and cohesin lose their occupancy after CTCF depletion (Fig. 3c,d and Extended Data Fig. 3e,f). Differential peak analysis[42] confirmed that >90% of CTCF peaks and 60% of cohesin peaks are significantly decreased on loss of CTCF ($P_{adj} < 0.05$; Fig. 3e and Extended Data Fig. 3g). Despite the substantial loss of cohesin peaks, biochemical fractionation experiments show that the fraction of RAD21 associated with chromatin remains fairly constant 3 h after CTCF degradation (Extended Data Fig. 2f, green box). Thus, our results are in line with the widely accepted conclusion that CTCF positions cohesin[43]. On the other hand, loss of cohesin affects a subset of CTCF binding (Fig. 3c,d)[13], resulting in ~20% reduction in the number of CTCF peaks (Fig. 3e) and a slight decrease in its global chromatin association (Extended Data Fig. 2f, blue box).

## Acute depletion of CTCF, cohesin and WAPL perturbs structural loops

Next, we used Micro-C to analyze the effect of CTCF, RAD21 and WAPL depletion on fine-scale 3D genome structures. We pooled the highly reproducible replicates to achieve ~1–2 billion unique reads for each sample (Extended Data Fig. 4a–c). At the levels of compartments and TADs, our findings largely agree with previous studies[12–14,16] (Extended

Data Fig. 4d,e). In addition, loop-strength analysis revealed that nearly 90% of cohesin loops were lost after depletion of CTCF or RAD21 (Fig. 3f,g), whereas most loops were retained in a similar or slightly higher strength after WAPL depletion (Fig. 3g and Extended Data Fig. 4f). Indeed, after WAPL depletion, an additional ~6,000 loops extended over longer distances (median size = 570 kb) were sufficiently strengthened to meet our detection threshold (Extended Data Fig. 4g,h). In summary, cohesin-mediated DNA extrusion operates in a more unrestricted manner after depletion of CTCF (loss of well-positioned loops) or WAPL (gain of longer-range loops).

## Acute loss of CTCF, cohesin and WAPL does not affect expression of most genes

We next asked whether acute disruption of active loop extrusion impacts the maintenance of gene expression. To capture the immediate and temporal effects of depleting loop extrusion factors on transcription, we profiled nascent transcription by mNET-seq[24] and messenger RNA by ribosomal RNA-depleted RNA-seq for untreated and IAA-treated degron lines at 0, 3, 12 and 24 h after depletion. After validating the reproducibility and sensitivity of the methods[44] (Extended Data Fig. 5a,b), we performed differential expression tests of ~30,000 genes and identified ~50 transcripts changed in CTCF depletion, ~5 changed in RAD21 depletion and only 2 changed in WAPL depletion after 3 h of IAA treatment (Fig. 3h,i and Extended Data Fig. 5c,d). Differentially expressed genes (DEGs) became more numerous with longer degradation times, in line with previous findings[12,39,45] (Extended Data Fig. 5c–e).

We noticed that the early deregulated genes after loss of CTCF and cohesin include many cell-type-specific TFs (for example, *Sox21*, *Myc* and *Klf4*; Fig. 3i and Extended Data Fig. 5f). Chromatin structures around the DEGs were strongly disrupted, often featuring loss of a boundary or domain and gain of de novo chromatin interactions (Fig. 3j and Extended Data Fig. 5g). Indeed, the early DEGs are associated with loop anchors and TAD boundaries, whereas the DEGs detected at the later time points are not (Fig. 3k). This finding highlights the importance of distinguishing between primary and indirect effects of perturbations in the study of 3D genome and gene expression[40].

In summary, despite CTCF, cohesin and WAPL probably regulating some gene expression in mESCs, their acute depletion affects the transcription of only a handful of genes that mostly encode pluripotency and differentiation factors.

## E–P and P–P interactions are largely maintained after degradation of loop extrusion factors

The very modest transcriptional changes seen after CTCF and cohesin degradation suggest that transcription-linked interactions may persist for at least 3 h after the depletion of CTCF, cohesin or WAPL. To test this hypothesis, we quantified the loop strength at all 75,000 dots identified in wild-type mESCs in both control and depletion conditions. About 20% of loops are significantly decreased, but the remaining 60,000 loops are largely unaltered (Fig. 4a,b). Consistent with our previous results, the disrupted loops are CTCF or cohesin dependent, whereas the persistent and upregulated loops are mostly anchored by promoters and enhancers (Fig. 4c and Extended Data Fig. 6a,b). To further validate this, we specifically quantified the strength of loops that are anchored by E–P/P–P. Remarkably, acute depletion of CTCF and cohesin has only a limited impact on the E–P/P–P loops, with ~80% of E–P contacts and 90% of P–P contacts remaining unaltered (Fig. 4d). Despite being less drastic than for cohesin loops (Fig. 3f), E–P interactions appear to be slightly weakened globally, deviating from the midpoint line, but P–P loops remain largely insensitive to CTCF/cohesin depletion (Fig. 4d,e and Extended Data Fig. 6c,d). WAPL depletion also has a negligible impact on E–P/P–P interactions (Fig. 4e and Extended Data Fig. 6e,f).

What are the CTCF-/cohesin-sensitive E–P/P–P loops? We found that these loops span a longer distance (Extended Data Fig. 6g) and have higher CTCF and cohesin occupancy at their anchors (Extended

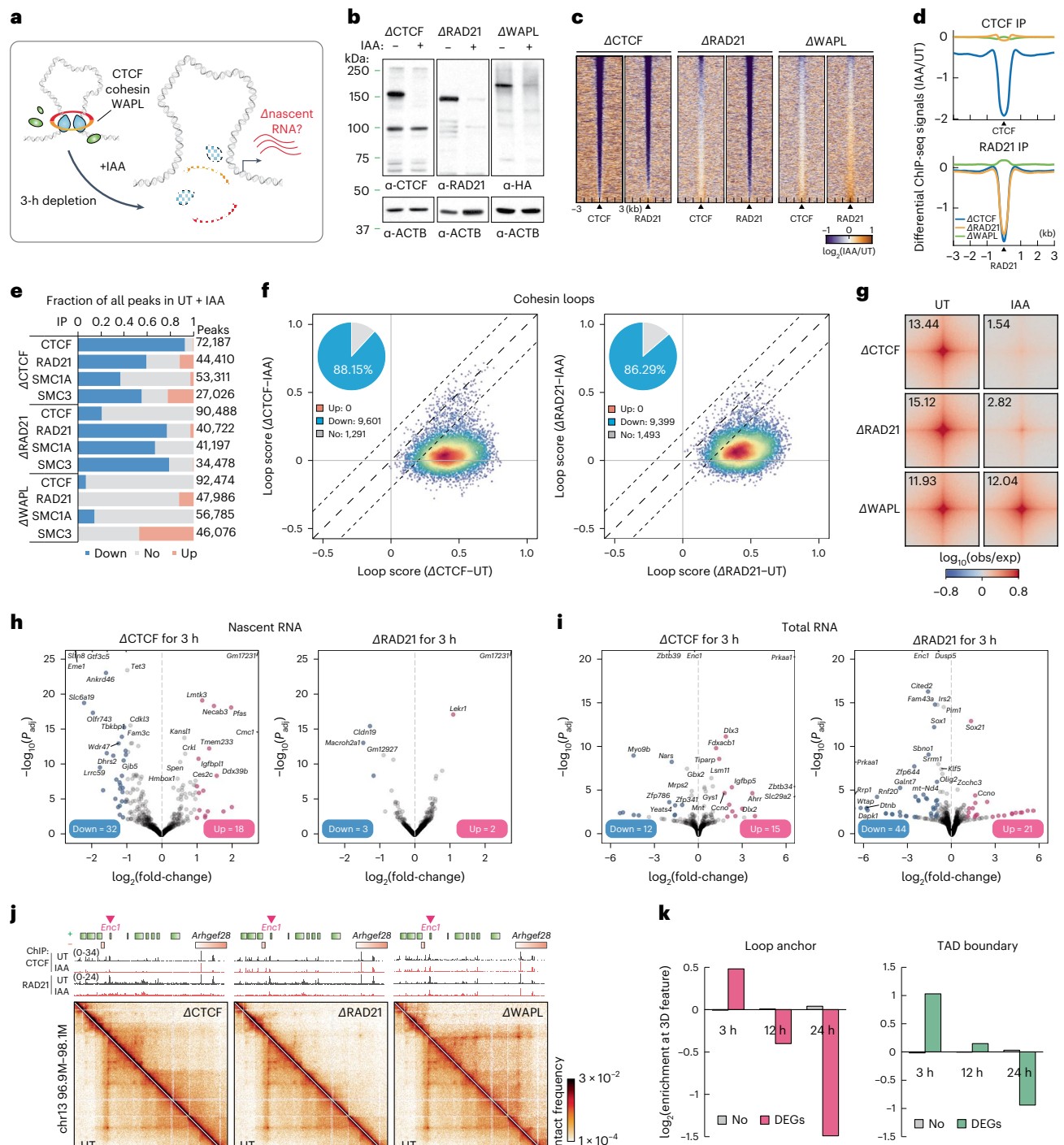

**Fig. 3 | Acute depletion of loop extrusion factors affects a small set of genes.**
**a**, Experimental design for CTCF, RAD21 or WAPL degradation. **b**, Western blots showing CTCF, RAD21 and WAPL degradation levels, and β-actin loading control 3 h after IAA treatment. **c**, CTCF and RAD21 differential ChIP-seq signals in cells depleted of CTCF, RAD21 or WAPL. MACS2 (model-based analysis of ChIP-seq 2)-called peaks are plotted at the center ± a 3-kb region. The color map shows increased signal (log₂) in orange and decreased signal in purple after IAA treatment. Data are not normalized with a spike-in control. **d**, Histogram profile of differential CTCF or RAD21 ChIP-seq signals in CTCF-, RAD21- or WAPL-depleted cells. **e**, Summary of differential ChIP-seq peak analysis. The chart shows the fraction of downregulated, upregulated or unchanged peaks after IAA treatment. The total number of peaks for each protein was summed from all peaks in untreated and IAA-treated cells. **f**, Scatter plots of loop scores for cohesin loops in untreated and IAA-treated cells. The loop score was quantified by using 2-kb Micro-C data. The overlaid heatmap indicates dot density (red,

highest; blue, lowest). Dashed lines along the diagonal delimit unchanged loops. The pie chart (inset) shows the fraction of increased, decreased or unchanged loop intensity after IAA treatment. Scatter plots comparing loop intensities between two conditions are plotted in this format throughout the manuscript unless noted. **g**, APAs plotted for paired cohesin peaks for untreated and IAA-treated cells. **h,i**, Volcano plots of nascent (**h**) or total (**i**) RNA-seq for CTCF or RAD21 depletion. DEGs with *q* value <0.01 and twofold change are labeled pink (up) or blue (down). The statistical tests for all RNA-seq and mNET-seq in the present study are obtained from the statistical model derived from DEseq2 unless otherwise indicated. **j**, Micro-C maps comparing chromatin interactions in untreated (top right) and IAA-treated (bottom left) cells surrounding *Enc1*. Contact maps are annotated with gene boxes and 1D chromatin tracks showing the ChIP-seq signal enrichment in the same region. **k**, Bar graph with log₂(enrichment) of unaffected genes (No) or DEGs ±5 kb around loop anchors (left) or TAD boundaries (right).

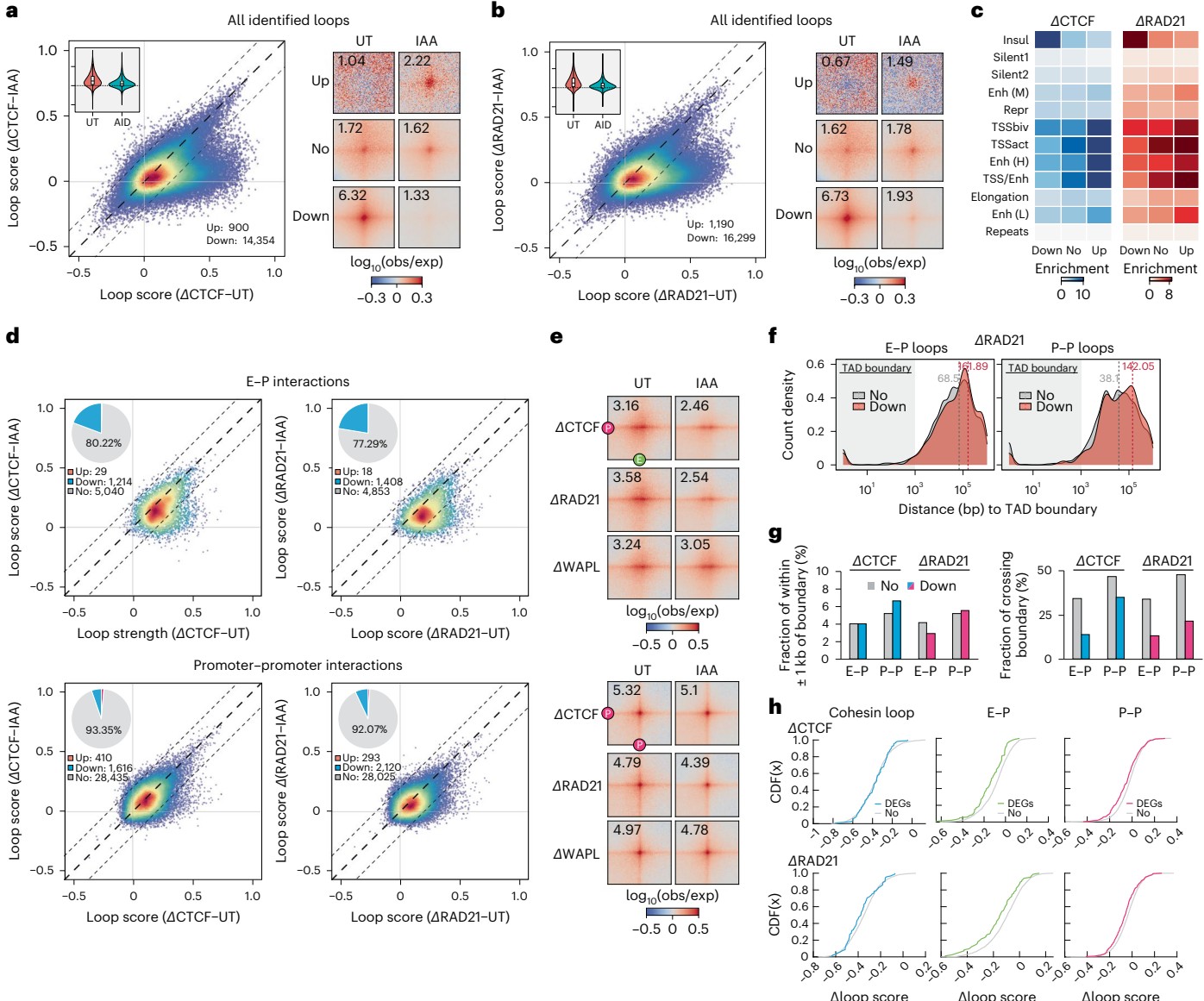

**Fig. 4 | E and P proximity persists after the acute loss of loop extrusion factors. a,b,** Scatter plots of loop scores for the loops called in untreated and IAA-treated ΔCTCF (**a**) and ΔRAD21 (**b**) degron cell lines (left). The loop score was quantified using Micro-C data at 2-kb resolution. The violin chart (inset) shows the distribution of loop scores for the untreated and IAA-treated conditions. The box plot distribution is described in Fig. 1b. APAs are plotted with loops sorted by upregulated (Up), downregulated (Down) or unchanged (No) loops (right; control = 2; IAA = 4 biological samples). **c,** Enrichment of ChromHMM states at loop anchors grouped by upregulated, downregulated or unchanged after IAA treatment. **d,** Scatter plots of loop scores plotted for paired E–P (top) or P–P (bottom) loops in the untreated and IAA-treated cells. Pairwise loops are limited to distances between 5 kb and 2 Mb. The loop score was quantified using Micro-C data at 2-kb resolution. The pie chart (inset) shows the fraction of loops with intensity increased, decreased or unchanged after IAA treatment. Note

that the average contact intensity of unchanged loops decreased by ~2.4% and ~8.0% after CTCF and cohesin depletion, respectively. **e,** APAs for E–P (top) or P–P (bottom) loops plotted for the indicated untreated and IAA-treated cell lines. When CTCF/cohesin is depleted, the contact intensity is decreased by ~22.2% or ~29.1% for E–P loops, but only ~4.1% or ~8.4% for P–P loops. **f,** Length distribution of the unchanged or downregulated E–P/P–P loops relative to TAD boundaries in the RAD21 degron line. **g,** Ratio of the unchanged or downregulated E–P/P–P loop anchors located within ±1 kb of TAD boundaries (left) or that can cross TAD boundaries (right). **h,** Cumulative distribution (CDF) curves as a function of differential loop score (IAA, untreated) for all loops or loop anchors within 1 kb of the promoter of unchanged genes (No) or DEGs in CTCF and RAD21 degron lines. A CDF curve shift to the left indicates a greater reduction in interaction frequency on IAA treatment.

Data Fig. 6h), but these anchors are not specifically associated with TAD boundaries (Fig. 4f,g and Extended Data Fig. 6i). We further tested whether the affected E–P/P–P interactions were associated with the DEGs in nascent RNA-seq. Indeed, E–P/P–P interactions showed a greater decrease when their associated genes were deregulated on loss of CTCF/cohesin (Fig. 4h).

Together, most E–P/P–P contacts and fine-scale gene folding largely persist and remain transcriptionally functional even after almost-complete depletion of CTCF, cohesin or WAPL, suggesting that,

in mESCs, these proteins are not strictly required to maintain E–P/P–P interactions and transcription at least within a 3-h degradation, despite a broad but weak reduction in E–P interactions after cohesin depletion.

## Acute YY1 depletion has little effect on global gene expression and E–P/P–P interactions

A multifunctional zinc finger-containing TF, YY1 (ref. [46]) (Extended Data Fig. 7a), has been implicated as a master structural regulator of chromatin looping[25], particularly during early neural lineage commitment[47].

To investigate the function of YY1 in genome organization and transcriptional regulation in mESCs, we fused the mini-IAA7 tag[48] to the endogenous *Yy1* locus to allow for rapid protein degradation within 3 h (Fig. 5a and Extended Data Fig. 7b). ChIP-seq analysis showed a clear depletion of YY1 at its cognate sites (Fig. 5b and Extended Data Fig. 7c), with ~90% of peaks (*n* = 34,342) being called significantly changed by differential peak analysis[42] (Fig. 5b and Extended Data Fig. 7d,e). These peaks are primarily enriched at promoters, enhancers and bivalent loop anchors (Fig. 5c and Extended Data Fig. 7f), consistent with its reported role in E–P interactions. We also noticed a modest decrease in cohesin occupancy after loss of YY1 (Fig. 5b and Extended Data Fig. 7g), which may be associated with YY1's potential to position or halt cohesin[25]. Similarly, biochemical fractionation analysis shows a decrease of ~87% in the chromatin-associated YY1 fraction and a reduction of ~7% in the chromatin-associated cohesin fraction (Extended Data Fig. 7h, orange box).

To characterize YY1's role in 3D genome organization, we acquired ~850 × 10^6 unique Micro-C reads after pooling high-quality replicates from untreated or YY1-depleted cells (Extended Data Fig. 7i). We found that YY1 depletion has no strong effect on chromatin compartments, TADs and cohesin loops (Fig. 5d and Extended Data Fig. 7j). YY1 was proposed to be a causally required structural regulator of transcription and E–P interactions in a study conducted with 24-h depletion in mESCs[25]. Surprisingly, acute removal of YY1 only mildly affected ~1% of loops (Fig. 5e) and ~11 and ~34 genes in the RNA-seq and mNET-seq profiling, respectively (Fig. 5f and Extended Data Fig. 7k). Genome-wide pileup analysis for YY1, E–P and P–P loops showed only a very minor change in loop intensity after YY1 depletion (Fig. 5g and Extended Data Fig. 7l). Nevertheless, a specific set of loci appears to require the presence of YY1 to interact with their *cis*-regulatory elements (for example, the *Ifnar2*, *Ikzf2* and *NES* gene loci) (Fig. 5h and Extended Data Fig. 7m). Taken together, although YY1 may be required for a limited set of E–P/P–P interactions, YY1 is generally dispensable for maintaining genome organization and transcription in mESCs, at least within a 3-h depletion window.

## Single-molecule imaging reveals YY1 binding dynamics and nuclear organization

The surprisingly modest effects of YY1 on chromatin looping might result from YY1 DNA binding being very transient and/or due to only a small fraction of YY1 proteins being bound to DNA. To better understand the dynamics and mechanisms underlying YY1 function in living cells, we homozygously tagged YY1 with HaloTag[49] (designated YN11 and YN31 clones) for live-cell, single-molecule imaging using CRISPR–Cas9-mediated genome editing (Fig. 6a and Extended Data Fig. 8a–c). Live-cell confocal imaging validated that HaloTag-YY1 was predominantly localized within the nucleus and appeared to be non-homogeneously distributed throughout the nucleoplasm, with noticeable puncta sporadically clustered within nucleoplasm and nucleoli (Fig. 5b,c and Extended Data Fig. 8d). We then visualized the nuclear distribution of YY1 at single-molecule resolution by using photoactivated localization microscopy (PALM) (Fig. 6d), confirming its high-density punctate clusters. Furthermore, YY1 has been thought to be evicted from chromosomes during mitosis in fixed-cell imaging experiments[50]. However, our live-cell imaging showed continued YY1 residency on mitotic chromosomes, suggesting that YY1 may be involved in mitotic bookmarking (Fig. 6b and Extended Data Fig. 8d)[51]. Together, these results validate our homozygous HaloTag-YY1 knock-in cell lines and reveal that YY1 binds mitotic chromosomes and forms local high concentration hubs in the nucleus.

Having characterized our cell lines, we next interrogated YY1 protein dynamics and target search mechanisms. We took advantage of the stroboscopic photoactivation, single-particle-tracking technique (spaSPT)[52,53] to minimize motion blur and tracking errors to unambiguously trace the movement of individual YY1 molecules at a frame rate

of ~133 Hz (Fig. 6e and Extended Data Fig. 8e). YY1 molecules were then subclassified into bound and freely diffusing populations using a Bayesian-based approach[54] (Extended Data Fig. 8e,f). We found that ~31% of YY1 is in an immobile state, presumably bound to chromatin, with the remaining population exhibiting either slow diffusion (~26%) or fast diffusion in the nucleoplasm (~43%) (Fig. 6f). These measurements largely agree with kinetic modeling of displacements obtained with the Spot-On algorithm (Extended Data Fig. 8g)[53] and biochemical fractionation experiments (Fig. 6g). We note that the fraction of YY1 stably associating with chromatin is substantially lower than CTCF (~43%) and cohesin (~65%).

The residence times of TFs bound at their targets often correlate with their functional outcomes[55–57]. To estimate the overall residence time of the bound fraction of YY1, we used fluorescence recovery after photobleaching (FRAP) to measure in vivo protein-binding kinetics by fitting the fluorescence recovery curve to a kinetic model[58,59]. Using a reaction-dominant FRAP model, we estimated a residence time of ~13 s for most of the YY1 molecules (Fig. 6h and Extended Data Fig. 8h,i). We also employed slow-SPT[60] as an orthogonal approach to measure YY1 residence times and obtained a residence time of ~13 s for YY1 at an exposure time of 100 ms (Fig. 6i). Slow-SPT with exposure times from 50 ms to 250 ms (ref. [61]) further revealed a subpopulation of YY1 that binds to chromatin for 40–60 s (Fig. 6j), consistent with the FRAP results showing that ~15% of YY1 recovers slowly (Fig. 6h and Extended Data Fig. 8h,i). Thus, YY1 proteins appear to have two distinct binding modes with apparent residence times of ~13 s and ~1 min.

Taken together, our imaging experiments suggest that a smaller fraction of YY1 (~31%) is bound to chromatin and that YY1 binding is more dynamic (average residence time of ~13 to 60 s) than CTCF (~50% bound for ~1 to 4 min) and cohesin (~40 to 50% bound for ~20 to 25 min), which may help explain why YY1 protein depletion has a much weaker effect on looping and 3D genome folding.

## Cohesin depletion alters TF chromatin-binding kinetics

We recently showed that CTCF clusters enrich diffusive CTCF proteins near their binding sites, thereby accelerating their target search[62]. To test whether CTCF and cohesin may similarly affect YY1's target search, we endogenously fused an AID to CTCF or RAD21 in the HaloTag-YY1 parental line and confirmed >90% depletion after 3 h of IAA treatment (Fig. 7a and Extended Data Fig. 9a). Despite the high degradation efficiency, neither YY1's nuclear distribution nor its clustering was strongly affected after acute loss of CTCF and cohesin in either live or fixed cells (Fig. 7b,c and Extended Data Fig. 9b). This suggests that the maintenance of YY1 hubs is independent of CTCF and cohesin.

We next examined YY1 nuclear target search efficiency in the absence of CTCF and cohesin using spaSPT. Although CTCF depletion had no major effect, cohesin depletion resulted in a modest but reproducible decrease from ~33% to 22% (~31% drop; *P* < 0.01) in the bound fraction of YY1 (Fig. 7d and Extended Data Fig. 9c). A lower bound fraction could either result from a shorter residence time ($k_{off}$) or slower target search ($k_{on}$). To distinguish between these possibilities, we analyzed the FRAP data and found YY1 residence times ($k_{off}$) to be only weakly affected by CTCF and cohesin depletion (Fig. 7e). We therefore conclude that cohesin loss may affect the YY1 target search ($k_{on}$). Specifically, we estimated a ~54% decrease in $k_{on}$ after cohesin depletion, resulting in a ~2.2-fold longer YY1 search time (UT = 28 s; IAA = 61 s), the time it takes YY1 on average to find and bind a cognate binding site after dissociating from DNA.

To independently test this SPT finding, we analyzed our ChIP-seq data. We found that ~3,504 YY1 peaks (total peaks = ~41,989 (~8.3%)) were lost after RAD21 degradation and >82% of these loci were associated with promoter regions (Fig. 7f and Extended Data Fig. 9d,e). In contrast, both CTCF and WAPL depletion had a negligible effect on YY1 occupancy (Fig. 7f and Extended Data Fig. 9d,e). In biochemical fractionation analysis, we also observed a similar, though less

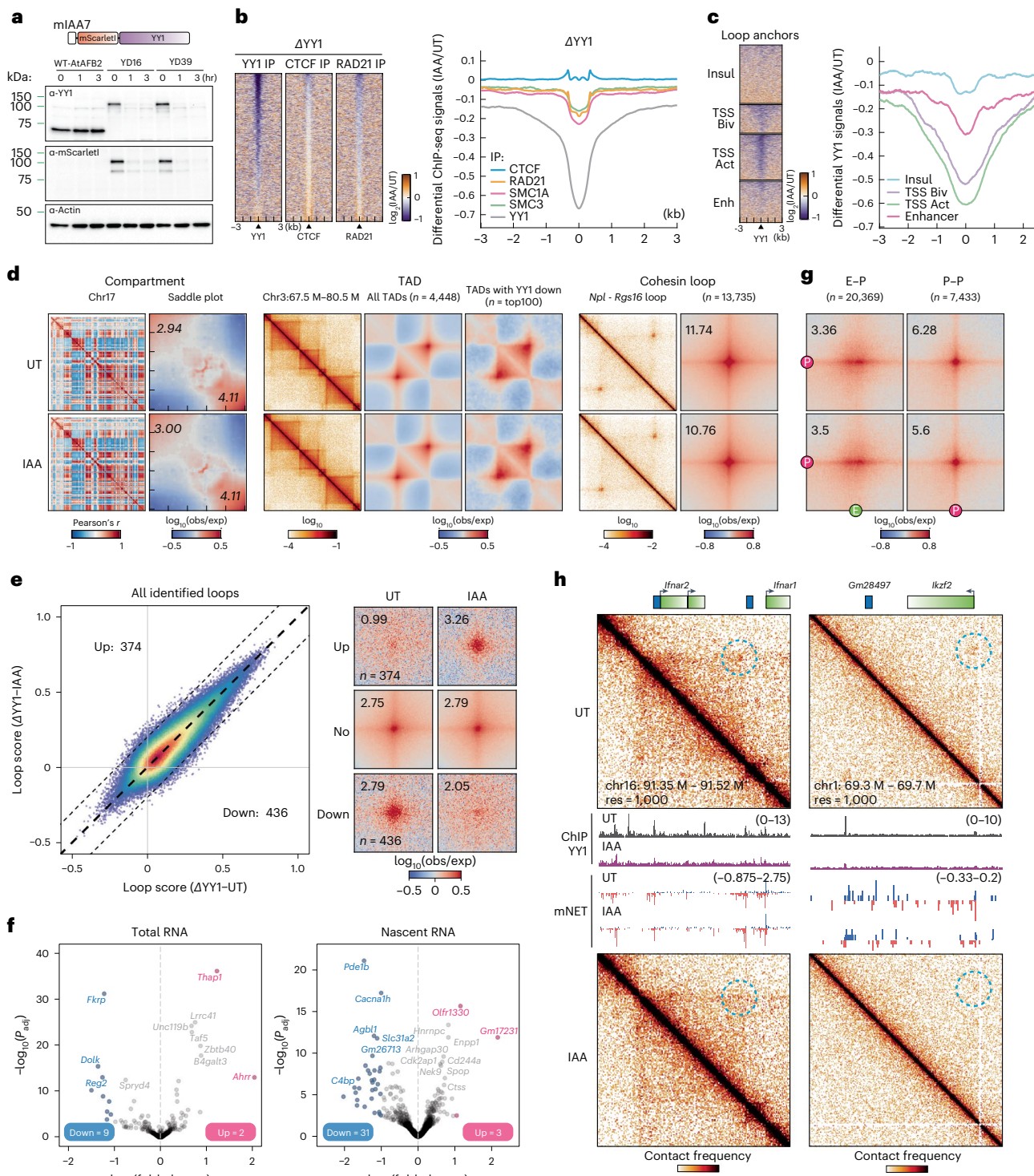

**Fig. 5 | YY1 depletion does not immediately alter global gene expression and E–P/P–P proximity. a**, Schematic for endogenous tagging for YY1 depletion and the results of western blots for YY1 and β-actin. **b**, Heatmaps (left) and histogram profiles (right) of differential ChIP-seq signals for YY1, CTCF and cohesin after YY1 depletion. **c**, Heatmaps (left) and histogram profiles (right) of differential ChIP-seq signals for YY1 around the four types of loop anchors. **d**, Overview of Micro-C contact maps at specific regions or genome-wide scale across multiple resolutions in the untreated and IAA-treated cells. Left to right, examples of Pearson's correlation matrices showing plaid-like chromosome compartments; saddle plots showing overall compartment strength (A–A: bottom right; B–B: top left); contact matrices showing TADs along the diagonal; aggregate domain analysis (ADA) showing all TADs; ADA showing TADs with downregulated YY1 ChIP-seq signals; contact matrices showing cohesin loops off the diagonal; and

APAs showing overall loop intensity for cohesin loops. **e**, Scatter plots of loop scores for the called loops in the untreated and IAA-treated cells (left). The loop score was quantified by using Micro-C data at 2-kb resolution. APAs are plotted with loops sorted by upregulated, downregulated or unchanged. **f**, Volcano plot of total RNA-seq (left) or nascent RNA-seq (right) for YY1 depletion. DEGs (q value <0.01 and twofold change) are colored in pink (up) or blue (down). **g**, APAs showing overall loop intensity for E–P/P–P loops in untreated and IAA-treated YY1 degron cells. **h**, Snapshots of Micro-C maps comparing chromatin interactions in the untreated (top) and IAA-treated (bottom) cells surrounding the *Ifnar2* or *Ikzf2* gene. Contact maps are annotated with gene boxes and genome browser tracks showing YY1 ChIP-seq signal enrichment and mNET-seq signals, with the plus strand in blue and the negative strand in red.

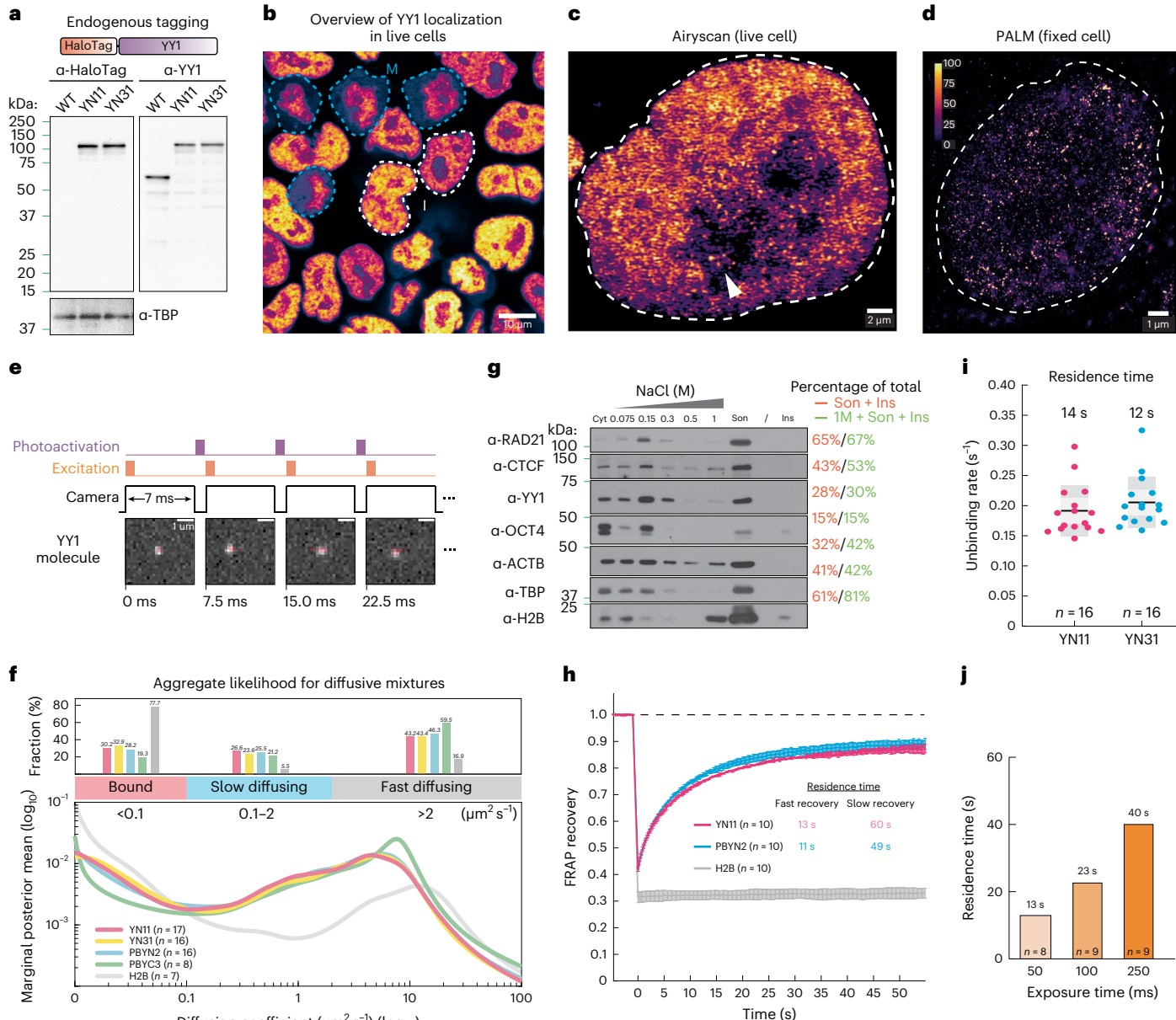

**Fig. 6 | YY1 binding dynamics. a**, Endogenously tagging YY1 with HaloTag and YY1 and TATA-box-binding protein (TBP) western blots. HaloTag is covalently conjugated with cell-permeable dyes for single-molecule imaging in live cells. **b**, HaloTag-YY1 live-cell confocal imaging after staining with 500 nM TMR Halo ligand. The white dashed lines show interphase cells and the blue dashed lines mitotic cells. Scale bar, 10 μm. **c**, YY1 Airyscan-resolved, live-cell confocal imaging ($n$ = 13). The arrow shows sporadic loci within the nucleolus. Scale bar, 2 μm. **d**, YY1 PALM imaging ($n$ = 30). The color map shows signal ranging from 0 to 100. Scale bar, 1 μm. **e**, The spaSPT illumination pattern and representative YY1 raw images with tracking overlaid. HaloTag-YY1 molecules were detected and tracked to form trajectories. The SASPT analysis package infers diffusion coefficient distributions from spaSPT data. Two major apparent diffusion states are a bound population (diffusion coefficient $D_{bound}$ < 0.1 μm² s⁻¹) and a mixture of freely diffusing molecules ($D_{free}$ > 0.1 μm² s⁻¹), which can be separated further into slow ($D_{slow}$ ~0.1–2 μm² s⁻¹) and fast moving ($D_{fast}$ > 2 μm² s⁻¹). Scale bar, 1 μm. **f**, Aggregate likelihood of diffusive YY1 molecules. Top, bar graph showing fractions of YY1 binned into bound, slow- and fast-diffusing subpopulations. Bottom, YY1 diffusion coefficient estimation by regular Brownian motion with marginalized localization errors. **g**, Western blots of cytoplasmic (Cyt) and nuclear proteins dissociating from chromatin at increasing salt concentrations (Extended Data Fig. 2b). A subpopulation (~30%) of YY1 stays on chromatin, resisting 1 M washes. Ins, insoluble pellet after sonication; Son, sonicated, solubilized chromatin. Percentage of total shows the signal intensity of the indicated fractions divided by the total signal intensity. Anti-histone 2B controls for chromatin integrity during fractionation. **h**, FRAP analysis of YY1 bleached with a square spot. Error bars are fitted curve ± s.e.m. with 95% CI. **i**, Slow-SPT measuring YY1 residence time. Individual molecules were tracked at 100-ms exposure time to blur fast-moving molecules into the background and capture stable binding. The unbinding rate is obtained by fitting a model to the molecules' survival curve. Each datapoint indicates the unbinding rate of YY1 molecules in a single cell. The box plot shows quartiles of data. Error bars are mean ± s.d. **j**, Slow-SPT measures YY1's residence time at multiple exposure times.

pronounced, reduction in YY1 chromatin association after RAD21 depletion (Extended Data Fig. 9f). To test whether cohesin facilitates the target search of TFs in general, we performed spaSPT on additional TFs. We thus generated RAD21–AID cell lines stably expressing either HaloTag-conjugated SOX2 or KLF4 and found that the bound fraction of both TFs was reduced by ~20% after 3-h cohesin degradation (Extended Data Fig. 9g). These results suggest that cohesin probably facilitates chromatin binding of TFs in general.

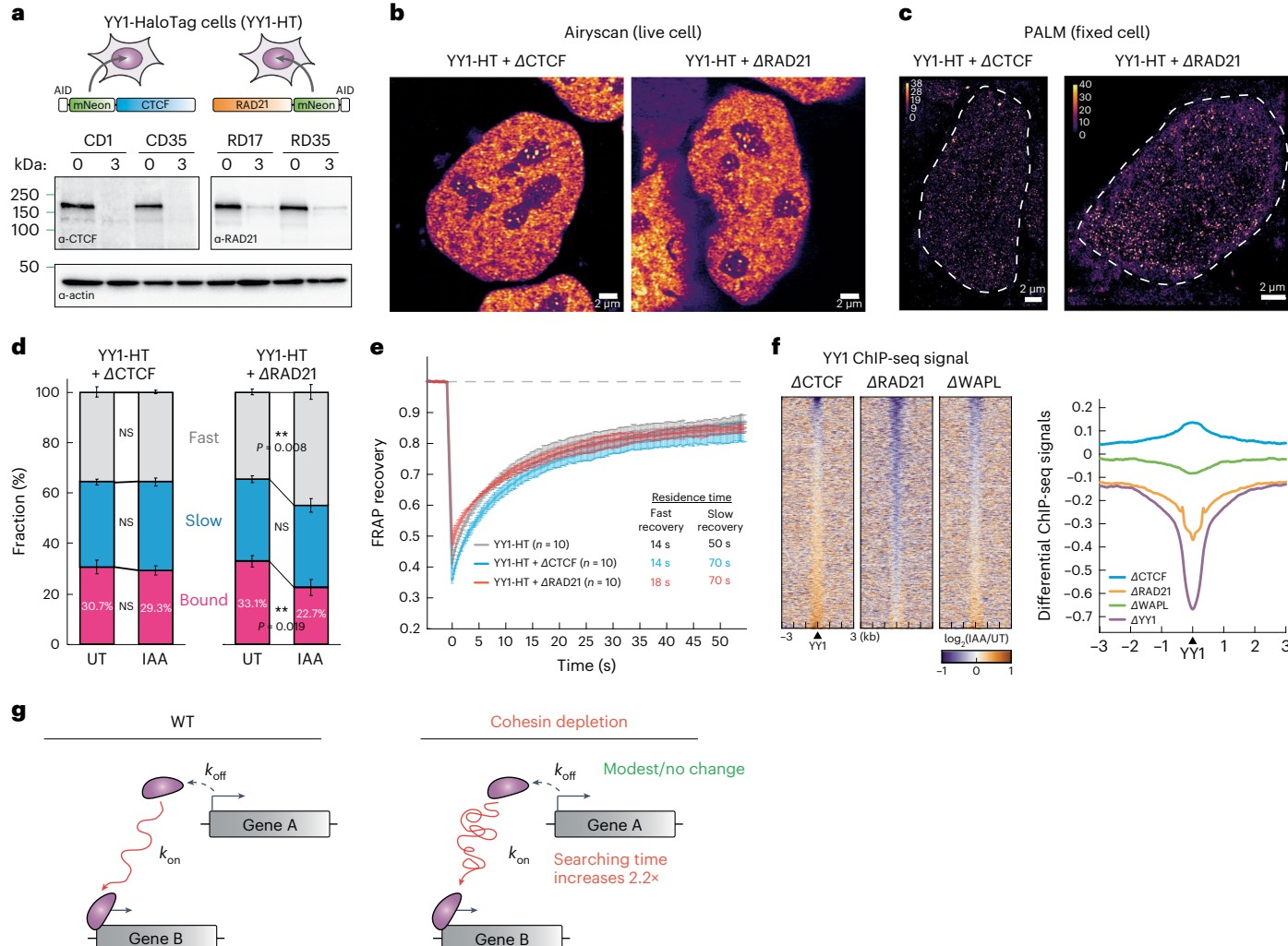

**Fig. 7 | Cohesin depletion alters YY1's target search efficiency. a**, Schematics for endogenously tagging CTCF/cohesin with AID in the HaloTag-YY1 cell line (YY1-HT, clone YN11) and western blots of CTCF, RAD21 and β-actin. **b**, Airyscan-resolved, live-cell confocal imaging for HaloTag-YY1 stained with 500 nM TMR Halo ligand in CTCF- or RAD21-depleted cells (n = 6 for each depletion). Scale bar, 1 μm. **c**, PALM imaging for YY1 (n = 13 for each depletion). Color maps color the signal ranging from 0 to 40. Scale bar, 1 μm. **d**, Stacked bar graph showing the fractions of bound, slow- and fast-diffusing YY1 in the untreated and IAA-treated cells, obtained by SASPT analysis (n = 8 cells examined over three independent experiments). The statistical test used was the two-sided Student's t-test. NS, not significant. Error bars indicate mean ± s.d. **e**, FRAP analysis of YY1 in the control, CTCF-depleted or RAD21-depleted cells. **f**, Heatmaps (left) and histogram profiles (right) of differential ChIP-seq signals for YY1 after CTCF, RAD21 or WAPL depletion. Error bars indicate mean ± s.d. **g**, Dynamic model of how cohesin or cohesin-mediated structures may accelerate TF target search.

Taken together, our results reveal a role for cohesin in accelerating the target search of TFs, resulting in increased YY1 chromatin binding as measured by SPT, FRAP and ChIP-seq. Cohesin or cohesin-mediated genome structure is likely to facilitate transcriptional establishment via more efficient target sampling of TFs (Fig. 7g). These findings also suggest that long-term cohesin depletion experiments must be interpreted with caution because cohesin depletion results in both direct and indirect effects, including diminished general TF binding to DNA.

## Discussion

Both the extent and mechanism by which CTCF- and cohesin-mediated loop extrusion regulates transcription have remained puzzling and hotly debated[12,13,32,39,43,63–67]. In the present study we applied high-resolution Micro-C to overcome this limitation. Surprisingly, we found that CTCF, cohesin, WAPL or YY1 is not required for the maintenance of most E–P/P–P loops or transcription at least within a 3-h depletion in mESCs. When affected, the altered E–P/P–P interactions only result in moderate expression changes of the underlying genes.

Our findings, together with other evidence[63,68], allow us to distinguish and/or eliminate several models of E–P interactions previously assigned to these ubiquitous structural proteins (Fig. 8).

First, CTCF and cohesin have been proposed to either directly bridge E–P interactions[69] or indirectly mediate E–P interactions by increasing contact frequency inside TADs (Fig. 8, Model 1)[70]. Our findings that acute CTCF, cohesin and WAPL depletion minimally affect gene expression (Fig. 3h–j) and E–P interactions (Fig. 4) disfavor this model for short-term maintenance of E–P interactions, although CTCF and cohesin may still help establish E–P interactions indirectly. We propose that loop extrusion may often be a separable mechanism from most E–P interactions and transcription, which is further supported by the following observations: (1) >20% of E–P/P–P loops can cross TAD boundaries and retain high contact probability and transcriptional activity (Fig. 2)[18,35]; (2) only a very small handful of genes showed altered expression levels after CTCF, cohesin or WAPL depletion (Fig. 3)[12–16]; (3) CTCF and cohesin loops are both rare (~5% of the time) and dynamic (median lifetime ~10–30 min)[34]; (4) most of the E–P/P–P

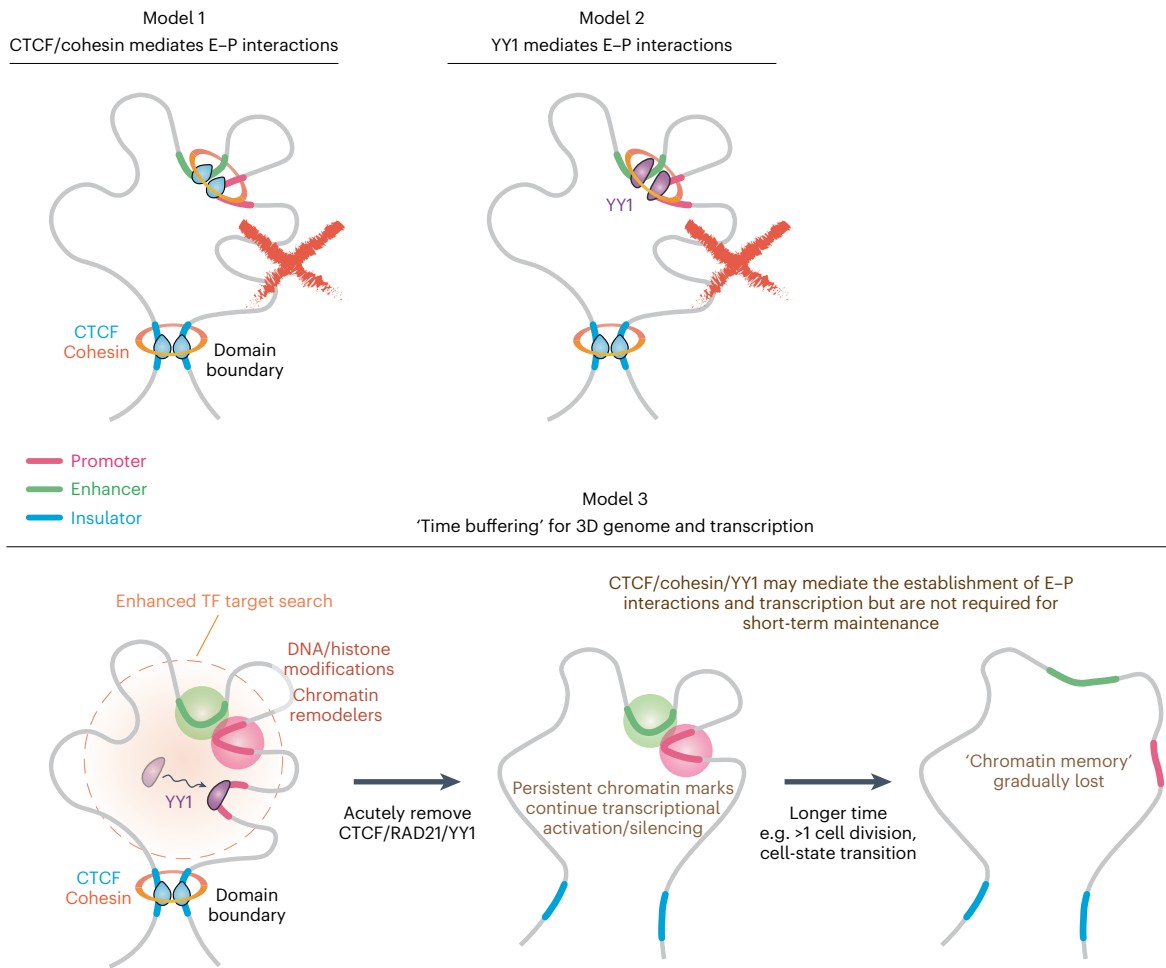

**Fig. 8 | Models of E–P interactions and transcription in the context of 3D genome organization.** Our findings exclude CTCF, cohesin or YY1 being required short term to maintain E–P interactions. Instead, we propose a time-buffering model to link 3D genome organization and gene expression. Once established, E–P interactions can temporarily sustain gene expression in the absence of architectural proteins, perhaps through an epigenetic molecular memory. We also propose that cohesin facilitates TF binding to chromatin.

loops persist after depletion of these structural proteins (Fig. 4)[39,63]; (5) CTCF/cohesin generally does not colocalize with transcription loci[67]; and (6) E–P loops and transcription can be established before CTCF/cohesin interactions on mitotic exit[71], in some cases even with no CTCF/cohesin expression[36,65,66]. Second, YY1 was proposed to be a master structural regulator of E–P interactions[25] (Fig. 8, Model 2). However, our Micro-C data are inconsistent with this model, because acute YY1 depletion has little effect on E–P/P–P interactions or gene expression. It is still possible that YY1 specifically connects development-related chromatin loops during neural lineage commitment[47], but is less important in the pluripotent state. In summary, we conclude that, in mESCs, CTCF, cohesin, WAPL or YY1 is not generally required for the short-term maintenance of most E–P interactions and the subsequent expression of most genes after acute depletion and loss of function.

The evidence that CTCF and cohesin can directly or indirectly regulate E–P interactions and affect gene expression in many cases is overwhelming[72–79]. To reconcile these studies with our observations, we propose a 'time-buffering' model (Fig. 8, Model 3). In this model, CTCF, cohesin and architectural factors contribute to the establishment of E–P interactions, but not to their maintenance. Instead, once established, a molecular memory (for example, histone modifications[80], chromatin remodeling[81–83], DNA modification[84–86], long noncoding RNAs[87,88]) may be sufficient to maintain E–P interactions and gene expression for several hours without the contribution of these architectural factors. We propose that this time-buffering model and its variants[89,90] reconcile our observations with the unambiguous genetic evidence that CTCF and cohesin regulate some E–P interactions. An alternative, more conservative, interpretation of our data and the evidence cited above is that CTCF and cohesin only regulate a very small, unique set of genes in specific biological processes and cell types and their effect on a handful of loci simply cannot be generalized as a universal rule.

In the present study, we also provide the first comprehensive study, to our knowledge, of YY1 dynamics and nuclear organization (Fig. 6). Surprisingly, we found that cohesin depletion, but not CTCF depletion, significantly reduces YY1 chromatin binding and slows down its target search time from 28 s to 61 s. A similar effect was also observed in SOX2 and KLF4 in the present study, as well as independently in glucocorticoid receptors by another group[91]. Furthermore, a study using high-throughput ChIP-seq analysis suggested that cohesin is critical to promote TF rebinding after mitosis[92]. We therefore propose that cohesin could facilitate TF binding to chromatin in general (Fig. 8, Model 3). After cohesin depletion, TFs take a longer time to find their targets, which may decrease transcription activation efficiency and eventually lead to changes in gene expression. It is interesting that the subunits of cohesin, as well as its loading and unloading complexes, are composed of multiple segments of intrinsically disordered regions, which may facilitate TF binding to chromatin via establishing weak

multivalent interactions[93,94]. Although more quantitative works will be necessary to unveil these mechanisms, in addition to its roles in loop extrusion, DNA repair, replication and chromosome segregation, cohesin might also facilitate TF binding to chromatin and could be critical for ensuring the precise timing of gene activation and silencing during embryonic development and cell-state transitions[36].

In summary, we have comprehensively investigated the role of CTCF, RAD21, WAPL and YY1 in finer-scale chromatin structure, nascent transcription, as well as YY1 dynamics and nuclear organization in mESCs. We propose a time-buffering model, where architectural proteins generally contribute to the establishment, but not the short-term maintenance, of E–P interactions and gene expression, and we also propose that cohesin plays an underappreciated role to facilitate TF binding to chromatin. The connection linking protein dynamics to chromatin structure opens a new avenue to rethink the mechanism of transcriptional regulation in the context of 3D genome organization.

## Online content

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

## Methods

### Nomenclature for chromatin 'loops' or 'dots'

The focal contact enrichment in Hi-C maps has historically been described as a 'loop' based on the assumption that motor proteins (that is, cohesin complex or RNA polymerase II) or TFs bridge long-range genomic loci together, forming a 'loop-shaped' structure in vitro and in vivo. Unlike cohesin, which is likely to form loops through loop extrusion, E–P or P–P interactions may occur by a variety of mechanisms without looping. Their interactions are typically detected as 'dots' in contact matrices. We agree that the term 'dot' is ideal for describing these enhanced focal contacts without making any assumptions about their folding mechanisms or actual 3D structures. However, we chose to use 'loop' over 'dots' (that is, cohesin loops or E–P loops) to make the manuscript more accessible to the general audience and to match the terms that are commonly used in the field.

### Cell culture, stable cell-line construction and dye labeling

JM8.N4 mESCs[95] (Research Resource Identifier: RRID:CVCL_J962; obtained from the KOMP Repository at University of California (UC), Davis) were used for all experiments. Cells were cultured on plates precoated with 0.1% gelatin (Sigma-Aldrich, catalog no. G9291) in knock-out Dulbecco's modified Eagle's medium (DMEM; Thermo Fisher Scientific, catalog no. 10829018) supplemented with 15% fetal bovine serum (HyClone FBS SH30910.03 lot no. AXJ47554), 0.1 mM minimal essential medium nonessential amino acids (Thermo Fisher Scientific, catalog no. 11140050), 2 mM GlutaMAX (Thermo Fisher Scientific, catalog no. 35050061), 0.1 mM 2-mercaptoethanol (Sigma-Aldrich, catalog no. M3148), 1% penicillin–streptomycin (Thermo Fisher Scientific, catalog no. 15140122) and 1,000 units of leukemia inhibitory factor (Millipore). Medium was replaced daily and cells were passaged every 2 d by trypsinization. Cells were grown at 37 °C and 5.5% $CO_2$ in a Sanyo copper alloy IncuSafe humidified incubator (MCO-18AIC(UV)). For imaging, the medium was identical except that knock-out DMEM lacking phenol red (Thermo Fisher Scientific, catalog no. 31053028) was used to minimize background fluorescence.

Cell lines stably expressing 3×FLAG-HaloTag-YY1 and YY1-HaloTag-3×FLAG were generated using PiggyBac transposition and drug selection. Full details are given in Supplementary Methods.

For PALM experiments, cells were grown overnight on Matrigel-coated (Corning, catalog no. 354277; purchased from Thermo Fisher Scientific, catalog no. 08-774-552), 25-mm circular no. 1.5H cover glasses (High-Precision, catalog no. 0117650). Before all experiments, the cover glasses were plasma cleaned and then stored in isopropanol until use. Cells were labeled with 500 nM PA-JFX549 HaloTag ligand for 30 min, washed twice with fresh medium for 5 min and then washed once with phosphate-buffered saline (PBS), pH 7.4. Labeled cells were fixed with 4% paraformaldehyde and 2% glutaraldehyde in PBS for 20 min at 37 °C, washed once with PBS and imaged in PBS with 0.01% (w:v) $NaN_3$.

For FRAP experiments, cells were grown overnight on Matrigel-coated glass-bottomed 35-mm dishes (MatTek P35G-1.5-14C). Cells were labeled with 500 nM HaloTag tetramethylrhodamine (TMR) ligand (Promega, catalog no. G8251) for 30 min and washed twice with PBS.

### Generation of CRISPR–Cas9-mediated knock-in cell lines

Endogenously tagged mESC lines were generated by CRISPR–Cas9-mediated genome editing as previously described[96] with modifications. Full details are given in Supplementary Methods.

### Western blotting

See Supplementary Methods.

### ChIP and ChIP-seq

ChIP was performed as described with a few modifications[97] (see Supplementary Methods for details).

ChIP-seq libraries were prepared using the NEBNext Ultra II DNA Library Prep Kit for Illumina (New England Biolabs (NEB), catalog no. E7645) according to the manufacturer's instructions with a few modifications (see Supplementary Methods for details). Library concentration, quality and fragment size were assessed by Qubit fluorometric quantification (Qubit dsDNA HS Assay Kit, Invitrogen, catalog no. Q32851), quantitative PCR and Fragment analyzer. Twelve multiplexed libraries were pooled and sequenced in one lane on the Illumina HiSeq4000 sequencing platform (50-bp, single-end reads) at the Vincent J. Coates Genomics Sequencing Laboratory at UC Berkeley, supported by National Institutes of Health (NIH, grant no. S10 OD018174) instrumentation grant.

See Supplementary Methods for the details on the ChIP-seq analysis.

### Biochemical fractionation

Wild-type JM8.N4 mESCs were seeded on to 15-cm plates, washed with ice-cold PBS, scraped in PBS and pelleted at 135$g$ for 10 min at 4 °C. Pellets were resuspended in 350 µl of cell lysis buffer A (10 mM Hepes, pH 7.9, 10 mM KCl, 3 mM $MgCl_2$, 340 mM sucrose, 10% glycerol, v:v, 1 mM dithiothreitol (DTT) and freshly added 0.1% Triton X-100, v:v, and protease inhibitors) and rocked for 8 min at 4 °C. Nuclei were pelleted at 3,000$g$ for 3 min at 4 °C and the supernatant containing the cytoplasmic fraction was saved. Nuclei were resuspended in 350 µl of buffer B with 75 mM NaCl (9 mM EDTA, 0.2 mM (ethylenebis(oxonitrilo)) tetra-acetate, 1 mM DTT, freshly added 0.1% Triton X-100, v:v, and protease inhibitors) and rocked at 4 °C for 15 min. Nuclei were pelleted again as above (supernatant saved as the 75 mM wash fraction) and washed with 350 µl of buffer B with increasing NaCl concentrations (150 mM, 300 mM, 500 mM and 1 M; see Extended Data Fig. 2f for a step-by-step procedure). After collecting the 1 M wash, the pellet was resuspended to 350 µl of 1 M buffer B and sonicated (Covaris S220 sonicator, 20% Duty factor, 200 cycles per burst, 100 peak incident power, 8 cycles of 20 s on and 40 s off). The sonicated lysate was spun down and the insoluble pellet boiled in sodium dodecylsulfate (SDS)-loading buffer. Then, 10 µl of each fraction was added to 2 µl of 4× SDS-loading buffer and subjected to western blotting as detailed above. Band intensities were quantified with the ImageJ 'Analyze Gels' function[98].

### Micro-C assay for mammalian cells

We briefly summarize the Micro-C experiment in Supplementary Methods. The detailed protocol and technical discussion are available in our previous study[9].

Micro-C-seq libraries were generated using the NEBNext Ultra II DNA Library Prep Kit for Illumina (NEB, catalog no. E7645) with some minor modifications (detailed in Supplementary Methods). We used Illumina 100-bp paired-end sequencing (PE100) to obtain ~400 M reads for each replicate in the present study.

### Micro-C data processing and analyses

Valid Micro-C contact read pairs were obtained from the HiC-Pro analysis pipeline (v.2.11.3)[99] and the detailed description and code can be found at https://github.com/nservant/HiC-Pro (see Supplementary Methods for a brief description).

Valid Micro-C contacts were assigned to the corresponding 'pseudo' nucleosome bin. The bin file was pregenerated from the mouse mm10 genome by a 100-bp window that virtually resembles the nucleosome resolution. The binned matrix can be stored in HDF5 format as a COOL file using the COOLER package (v.0.8.10) (https://github.com/mirnylab/cooler)[100] or in HIC file format using the JUICER package (v.1.22.01) (https://github.com/aidenlab/juicer)[101]. Contact matrices were then normalized by using iterative correction in COOL files[102] or Knight–Ruiz in HIC files[103]. Regions with low mappability and high noise were blocked before matrix normalization. We expect that matrix-balancing normalization corrects systematic biases such as

nucleosome occupancy, sequence uniqueness, GC content or crosslinking effects[102]. We notice that both normalization methods produce qualitatively equal contact maps. To visualize the contact matrices, we generated a compilation of COOL files with multiple resolutions (100-bp to 12,800-bp bins) that can be browsed on the HiGlass 3D genome server (http://higlass.io)[104]. In the present study, all snapshots of Micro-C or Hi-C contact maps and the one-dimensional (1D) browser tracks (for example, ChIP-seq) were generated by the HiGlass browser (v.1.11.7) unless otherwise stated.

We evaluated the reproducibility and data quality for the Micro-C replicates using two published methods independently (https://github.com/kundajelab/3DChromatin_ReplicateQC)[105] (see Supplementary Methods for details).

To analyze the genome-wide, contact-decaying *P*-value curve, we used intrachromosomal contact pairs to calculate the contact probability in bins with exponentially increasing widths from 100 bp to 100 Mb. Contacts shorter than 100 bp were removed from the analysis to minimize noise introduced by self-ligation or undigested DNA products. The orientations of ligated DNA are parsed into 'IN-IN (+/−)', 'IN-OUT (+/+)', 'OUT-IN (−/−)' and 'OUT-OUT (−/+)' according to the readouts of Illumina sequencing[22,23]. 'UNI' pairs combine 'IN-OUT' and 'OUT-IN' because both orientations are theoretically interchangeable. In the present study, we plotted the contact decaying curves with the 'UNI' pairs and then normalized to the total number of valid contact pairs. Slopes of contact decay curves were obtained by measuring slopes in a fixed-width window searching across the entire range of decaying curves. We then plotted the derivative slope in each window against the corresponding genomic distance.

To identify chromosome compartments, we first transformed the observed:expected Micro-C matrices at the 200-kb resolution to Pearson's correlation matrices and then obtained the eigenvector of the first principal component of Pearson's matrix by principal component analysis. The sign of the eigenvector was corrected using active histone marks (H3K27ac and H3K4me3), because positive values are the A compartment (gene-rich or active chromatin) and negative values are the B compartment (gene-poor or inactive chromatin). The detailed description can be found in Lieberman-Aiden et al.[3]. The genome-wide compartment strength analysis shown as a saddle plot represents the rearrangement and aggregation of the genome-wide, distance-normalized contact matrix with the order of increasing eigenvector values. The chromosome arm is first divided into quantiles based on the compartment score. All combinations of quantile bins are averaged and rearranged in the saddle plot. The Cooltools package (v.0.3.2; https://github.com/mirnylab/cooltools) has implemented the 'call-compartments' and 'compute-saddle' functions with the COOL files.

To identify chromatin domains (TADs) along the diagonal, we used insulation score analysis from the Cooltools package (v.0.3.2; https://github.com/mirnylab/cooltools) or arrowhead transformation analysis from the JUICER package (v.1.22.01; https://github.com/aidenlab/juicer)[101] (see Supplementary Methods for more details).

Details of loops/dots identification and related analyses are in Supplementary Methods.

### Definition of chromatin states and structure observed by Micro-C

We first used the published ChromHMM (http://compbio.mit.edu/ChromHMM)[106,107] to define the chromatin states in mESCs, which subclassifies chromatin into 12 states including: (1) CTCF/insulator, (2) active promoter (designated as 'P'), (3) strong enhancer, (4) medium enhancer, (5) weak enhancer, (6) mix of promoter and enhancer, (7) bivalent promoter, (8) gene body, (9) polycomb repressor, (10) intergenic regions, (11) heterochromatin and (12) repeats. To simplify the analysis, we further combined the groups of strong, medium and weak enhancers and mix of promoter and enhancer into 'enhancer' (designated as 'E'). In the present study, we use the terms that are widely

accepted in the field to describe the chromatin structures in Micro-C contact maps as well as avoid any ambiguous description that implicates their biological functions if they have not been well characterized, including: (1) TAD: squares along matrix diagonal enriched with self-interactions, which are defined as genomic intervals demarcated by the boundaries characterized by the insulation score analysis or the arrowhead transformation analysis; (2) cohesin loops: focal enrichment of contacts in contact maps with the coenrichment of CTCF/cohesin ChIP-seq peaks at loop anchors, which is thought to be formed by active loop extrusion halted by CTCF; and (3) E−P/P−P dots: focal enrichment of contacts in contact maps with the coenrichment of chromatin states for 'active promoter (P)' or 'enhancer (E)' at loop anchors. Although not all cohesin loops and E−P/P−P loops are formed through 'looping', and some studies suggest using 'dots' instead of 'loops', to simplify and be consistent with most of the findings, we chose to use 'loops' for cohesin-mediated focal contacts and 'dots' for other categories of enhanced focal contacts in this manuscript.

### RNA-seq experiments and analysis

Total RNA was extracted from ~$1 \times 10^7$ mESCs (~70% confluent P10 dish) using the standard TRIzol RNA extraction protocol. The abundant rRNAs were depleted from the sample using the NEBNext rRNA Depletion Kit (NEB, catalog no. E6310). The rRNA-depleted RNAs were then subjected to RNA-seq library construction using the NEBNext Ultra II Directional RNA Library Prep Kit for Illumina (NEB, catalog no. E7765). The final RNA-seq libraries were amplified with seven to eight PCR cycles.

For RNA-seq analysis, we used Kallisto (v.0.46.2)[108] to quantify the number of transcripts and performed DEseq2 (v.1.30.1)[109] analysis for DEG identification according to the recommended settings in the walkthrough (http://bioconductor.org/packages/devel/bioc/vignettes/DESeq2/inst/doc/DESeq2.html) with $P_{adj} < 0.01$ and fold-change >2. Full lists of DEGs are available in Supplementary Table 11.

### Nascent RNA-seq experiment and analysis

We used the nascent RNA-seq (mNET-seq) protocol described in Nojima et al.[110] with minor changes, detailed in Supplementary Methods.

RNA libraries were prepared according to the protocol of the NEBNext Small RNA Library Prep Kit (NEB, catalog no. E7330). The mNET-seq library was obtained by PCR for 12–14 cycles.

For mNET-seq analysis, we wrote a customized pipeline to process raw data as follows: (1) adapter trimming: we used TrimGalore (v.0.6.7) (https://github.com/FelixKrueger/TrimGalore) to remove sequencing adapters 'AGATCGGAAGAGCACACGTCTGAACTCCAGTCAC' and 'GATCGTCGGACTGTAGAACTCTGAAC' at each side of the reads; (2) mapping: trimmed reads were mapped to the mouse mm10 reference genome with STAR RNA-seq aligner (v.2.7.10a)[111]; (3) identifying the last nucleotide incorporated by Pol II: we used the Python script mNET_snr (https://github.com/tomasgomes/mNET_snr) to locate the 3′-nucleotide of the second read and the strand sign of the first read. The bigWig files were generated using Deeptools (v.3.5.0) as described in Supplementary Methods. To identify DEGs in mNET-seq, we used either the Nascent RNA Sequencing Analysis (v.2)[112] package or FeatureCounts (v.1.22.2)[113] and DEseq2 (v.1.30.1)[109] to statistically quantify differential changes of the mNET-seq signal at the gene body between UT- and IAA-treated cells (with $P_{adj} < 0.01$ and fold-change >2). Full lists of DEGs are available in Supplementary Table 12.

### Single-particle imaging experiments

All single-molecule imaging experiments were performed with a similar setting as described in our previous studies[52,53] and detailed in Supplementary Methods.

For PALM experiments, continuous illumination was used for both the main excitation laser (633 nm for PA-JF646 or 561 nm for PA-JF549) and the photoactivation laser (405 nm). The intensity of the 405-nm

laser was gradually increased over the course of the illumination sequence to image all molecules and avoid too many molecules being activated at any given frame. The camera was set for 25-ms exposure time, frame transfer mode and vertical shift speed at 0.9 µs. In total, 40,000–60,000 frames were recorded for each cell (~20–25 min), which was sufficient to image and bleach all labeled molecules.

## The spaSPT analysis
For analysis of spaSPT experiments, we used the QUOT package (v.1; https://github.com/alecheckert/quot) to generate trajectories from raw spaSPT videos with the steps of spot detection, subpixel localization and tracking. All localization and tracking for this manuscript were performed with the following settings: (1) detection: generalized log(likelihood ratio test) with a 2D Gaussian kernel ('llr' with $k = 1.0$, pixel window size ($w$) = 15 and a log(ratio threshold ($t$)) = 26.0). (2) Subpixel localization: Levenberg–Marquardt fitting of a 2D integrated Gaussian point spread function model ('ls_int_gaussian' with $w = 9$, sigma = 1.0, ridge = 0.001, maximal iterations = 20 per point spread function and damping term = 0.3). (3) Tracking: we chose to use a conservative tracking algorithm with a 1.3-µm search radius ('conservative' with maximal blinks = 0). This setting makes the algorithm search for spot reconnections unambiguously, meaning that no other reconnections are possible within the specified search radius. Jumps were discarded if other reconnection possibilities given the search radius existed.

We next used the SASPT package (v.1; https://github.com/alecheckert/saspt)[54] to estimate the likelihood of diffusion coefficients for each trajectory. The detailed discussion is available in Heckert et al.[114] and described in Supplementary Methods.

Alternatively, we analyzed the spaSPT data with the kinetic modeling framework implemented in the Spot-On package (v.1.04)[53], briefly described in Supplementary Methods.

## Slow-SPT analysis
For analysis of slow-SPT experiments, we used the following tracking settings for this manuscript: (1) detection: 'llr' with $k = 1.0$, $w = 15$, $t = 18$; (2) subpixel localization: 'ls_int_gaussian' with $w = 9$, sigma = 1.0, ridge = 0.001, maximal iteration = 20 and damping = 0.3; (3) tracking: 'euclidean' with search radius = 0.5, maximal blinks = 1 and maximal diffusion constant ($\mu m^2 s^{-1}$) = 0.08.

Details on how we extracted residence times from slow-SPT are in Supplementary Methods.

## FRAP imaging analysis
FRAP was performed on an inverted Zeiss LSM 900 Axio Observer confocal microscope equipped with Airyscan 2 detector, a motorized stage, a full incubation chamber maintaining 37 °C/5% $CO_2$, a heated stage and an X-Cite 120 illumination source, as well various laser lines. Images were acquired on a ×40 Plan NeoFluar, numerical aperture 1.3, oil-immersion objective at a zoom corresponding to a 76 nm × 76 nm pixel size. The microscope was controlled using the Zeiss Zen imaging software.

In this manuscript, we recorded 60 s of videos for YY1-HaloTag at 1 frame per 250 ms, corresponding to a total of 240 frames. The first 20 frames were acquired before the bleach pulse, allowing us to accurately measure baseline fluorescence. A circular bleach spot ($r = 6$ pixels) was chosen in a region of homogeneous fluorescence at a position at least 1 µm from nuclear or nucleolar boundaries. Alternatively, we bleached a square at one corner of the nucleus, which reduces noise while introducing some uncertainty for our downstream fitting analysis. The spot was bleached using maximal laser intensity and pixel dwell time corresponding to a total bleach time of ~1 s. We note that, because the bleach duration was relatively long compared with the timescale of molecular diffusion, it is not possible to accurately estimate the bound and free fractions from our FRAP curves.

Details on the analysis of FRAP videos are in Supplementary Methods.

## Inferring parameters related to YY1's target search mechanism
We used the parameters inferred from our spaSPT and the residence time measurements from our FRAP or slow-SPT analysis. The detailed discussion is available in both Hansen et al.[52] and Supplementary Methods.

## PALM analysis
For analysis of PALM experiments, we used the publicly available ThunderSTORM package (v.1.3; https://github.com/zitmen/thunderstorm)[115] with the following setting for this manuscript: (1) image filtering: 'Wavelet filter (B-Spline)' with B-Spline order = 3 and B-Spline scale = 2.0; (2) approximate localization: 'Local maximum' with peak intensity threshold = 1.5 × std(Wave.F1) and 8-neighbourhood connectivity; (3) subpixel localization: 'Integrated Gaussian' with fitting radius = 3 pixels, fitting method = maximum likelihood, initial sigma = 1.6, multi-emitter analysis disabled; and (4) image reconstruction: 'Averaged shifted histogram'. After tracking, we further filtered ambiguous emitters with the following setting: (1) filtering: frame > 100 & intensity > 100 & sigma < 220 & uncertainty_xy < 50; (2) merge: Max distance = 10 & Max frame off = 1 & Max frames = 0; and (3) remove duplicates enabled. This setting combines the blinking molecules into one and removes the multiple localizations in a frame.

## Antibodies
See Supplementary Table 1 for a complete list of the antibodies used in the present study.

## Statistics and reproducibility
No statistical method was used to predetermine sample size. No data were excluded from the analyses. The experiments were not randomized. The Investigators were not blinded to allocation during experiments and outcome assessment. Western blotting, biochemical fractionation and flow cytometry experiments were repeated and confirmed at least twice.

## Reporting summary
Further information on research design is available in the Nature Portfolio Reporting Summary linked to this article.

## Data availability
The Micro-C, ChIP-seq, nascent RNA-seq and total RNA-seq data generated in this publication are available in National Center for Biotechnology Information's Gene Expression Omnibus (GEO) through accession no. GSE178982. We also reanalyzed data that we previously generated in wild-type mESCs (GEO accession no. GSE130275)[9]. The spaSPT raw data are accessible through https://doi.org/10.5281/zenodo.5035837. The reference genome mm10 and sacCer3 are available through UC Santa Cruz genome browser (https://hgdownload.soe.ucsc.edu/downloads.html). Source data are provided with this paper.

## Code availability
The availability of the codes used in this manuscript is specified in Methods and Supplementary Methods.

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

## Acknowledgements

This work was supported by the California Institute of Regenerative Medicine (grant no. LA1-08013 to X.D.), the Koret UC Berkeley–Tel Aviv University Initiative grant (to T.S.H. and X.D.) and the Howard Hughes Medical Institute (grant no. 003061 to R.T.). T.S.H. was a postdoctoral fellow of the Koret UC Berkeley–Tel Aviv University Initiative. E.S. was an undergraduate fellow of SURF Rose Hills Independent at UC Berkeley. A.S.H. acknowledges support from the NIH under grant nos. R00GM130896, DP2GM140938, R33CA257878 and NSF 2036037. We thank Gina M. Dailey for assisting with cloning and all members of the Tjian and Darzacq laboratory for comments on the manuscript.

## Author contributions

T.S.H., C.C. and E.S. conceived and designed the project. C.C. performed all biochemical and ChIP-seq experiments and analysis with E.S.'s assistance. E.S. generated all plasmids and cell lines with the assistance of T.S.H. C.C. and A.S.H. generated the parental degradation cell lines. T.S.H. performed Micro-C assays and analysis. E.S. and T.S.H. performed all imaging experiments and analyses. T.S.H. drafted the manuscript. C.C., E.S., A.S.H. and R.T. edited the manuscript. R.T. and X.D. supervised the project.

## Competing interests

The authors declare no competing interests.

## Additional information

**Extended data** is available for this paper at https://doi.org/10.1038/s41588-022-01223-8.

**Correspondence and requests for materials** should be addressed to Xavier Darzacq or Robert Tjian.

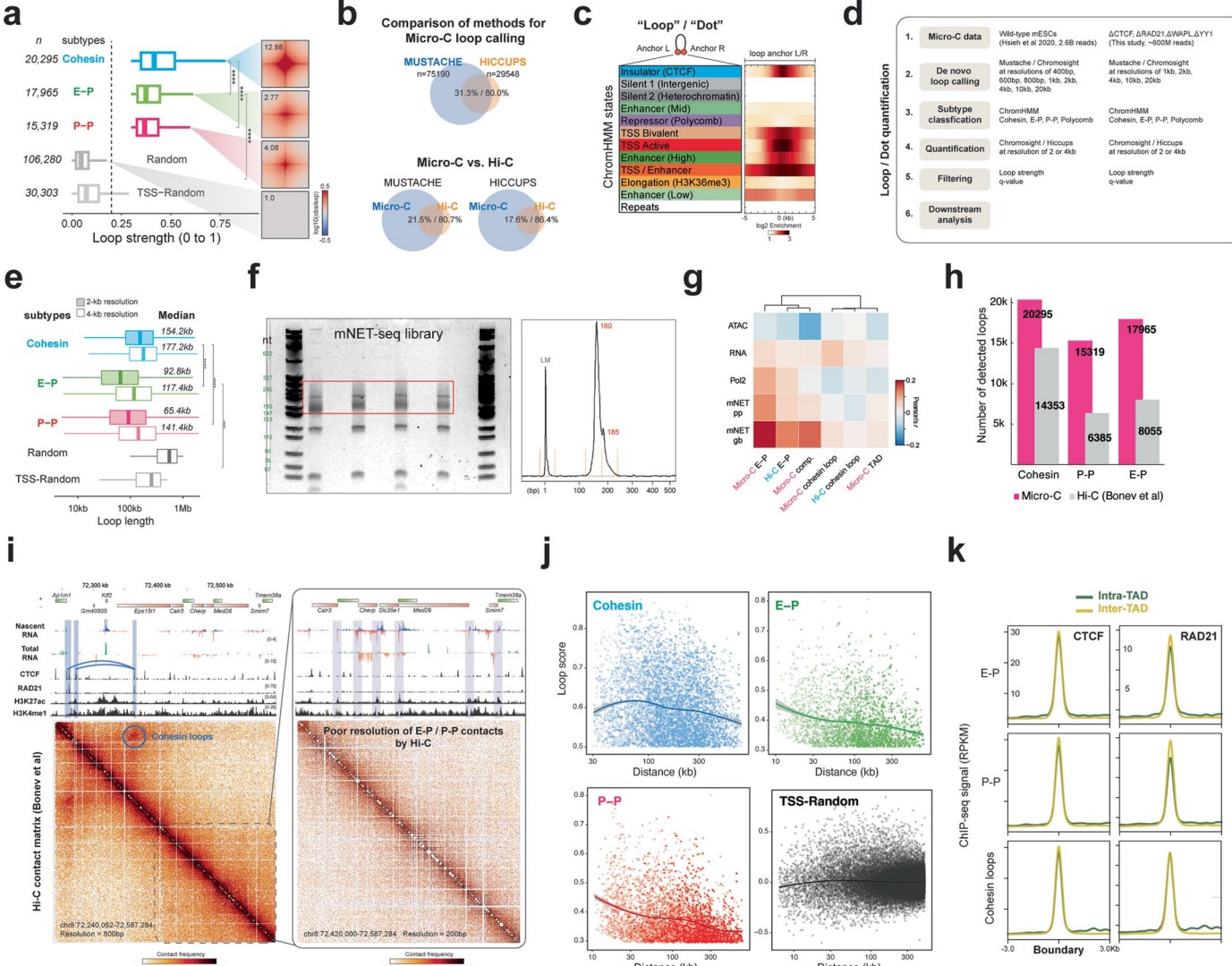

**Extended Data Fig. 1 | Genome-wide identification of chromatin loops.**
**a**. Loop strengths. Paired loci quantified using Chromosight[27] (results comparable to Mustache[26] in Fig. 1b). Loop numbers shown on the left. Box plot: quartiles for the loop strength score distribution as in Fig. 1b. Right: genome-wide averaged contact signals. Contact map normalized by matrix balancing and distance, shown as a diverging colormap with the gradient of normalized contact enrichment in log10 (red: positive enrichment; blue: negative signal). Ratio of contact enrichment for the center pixels annotated within each plot. Asterisks: $P < 10^{-16}$, two-sided Wilcoxon test. n = 37 biological replicates. **b**. Comparison of Mustache and Hiccups loop calling algorithms on Micro-C data (top) and Micro-C vs. Hi-C loops called by different algorithms (bottom). **c**. Enrichment of mESC ChromHMM[107] states at loop anchors. Heatmap: log2 enrichment of each state ± 5-kb around loop anchors. Loops = 75,190; loop anchors = 118,733 after removing duplicates. **d**. Loop analysis pipeline. **e**. Loop length distributions. Colored box: 2-kb resolution Micro-C data; white box: 4-kb resolution data. Box plot: quartiles for the loop length distribution as in Fig. 1b. Median size of loops annotated

on the right. Median lengths are larger than our previous analysis with the insulation score[4] due to the high computational expense to quantify the short-range loops with Micro-C data finer than 1-kb resolution. Asterisks: $P < 10^{-16}$, two-sided Wilcoxon test. n = 37 biological replicates. **f**. Left: gel image of the mNET-seq library size (6% PAGE). Right: resolved bands on a Fragment Analyzer electropherogram. **g**. Heatmap of Pearson's correlation between sequencing data and chromatin structures by Micro-C or Hi-C. Compartment, TAD and loop scores obtained from Cooltools, Arrowhead and Chromosight, respectively. **h**. Micro-C or Hi-C[28] loop numbers. Contacts surrounding the intersections of targets quantified with data at 4-kb resolution using Chromosight. **i**. Snapshots of Hi-C data in the same region as Fig. 2a. **j**. Rank-ordered distribution of loop length against loop strength. Distributions fitted and smoothed by LOESS regression. Error bands: fitted curve ±SEM with 95% confidence interval. **k**. CTCF and RAD21 ChIP-seq signal at TAD boundaries grouped by intra-TAD/inter-TAD cohesin, E-P, or P-P loops.

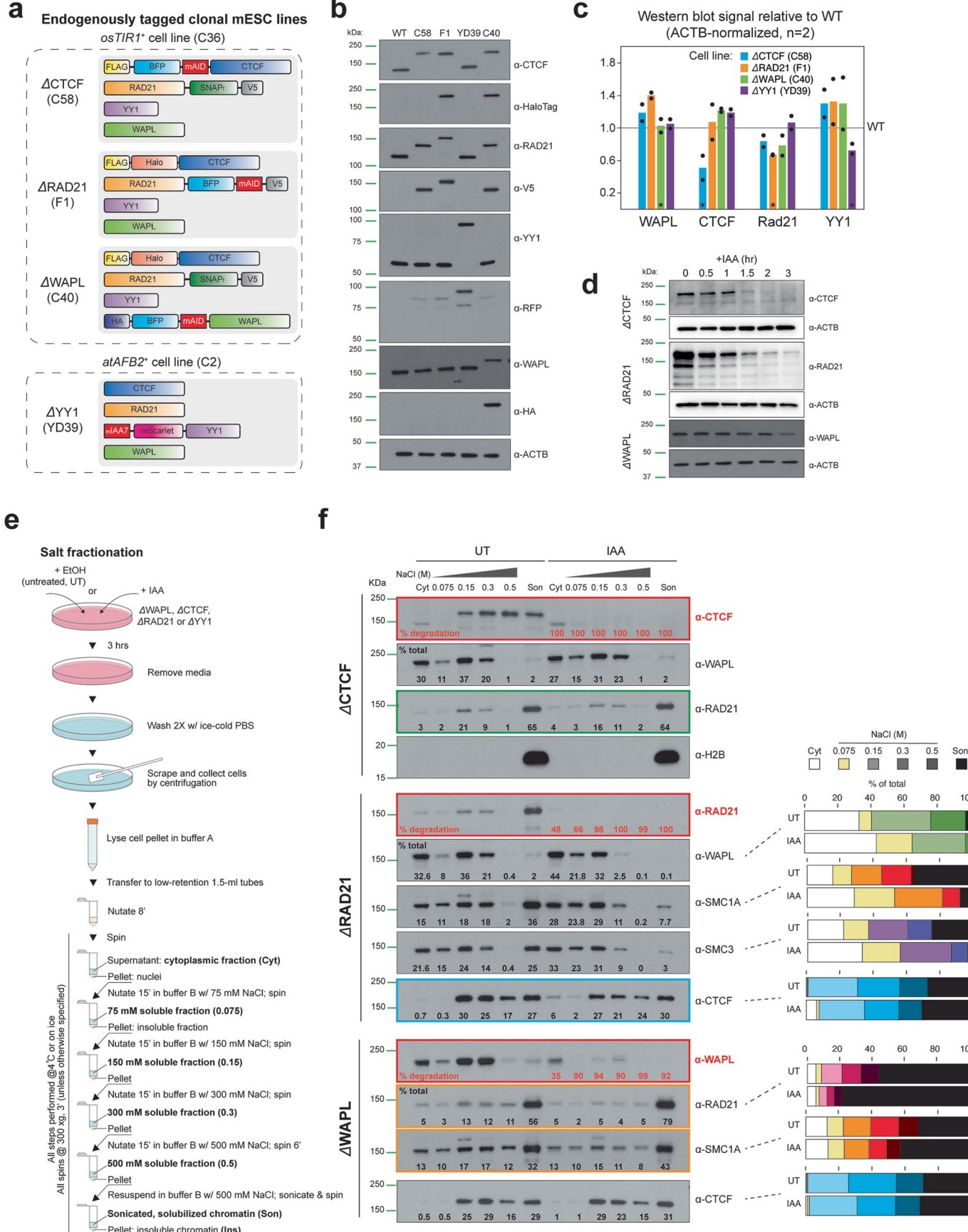

**Extended Data Fig. 2 | See next page for caption.**

**Extended Data Fig. 2 | Cell lines generation, validation, and biochemical fractionation assay. a**. Schematics for endogenously tagging CTCF, RAD21, WAPL with the mAID degron and for endogenously tagging YY1 with miniIAA7. **b**. Immunoblots of CTCF, RAD21, WAPL, YY1, and their tags (HaloTag for CTCF, V5 for RAD21, RFP (mScarletI) for YY1, and HA for WAPL) for the protein expression levels and sizes in wild type mESCs and degron clones C58 (ΔCTCF), F1 (ΔRAD21), YD39 (ΔYY1), and C40 (ΔWAPL). **c**. Quantification of the levels of WAPL, CTCF, Rad21 and YY1 proteins in the degron clones C58 (ΔCTCF), F1 (ΔRAD21), C40 (ΔWAPL) and YD39 (ΔYY1) relative to wild type mESCs by immunoblotting (n = 2 independent immunoblots ran on the same cell lysates). Black asterisks point to the basal degradation level of each degron-tagged factor in the corresponding cell line. **d**. Immunoblots of CTCF, RAD21 and WAPL proteins across a degradation time course from 0 (untreated) to 3 hr (IAA treatment) in

ΔCTCF, ΔRAD21, and ΔWAPL degron clones. **e**. Schematic for biochemical salt fractionation experiment in mock-treated (UT) or IAA-treated degron clones. **f**. Immunoblots of cytoplasmic (Cyt) and nuclear proteins dissociating from chromatin at increasing salt concentrations (75, 150, 300 and 500 mM NaCl) as schematized in g, probed with the indicated antibodies (α). Son: sonicated, solubilized chromatin; % of total: signal intensity of each fraction divided by the total signal intensity across all fractions; % of degradation: 1 – (signal intensity of each fraction in the IAA treated condition divided by the untreated condition), after normalization for total protein amounts (normalizer for ΔWAPL degron: total CTCF; normalizer for ΔRAD21 degron: total YY1). A blot with anti-histone 2B antibody (almost exclusively found in the solubilized chromatin) controls for chromatin integrity during the fractionation steps.

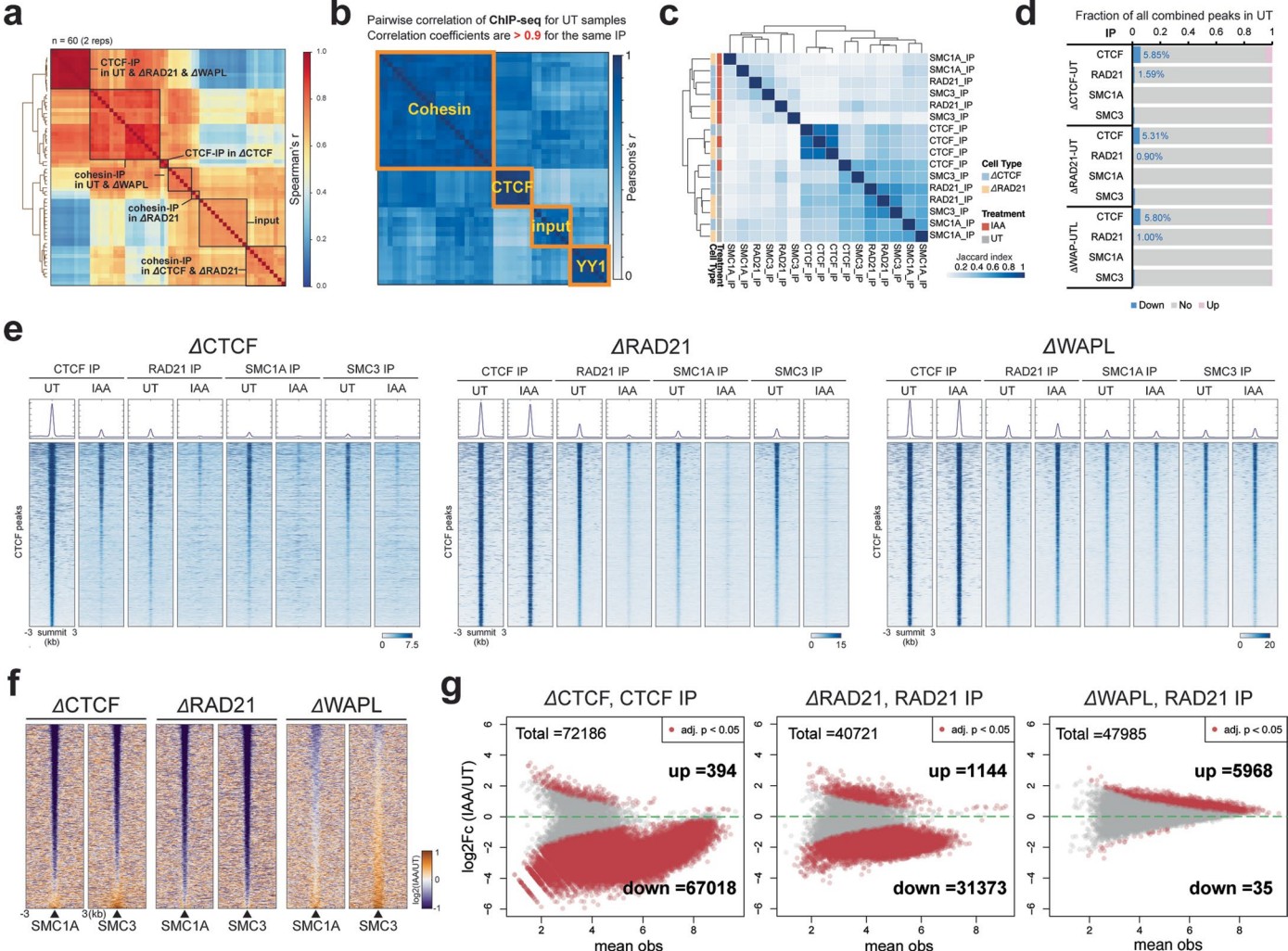

**Extended Data Fig. 3 | ChIP-Seq analysis. a**. Heatmap of 'all-by-all' Spearman's correlation for all ChIP-seq replicates samples (*n* = 96). **b**. Pairwise correlation of ChIP-seq data for all UT samples. **c**. Heatmap of Jaccard's index for the ratio of co-enriched peaks between the ChIP-seq replicates. **d**. Summary of differential ChIP-seq peak analysis for all UT degron cell clones. The chart shows the fraction of down-regulated, up-regulated, or unchanged peaks in the UT condition. The total number of peaks for each protein was summed from all peaks in UT cells. **e**. Heatmaps of CTCF and cohesin (RAD21, SMC1A, and SMC3 subunits) ChIP-seq signal around WT-CTCF peaks called by MACS2 in the CTCF-, RAD21-, or WAPL-degron cells. The peaks called by MACS2 are plotted at the center across a ±3-kb region. The colormap shows the maximum signal (log2) in blue and the minimum signal in white. **f**. Heatmaps of differential ChIP-seq signals for SMC1A and SMC3 in cells depleted of CTCF, RAD21, or WAPL. The peaks called by MACS2 are plotted at the center across a ±3-kb region. The colormap shows an increased signal (log2) in orange and a decreased signal in purple after IAA treatment. **g**. MA plots show the differential ChIP-seq peaks between the UT and IAA-treated cells. The significantly changed peaks ($P_{adj}$ < 0.05) are colored in red. X-axis: mean observations of UT and IAA cells. Y-axis: log2 fold-change comparing the UT and IAA-treated cells. The statistical test for all ChIP-seq in this study are obtained from the statistical model derived from MAnorm2 unless otherwise indicated.

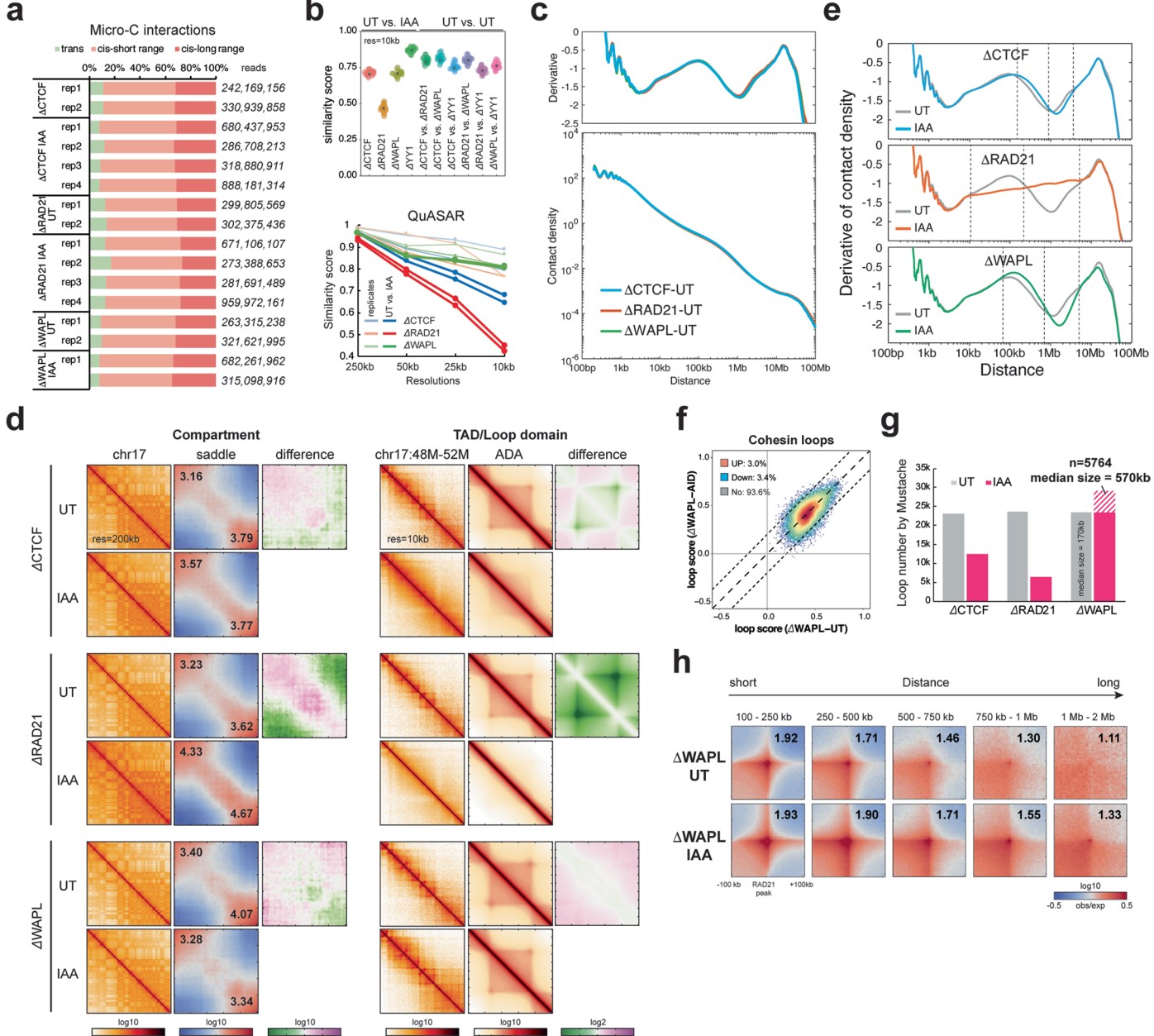

**Extended Data Fig. 4 | Micro-C analysis. a**. Summary of Micro-C experiments in the degron cell lines. Total unique reads annotated for each replicate on the right, consisting of trans-interactions (inter-chromosome), short-range cis-interactions (<20 kb), and long-range cis-interactions (>20 kb). **b**. Micro-C reproducibility tests. Top: pairwise similarity scores measured by GenomeDisco between UT vs. IAA and UT vs. UT samples using 10-kb resolution of Micro-C matrices. Bottom: similarity scores measured by QuASAR between replicates (light lines) or comparing the UT and IAA-treated samples (dark lines) using Micro-C matrices at 250-kb, 50-kb, 25-kb, and 10-kb resolutions. **c**. Genome-wide contact decaying P(s) analysis (bottom) and slope distributions of the P(s) curves (top) for UT cells. **d**. Micro-C contact maps at specific regions or at genome-wide scale across multiple resolutions in the UT and IAA-treated cells. Left to right: examples of Pearson's correlation matrices showing plaid-like chromosome compartments; saddle plots showing overall compartment strength (A-A: bottom-right; B-B: top left); differential saddle plots showing changes in compartment strength; contact matrices showing TADs along the diagonal; ADA showing all TADs; differential ADA showing TAD strength changes. **e**. Slope distribution of P(s) curves for UT and IAA-treated cells. Dashed lines highlight the range of genome distances affected by CTCF, RAD21, or WAPL depletion. CTCF depletion had minimal impact on overall interactions across the genome. RAD21 depletion reduced contact frequencies in the range of 10–200 kb but increased interactions at 300 kb – 5 Mb. WAPL depletion showed the opposite trend, with increased contacts at 70–700 kb but reduced contacts at 1–5 Mb. **f**. Scatter plot of cohesin loops scores in UT and IAA-treated cells. The overlaid heatmap indicates dot density (red: highest, blue: lowest). Dashed lines along the diagonal delimit unchanged loops. **g**. Loop numbers called by Mustache for UT and IAA-treated cells. The additional loops (n = 5764) identified after WAPL depletion show longer lengths, with a 570-kb median. **h**. APA for loops across multiple ranges of genomic distance in UT and IAA-treated cells.

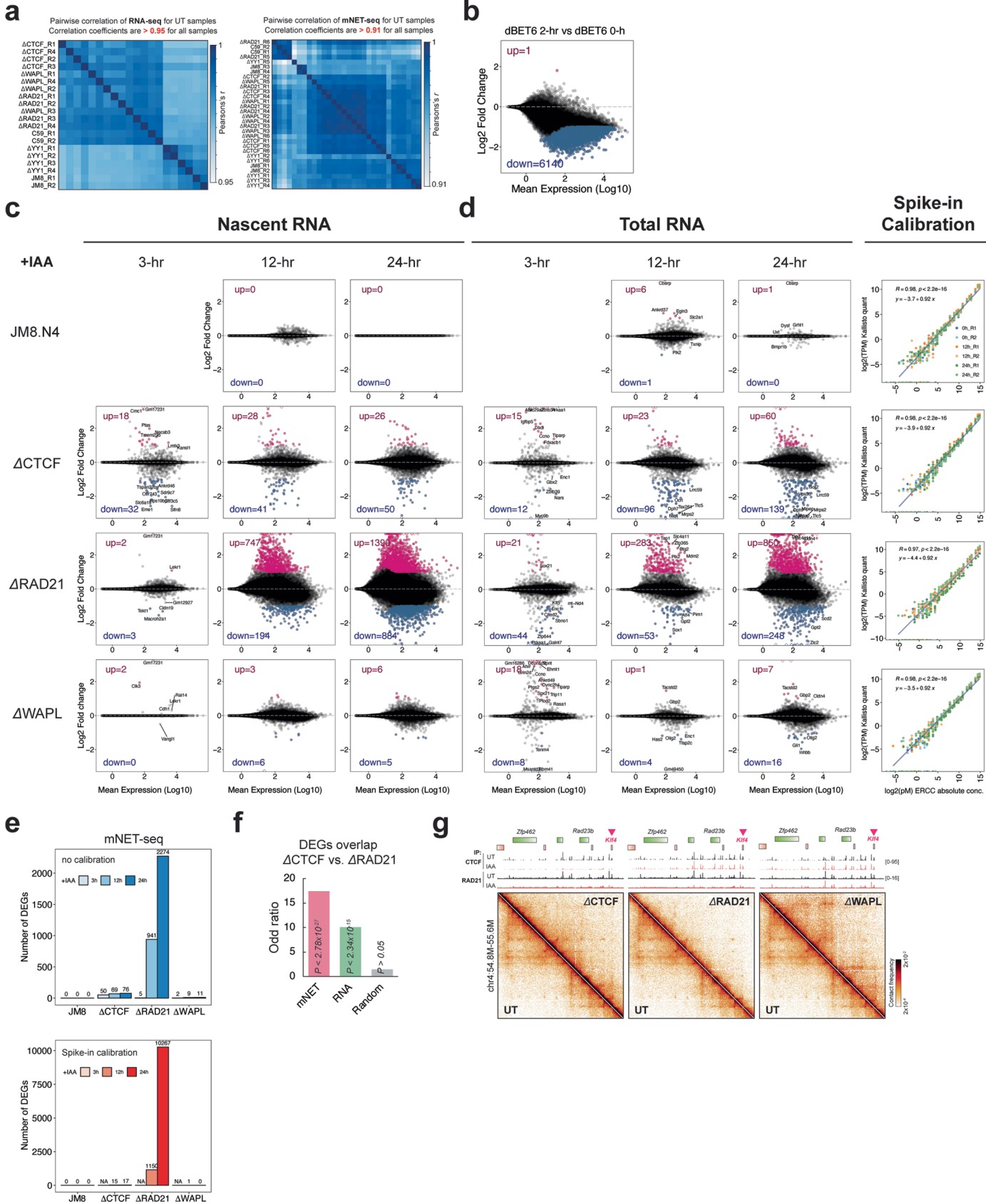

**Extended Data Fig. 5 | See next page for caption.**

**Extended Data Fig. 5 | RNA-seq and mNET-seq analysis. a**. Pairwise correlation of RNA-seq (left) and nascent RNA-seq (right) data for all UT samples. **b**. MA plots of nascent RNA-seq comparing UT wild type JM8.N4 mESCs with cells treated for 6 hours with the BRD inhibitor dBET6. Differentially expressed genes (DEGs) with q-value < 0.01 and 2-fold change are highlighted in pink (up) or blue (down). **c**. MA plot of nascent RNA-seq comparing wild-type JM8.N4 mESCs with ΔCTCF, ΔRAD21, or ΔWAPL degron cell lines after IAA treatment for 3, 12, and 24 hours. DEGs (q-value < 0.01 and 2-fold change) are highlighted in pink (up) or blue (down). **d**. MA plot of total RNA-seq comparing wild-type JM8.N4 mESCs with ΔCTCF, ΔRAD21, or ΔWAPL degron cell lines after IAA treatment for 3, 12, and 24 hours. DEGs (q-value < 0.01 and 2-fold change) are highlighted in pink (up) or blue (down). Quality of spike-in control (ERCC spike-in) for each condition is plotted in the right panel. **e**. Bar graph showing the summary of DEGs identified by nascent RNA-seq with (bottom) or without (top) spike-in calibration. **f**. Overlap of DEGs between different depletions and assays. Bar graph shows the odds ratio on the y-axis and is annotated with the corresponding p-value. Many DEGs are consistent between CTCF and cohesin depletion (Odd ratio > 10), suggesting that while CTCF and cohesin are required for the transcriptional maintenance of only a small subset of genes, those genes tend to require the presence of both factors. Statistical test: Fisher's exact test. **g**. Snapshots of Micro-C maps comparing chromatin interactions in the UT (top-right) and IAA-treated (bottom-left) cells surrounding *Klf4* locus. Contact maps are annotated with gene boxes and 1D chromatin tracks showing the ChIP-seq signal enrichment in the same region.

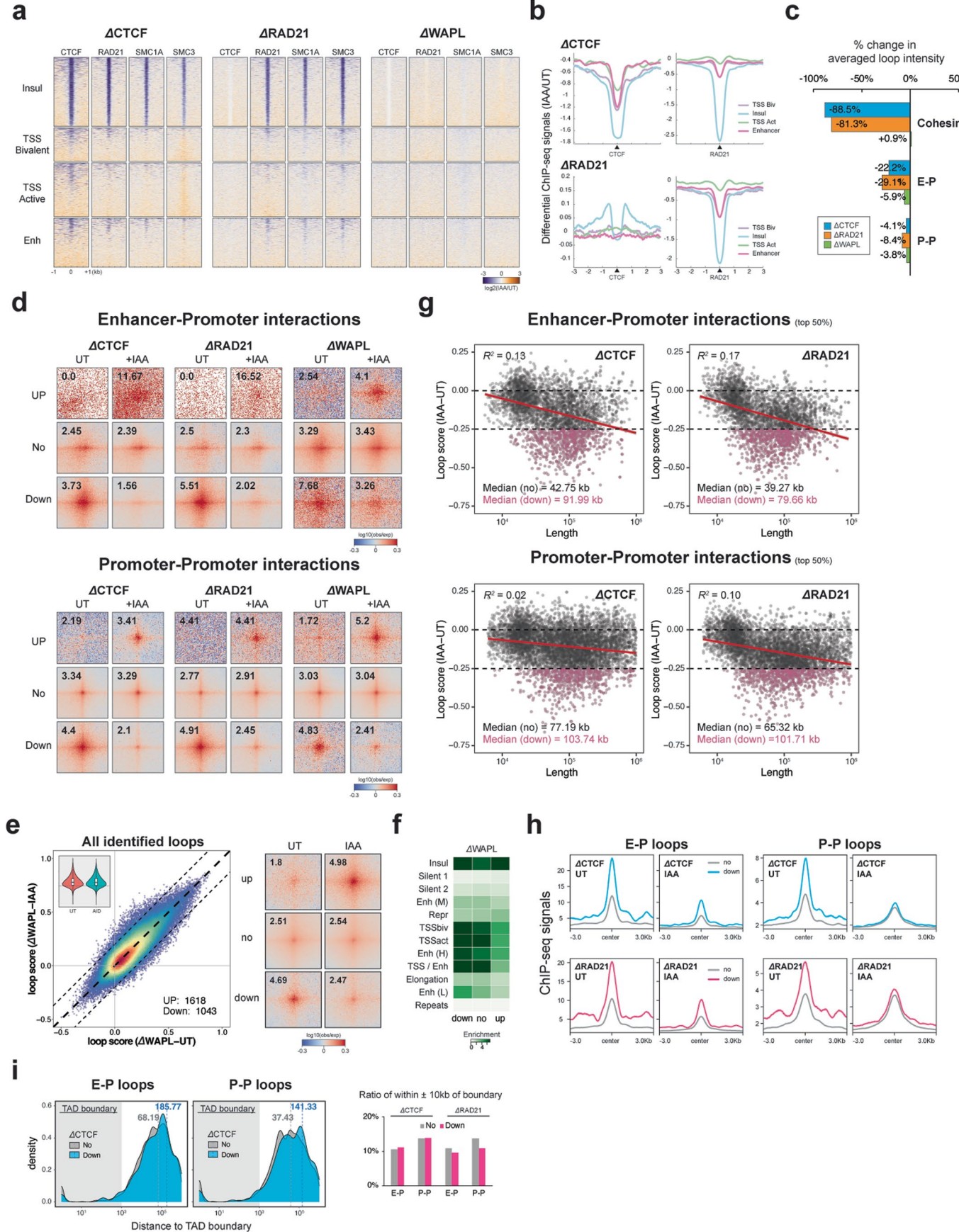

**Extended Data Fig. 6 | See next page for caption.**

**Extended Data Fig. 6 | Analysis of E-P/P-P interactions. a**. Heatmaps of differential ChIP-seq signals for CTCF, RAD21, SMC1A, and SMC3 comparing UT and IAA-treated degron cell lines. Heatmaps were plotted across a ±3-kb region around four major types of loop anchors. The colormap shows an increased signal (log2) in orange and a decreased signal in purple after IAA treatment. **b**. Profiles of differential ChIP-seq signals for CTCF or RAD21 comparing the UT and IAA-treated degron cell lines across the same regions as in a. **c**. Bar graph showing the changes in loop intensity quantified from Fig. 4e. **d**. APA is plotted for E-P (top) or P-P (bottom) loops that are grouped by up-regulated, down-regulated, or unchanged loops (right) in untreated and IAA-treated cells. **e**. Scatter plot of loop scores for the called loops in the UT and IAA-treated cells (left). The violin chart (inset) shows the distribution of loop scores for the UT and IAA-treated conditions. The box plot indicates the quartiles for the loop strength score distribution (see Fig. 1b). The pile-up contact maps are plotted with loops grouped by up-regulated, down-regulated, or unchanged loops (right). (control = 2; IAA = 3 biological replicates). **f**. Enrichment of the ChromHMM states at loop anchors grouped by up-regulated, down-regulated, or unchanged after IAA treatment. **g**. Scatter plot shows the relationship between loop length (x-axis) and the changes in loop intensity (y-axis) for E-P (top) and P-P (bottom) loops. **h**. Profiles of ChIP-seq signals for CTCF (top) or RAD21 (bottom) across a ±3-kb region around the anchors of E-P or P-P loops that are either unchanged (gray) or reduced after CTCF (blue) or RAD21 (pink) depletion. **i**. Length distribution of the unchanged or down-regulated E-P/P-P loops relative to TAD boundaries (left). Ratio of the unchanged (gray) or down-regulated (pink) E-P/P-P loop anchors located within ±10 kb of TAD boundaries (right).

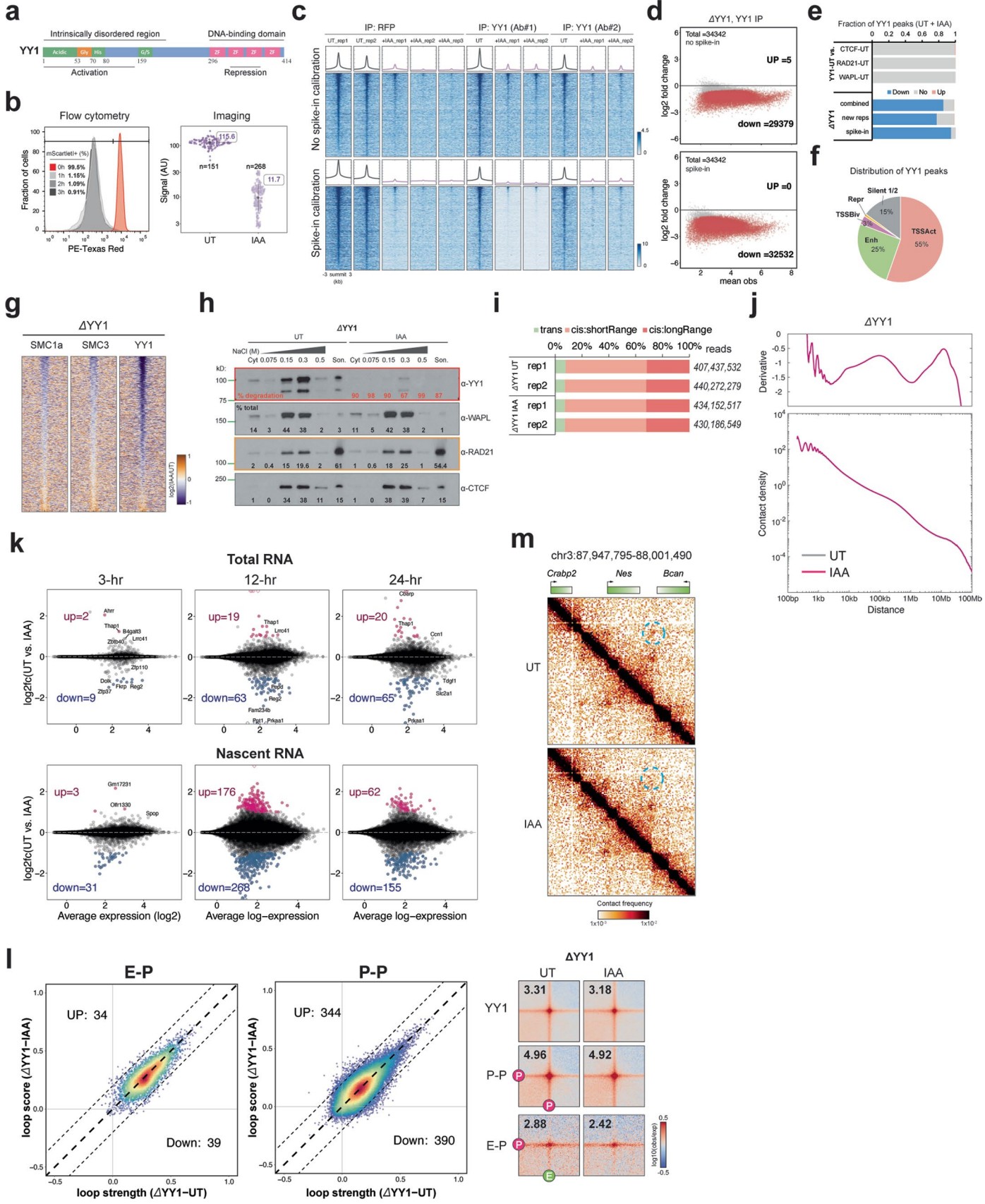

**Extended Data Fig. 7 | See next page for caption.**

**Extended Data Fig. 7 | Analysis of ChIP-seq, RNA-seq, mNET-seq, and Micro-C in YY1-AID cells. a**. YY1 protein domains. **b**. mScarletI signal in HaloTag-YY1 cells (YN11) treated with IAA by flow cytometry (left) and confocal imaging (right). **c**. YY1 ChIP-seq signal around YY1 peaks with or without spike-in calibration. Peaks called by MACS2 in the YY1-degron cells with antibodies against RFP or two different YY1 epitopes. Final YY1 peaks used throughout this manuscript summed from all peaks. Colormap: blue: maximum signal (log2), white: minimum signal. **d**. MA plots showing differential ChIP-seq peaks between UT and IAA-treated cells. Significantly changed peaks ($P_{adj}$ < 0.05) in red. X-axis: mean observations of UT and IAA-treated cells. Y-axis: log2 fold-change (UT/ IAA-treated). **e**. Differential ChIP-seq peak analysis. Fraction of down-regulated, up-regulated, or unchanged peaks after IAA treatment. Total number of peaks for each protein summed from all peaks in UT and IAA-treated cells. **f**. Percentage of YY1 peaks enriched with four primary types of ChromHMM states and silent chromatin. **g**. Heatmaps of differential YY1, SMC1A, and SMC3 ChIP-seq signals

after YY1 depletion. **h**. Immunoblots of cytoplasmic and nuclear proteins dissociating from chromatin at increasing salt concentrations (Extended Data Fig. 2f), probed with various antibodies (α). Son: solubilized chromatin; % of total: signal of each fraction / total signal; % of degradation: 1 – (signal of each fraction in IAA-treated / UT), normalized by total CTCF protein. **i**. Micro-C of UT and YY1-depleted cells. Right: total unique reads annotated for each replicate, consisting of trans- (inter-chromosome), short-range cis- (<20 kb), and long-range cis- (>20 kb) interactions. **j**. Genome-wide contact decaying P(s) analysis (bottom) and slope distributions of the P(s) curves (top) for UT cells. **k**. MA plot of total RNA-seq and nascent RNA-seq for YY1 degron 3 to 24 hours after IAA treatment. **l**. Scatter plots of loop scores (quantified using 2-kb-resolution Micro-C data) plotted for E-P or P-P loops in UT and IAA-treated cells. APA for YY1, E-P, or P-P anchored loops plotted for the ΔYY1 degron cell line in UT and IAA-treated cells. **m**. Micro-C maps comparing chromatin interactions in UT and IAA-treated ΔYY1 cells surrounding *Nes* gene.

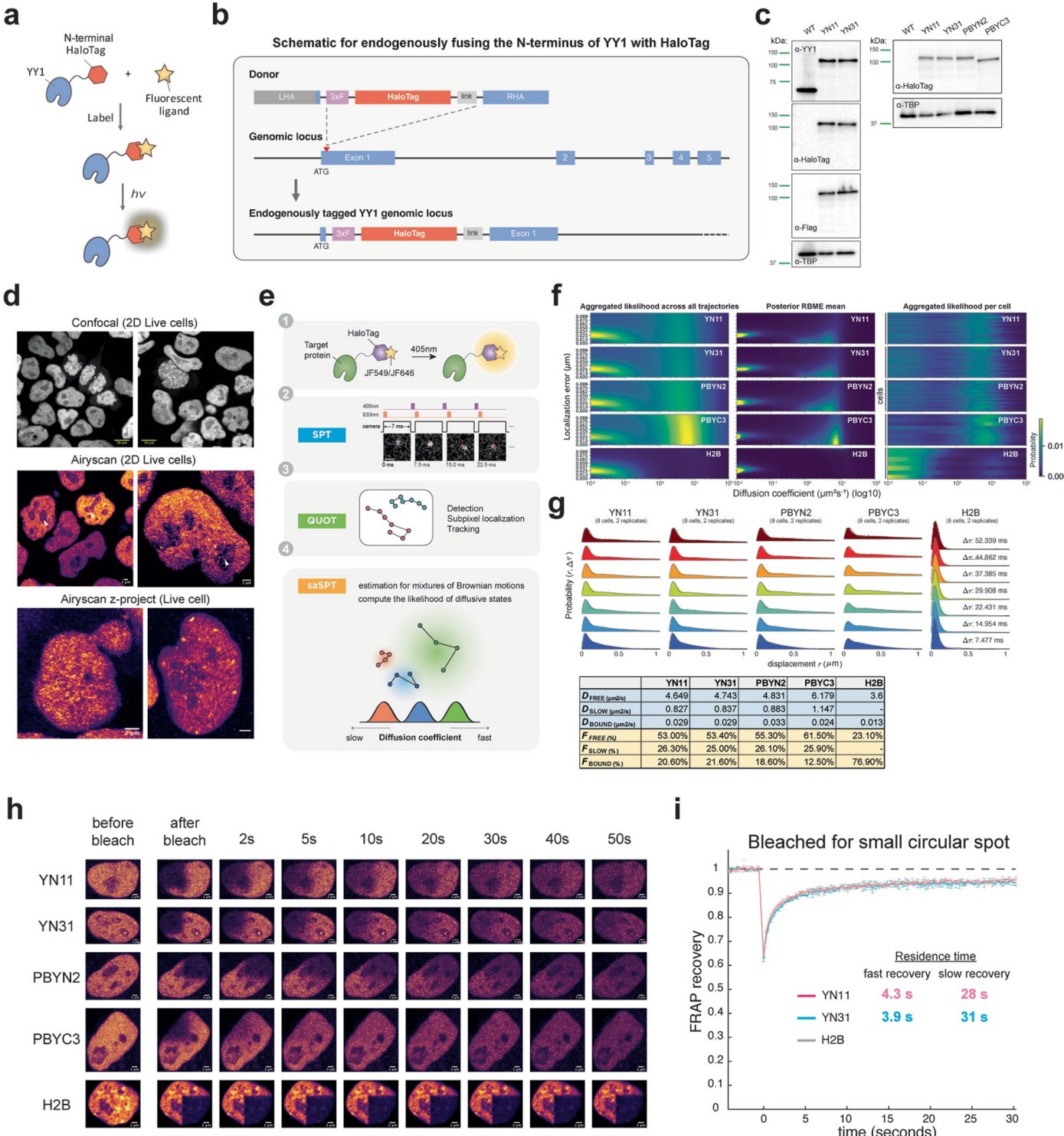

**Extended Data Fig. 8 | Dynamic analysis of YY1 protein. a.** Schematic for conjugating a fluorescent dye with the HaloTag-YY1 fusion protein, which emits fluorescence upon excitation by a specific wavelength. **b.** Schematic for endogenously fusing the N-terminus of YY1 with HaloTag. **c.** Immunoblots of wild-type (WT), HaloTag-YY1 knock-in (YN11 and YN31), and stably expressing Halotag-YY1/YY1-HaloTag (PBYN2 and PBYC3) mESC lines for YY1, HaloTag, and FLAG proteins. TBP was used as a loading control. We either added a HaloTag to the N-terminus of the endogenous YY1 via CRISPR-Cas9-mediated genome editing or ectopically expressed YY1 fused with HaloTag using a minimal L30 promoter via PiggyBac transposition. **d.** Confocal or Airyscan-resolved live-cell imaging for HaloTag-YY1 stained with 500 nM TMR Halo ligand. Arrow points to sporadic loci within the nucleolus. Images at the bottom panel are a z-projection with the mean signal. **e.** Schematic for the spaSPT experiment and the analysis

pipeline with Quot and SASPT[54]. **f.** Heatmaps of localization errors obtained by aggregated likelihood across all trajectories (left) or posterior marginalized localization error (middle) for clones YN11, YN31, PBYN2, PBYC3 and H2B. The distribution of the likelihood of diffusion coefficients (x-axis) for single cells (each row at the y-axis) is plotted on the right panel. **g.** spaSPT displacement histograms for YN11, YN31, PBYN2, PBYC3, and H2B. Raw displacement data for seven different lag times are shown with a three-state Spot-On model[53] fit overlaid. The inferred fractions and diffusion coefficients for each cell are shown in the table in the bottom panel. **h.** Snapshots of FRAP experiments for multiple time points from 'before bleach' to '50 sec after bleach'. **i.** FRAP analysis of YY1 bleached with a small circular spot. Error bars indicate the standard deviation of each acquired data point. (*n* = 8 cells examined over 2 independent experiments; error bars: the fitted curve ±SEM with 95% confidence interval).

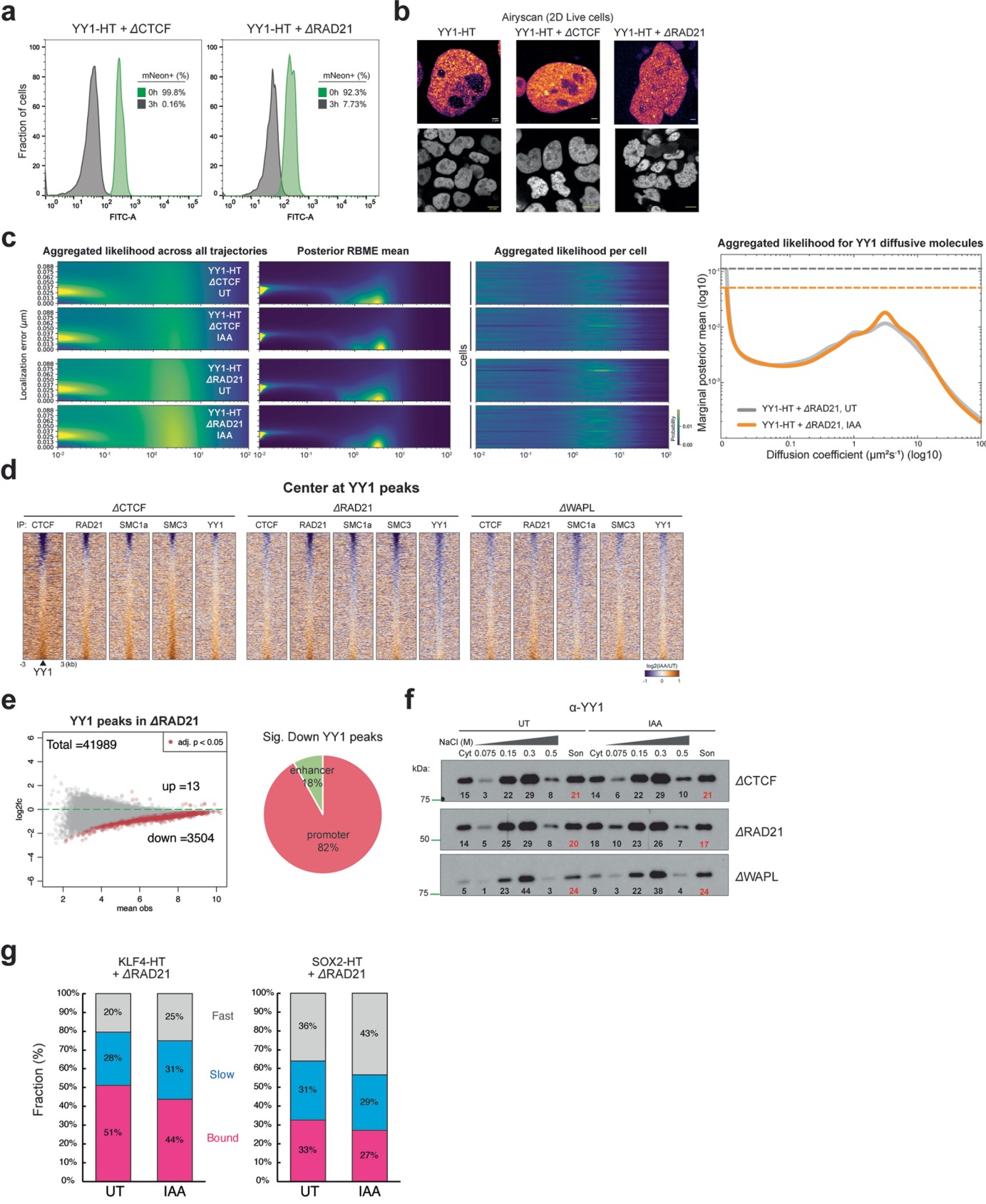

**Extended Data Fig. 9 | See next page for caption.**

**Extended Data Fig. 9 | YY1 dynamics after CTCF or cohesin loss. a**. Histogram of mNeonGreen intensity of HaloTag-YY1 CTCF or RAD21 degron cells (clones CD1 and RD35) treated with IAA for 0 or 3 hours. **b**. Airyscan-resolved live-cell imaging of HaloTag-YY1 stained with 500-nM TMR Halo ligand in wild type, CTCF-, or RAD21-depleted cells. **c**. Localization error heatmaps obtained by aggregated likelihood across all trajectories (first panel on the left) or posterior marginalized localization error (second panel) for UT or IAA-treated HaloTag-YY1 CTCF or RAD21 degron cells. Third panel: distribution of the diffusion coefficients likelihood (x-axis) for single cells (each y-axis row). Fourth panel: estimation of YY1 diffusion coefficients by regular Brownian motion with marginalized localization errors. **d**. Heatmaps of differential CTCF, RAD21, SMC1A, SMC3, and YY1 ChIP-seq signals in CTCF-, RAD21-, and WAPL-depleted cells. Peaks are centered on wild-type YY1 peaks across a ±3-kb region. Colormap: increased signal (log2) in orange and decreased signal in purple after IAA treatment.

**e**. MA plot showing the differential ChIP-seq peaks between UT and IAA-treated cells (left). Significantly changed peaks ($P_{adj} < 0.05$) colored in red. X-axis: mean observations of UT and IAA cells. Y-axis: log2 fold-change comparing UT and IAA-treated cells. Pie chart: percentages of downregulated YY1 peaks enriched at promoters and enhancers (right). **f**. Immunoblots of cytoplasmic (Cyt) and nuclear YY1 protein dissociating from chromatin at increasing salt concentrations (75-, 150-, 300- and 500-mM NaCl, Extended Data Fig. 2f) in CTCF, RAD21 or WAPL degron lines UT or treated with auxin (IAA). Son: sonicated, solubilized chromatin. Numbers represent the signal intensity of each fraction divided by the total signal intensity across all fractions (% of total). Red highlights the percent of YY1 retained on chromatin after all salt washes. **g**. Stacked bar graph of bound, slow, and fast diffusing KLF4 (left) or SOX2 (right) populations in UT and IAA-treated RAD21 degron cells, obtained by spaSPT analysis.

# Reporting Summary

## Statistics

For all statistical analyses, confirm that the following items are present in the figure legend, table legend, main text, or Methods section.

| n/a | Confirmed | |
|---|---|---|
| ☐ | ☒ | The exact sample size (*n*) for each experimental group/condition, given as a discrete number and unit of measurement |
| ☐ | ☒ | A statement on whether measurements were taken from distinct samples or whether the same sample was measured repeatedly |
| ☐ | ☒ | The statistical test(s) used AND whether they are one- or two-sided *Only common tests should be described solely by name; describe more complex techniques in the Methods section.* |
| ☒ | ☐ | A description of all covariates tested |
| ☐ | ☒ | A description of any assumptions or corrections, such as tests of normality and adjustment for multiple comparisons |
| ☐ | ☒ | A full description of the statistical parameters including central tendency (e.g. means) or other basic estimates (e.g. regression coefficient) AND variation (e.g. standard deviation) or associated estimates of uncertainty (e.g. confidence intervals) |
| ☐ | ☒ | For null hypothesis testing, the test statistic (e.g. *F*, *t*, *r*) with confidence intervals, effect sizes, degrees of freedom and *P* value noted *Give P values as exact values whenever suitable.* |
| ☒ | ☐ | For Bayesian analysis, information on the choice of priors and Markov chain Monte Carlo settings |
| ☒ | ☐ | For hierarchical and complex designs, identification of the appropriate level for tests and full reporting of outcomes |
| ☐ | ☒ | Estimates of effect sizes (e.g. Cohen's *d*, Pearson's *r*), indicating how they were calculated |

*Our web collection on statistics for biologists contains articles on many of the points above.*

## Software and code

Policy information about availability of computer code

| Data collection | Nikon NIS-Elements software (v 4.6) for collecting single-molecule imaging <br> Zeiss Zen Blue imaging software (v 3.1) for collecting confocal and Airyscan imaging |
|---|---|
| Data analysis | Sequencing analysis: TrimGalore (v 0.6.5) (https://github.com/FelixKrueger/TrimGalore), Bowtie2 (v 2.3.0) (http://bowtie-bio.sourceforge.net/bowtie2/index.shtml), samtools (v 1.9) (http://www.htslib.org/), deepTools (v 2.4.1 & v 3.5.0) (https://github.com/deeptools/deepTools), MACS (v 2.2.6) (https://github.com/macs3-project/MACS), MAnorm2 (v 1.0) (https://github.com/tushiqi/MAnorm2), HiC-Pro (v 2.11.3) (https://github.com/nservant/HiC-Pro), COOLER (v 0.8.10) (https://github.com/open2c/cooler), JUICER (v 1.22.01) (https://github.com/aidenlab/juicer), HiGlass (v 1.11.7) (https://github.com/higlass/higlass), cooltools (v 0.3.2) (https://github.com/open2c/cooltools), 3DChromatin_ReplicateQC (v 1.0.1) (https://github.com/kundajelab/3DChromatin_ReplicateQC), coolpuppy (v 0.9.5) (https://github.com/open2c/coolpuppy), Mustache (v 1.0.1) (https://github.com/ay-lab/mustache), chromosight (v 0.9.8) (https://github.com/koszullab/chromosight), chromHMM (v 1.22) (http://compbio.mit.edu/ChromHMM/), Bedtools (v 2.30.0) (https://bedtools.readthedocs.io/en/latest/index.html), kallisto (0.46.2) (https://github.com/pachterlab/kallisto), DeSeq2 (v 1.30.1) (https://bioconductor.org/packages/release/bioc/html/DESeq2.html), NRSA (v 2.0) (http://bioinfo.vanderbilt.edu/NRSA/), Rsubread (v 1.22.2) (https://bioconductor.org/packages/release/bioc/html/Rsubread.html) <br> Imaging analysis: ImageJ (v 1.53c) (https://imagej.nih.gov/ij/), Spot-on (v 1.0.4) (https://gitlab.com/tjian-darzacq-lab/spot-on-matlab), quot (v 3.0) (https://github.com/alecheckert/quot), SASPT (v 1.0) (https://github.com/alecheckert/saspt), Thunderstorm (v 1.3) (https://github.com/zitmen/thunderstorm) <br> Flow cytomotry analysis: FlowJo (v 10.3) (https://www.flowjo.com/) |

For manuscripts utilizing custom algorithms or software that are central to the research but not yet described in published literature, software must be made available to editors and reviewers. We strongly encourage code deposition in a community repository (e.g. GitHub). See the Nature Portfolio guidelines for submitting code & software for further information.

## Data

Policy information about availability of data

All manuscripts must include a data availability statement. This statement should provide the following information, where applicable:
- Accession codes, unique identifiers, or web links for publicly available datasets
- A description of any restrictions on data availability
- For clinical datasets or third party data, please ensure that the statement adheres to our policy

The Micro-C, ChIP-seq, nascent RNA-seq and total RNA-seq data generated in this publication are available in NCBI Gene Expression Omnibus through GEO Series accession number GSE178982. We also reanalyzed data that we previously generated in wild type mESCs (GSE130275). spaSPT raw data are accessible through DOI: 10.5281/zenodo.5035837. The reference genome mm10 and sacCer3 are available through UCSC genome browser (https://hgdownload.soe.ucsc.edu/downloads.html).

## Human research participants

Policy information about studies involving human research participants and Sex and Gender in Research.

| | |
|---|---|
| Reporting on sex and gender | n/a |
| Population characteristics | n/a |
| Recruitment | n/a |
| Ethics oversight | n/a |

Note that full information on the approval of the study protocol must also be provided in the manuscript.

# Field-specific reporting

Please select the one below that is the best fit for your research. If you are not sure, read the appropriate sections before making your selection.

☒ Life sciences ☐ Behavioural & social sciences ☐ Ecological, evolutionary & environmental sciences

For a reference copy of the document with all sections, see nature.com/documents/nr-reporting-summary-flat.pdf

# Life sciences study design

All studies must disclose on these points even when the disclosure is negative.

| | |
|---|---|
| Sample size | 1) Confocal/Airyscan imaging is mainly qualitative. We routinely collected at least six regions of interest (ROIs) with two different cell cultures on two separate days; 2) FRAP: We generally collected data from at least 6 cells per cell line per condition per day, and all presented data are from at least two independent replicates on different days; 3) SPT: We recorded movies for six cells per cell line or condition per day, and all data presented are from at least two independent experiments conducted corresponding to at least 12 cells and at least 100,000 localizations. No statistical method was used to predetermine sample size. We chose the sample size as suggested in the previous report (doi: 10.7554/elife.25776). |
| Data exclusions | 1) FRAP: We excluded data if the bleached spot is not detectable by our algorithm, and if the cell drifted away from the focus during acquisition; 2) SPT: We excluded data if the total localization was lower than 20,000 per cell. |
| Replication | Sequencing data: We generally collected at least 2 biological replicates per condition per day to gain statistical power. For some samples (Micro-C_ΔCTCF_IAA, Micro-C_ΔRAD21_IAA, Micro-C_ΔWAPL_IAA), we performed pilot tests so that the sample sizes will increase to 4. Imaging data: We generally collected at least 2 biological replicates per condition per day. Immunoblotting, biochemical fractionation, flow cytometry experiments were repeated and confirmed at least twice. All attempts at replication were successful. |
| Randomization | Samples were divided into groups based on genomic perturbations.  Each auxin-degradation sample is coupled with untreated control. |
| Blinding | The study does not involve therapeutic or animal experiments, so blinding was not necessary. The labeling of samples is also required for all computational analyses. |

# Reporting for specific materials, systems and methods

We require information from authors about some types of materials, experimental systems and methods used in many studies. Here, indicate whether each material, system or method listed is relevant to your study. If you are not sure if a list item applies to your research, read the appropriate section before selecting a response.

## Materials & experimental systems

| n/a | Involved in the study |
|-----|------------------------|
| ☐ | ☒ Antibodies |
| ☐ | ☒ Eukaryotic cell lines |
| ☒ | ☐ Palaeontology and archaeology |
| ☒ | ☐ Animals and other organisms |
| ☒ | ☐ Clinical data |
| ☒ | ☐ Dual use research of concern |

## Methods

| n/a | Involved in the study |
|-----|------------------------|
| ☐ | ☒ ChIP-seq |
| ☐ | ☒ Flow cytometry |
| ☒ | ☐ MRI-based neuroimaging |

# Antibodies

| Antibodies used | anti-CTCF Novus NBP2-52909 |
|-----------------|----------------------------|
| | anti-CTCF EMD 07-729 |
| | anti-CTCF Abcam ab128873 |
| | anti-Halo Promega G921A |
| | anti-RAD21 EMD 05-908 |
| | anti-RAD21 Abcam ab154769 |
| | anti-RAD21 Abcam ab154769 |
| | anti-V5 ThermoFisher R960-25 |
| | anti-YY1 Santa Cruz Biotechnology sc-7341 |
| | anti-YY1 Abcam ab38422 |
| | anti-YY1 Abcam ab109237 |
| | anti-YY1 Bethyl Laboratories Inc. A302-778A |
| | anti-YY1 Bethyl Laboratories Inc. A302-779A |
| | anti-RFP / anti-mScarletI Chromotek 6G6 |
| | anti-RFP / anti-mScarletI Rockland 600-401-379 |
| | anti-WAPL Proteintech 16370-1-AP |
| | anti-HA Abcam ab9110 |
| | anti-ACTB Sigma A2228 |
| | anti-OCT4 Santa Cruz Biotechnology sc-8628 |
| | anti-TBP abcam ab51841 |
| | anti-H2B ThermoFisher MA524697 |
| | anti-SMC1A Bethyl laboratories A300-055A |
| | anti-SMC3 Bethyl laboratories A300-060A |

| Validation | CTCF, RAD21, WAPL, YY1, HaloTag, V5, HA, RFP antibodies were validated by WB in cells depleted with the corresponding protein. SMC1A and SMC3 were validated by ChIP-seq, which signals are largely overlapped with RAD21. ACTB, TBP, OCT4, and H2B antibodies are well-validated by various studies and manufacturers. |
|------------|------------|

# Eukaryotic cell lines

Policy information about cell lines and Sex and Gender in Research

| Cell line source(s) | JM8.N4 mESC was obtained from the KOMP Repository at UC Davis. |
|---------------------|------------|
| Authentication | JM8.N4 mESCs were authenticated by whole-genome sequencing and morphology. |
| Mycoplasma contamination | JM8.N4 mESCs were pathogen tested using the IMPACT II test by IDEXX BioResearch (Westbrook, ME). All cells were negative for all pathogens, including Ectromelia, EDIM, LCMV, LDEV, MAV1, MAV2, mCMV, MHV, MNV, MPV, MVM, Mycoplasma pulmonis, Mycoplasma sp., Polyoma, PVM, REO3, Sendai, and TMEV. |
| Commonly misidentified lines (See ICLAC register) | No commonly misidentified line was used. |

# ChIP-seq

## Data deposition

☒ Confirm that both raw and final processed data have been deposited in a public database such as GEO.

☒ Confirm that you have deposited or provided access to graph files (e.g. BED files) for the called peaks.

| Data access links | https://www.ncbi.nlm.nih.gov/geo/query/acc.cgi?acc=GSE178982 |
|-------------------|------------|
| *May remain private before publication.* | |

Files in database submission

RTCC51A_S1_L001_R1_001.fastq.bz2
RTCC51B_S2_L001_R1_001.fastq.bz2
RTCC51C_S3_L001_R1_001.fastq.bz2
RTCC51D_S4_L001_R1_001.fastq.bz2
RTCC51E_S5_L001_R1_001.fastq.bz2
RTCC51F_S6_L001_R1_001.fastq.bz2
RTCC51G_S7_L001_R1_001.fastq.bz2
RTCC51H_S8_L001_R1_001.fastq.bz2
RTCC51I_S9_L001_R1_001.fastq.bz2
RTCC51J_S10_L001_R1_001.fastq.bz2
RTCC51K_S11_L001_R1_001.fastq.bz2
RTCC51L_S12_L001_R1_001.fastq.bz2
RTCC52A_S13_L002_R1_001.fastq.bz2
RTCC52B_S14_L002_R1_001.fastq.bz2
RTCC52C_S15_L002_R1_001.fastq.bz2
RTCC52D_S16_L002_R1_001.fastq.bz2
RTCC52E_S17_L002_R1_001.fastq.bz2
RTCC52F_S18_L002_R1_001.fastq.bz2
RTCC52G_S19_L002_R1_001.fastq.bz2
RTCC52H_S20_L002_R1_001.fastq.bz2
RTCC52I_S21_L002_R1_001.fastq.bz2
RTCC52J_S22_L002_R1_001.fastq.bz2
RTCC52K_S23_L002_R1_001.fastq.bz2
RTCC52L_S24_L002_R1_001.fastq.bz2
RTCC53A_S25_L003_R1_001.fastq.bz2
RTCC53B_S26_L003_R1_001.fastq.bz2
RTCC53C_S27_L003_R1_001.fastq.bz2
RTCC53D_S28_L003_R1_001.fastq.bz2
RTCC53E_S29_L003_R1_001.fastq.bz2
RTCC53F_S30_L003_R1_001.fastq.bz2
RTCC53G_S31_L003_R1_001.fastq.bz2
RTCC53H_S32_L003_R1_001.fastq.bz2
RTCC53I_S33_L003_R1_001.fastq.bz2
RTCC53J_S34_L003_R1_001.fastq.bz2
RTCC53K_S35_L003_R1_001.fastq.bz2
RTCC53L_S36_L003_R1_001.fastq.bz2
RTCC54A_S37_L004_R1_001.fastq.bz2
RTCC54B_S38_L004_R1_001.fastq.bz2
RTCC54C_S39_L004_R1_001.fastq.bz2
RTCC54D_S40_L004_R1_001.fastq.bz2
RTCC54E_S41_L004_R1_001.fastq.bz2
RTCC54F_S42_L004_R1_001.fastq.bz2
RTCC54G_S43_L004_R1_001.fastq.bz2
RTCC54H_S44_L004_R1_001.fastq.bz2
RTCC54I_S45_L004_R1_001.fastq.bz2
RTCC54J_S46_L004_R1_001.fastq.bz2
RTCC54K_S47_L004_R1_001.fastq.bz2
RTCC54L_S48_L004_R1_001.fastq.bz2
RTCC55A_S49_L005_R1_001.fastq.bz2
RTCC55B_S50_L005_R1_001.fastq.bz2
RTCC55C_S51_L005_R1_001.fastq.bz2
RTCC55D_S52_L005_R1_001.fastq.bz2
RTCC55E_S53_L005_R1_001.fastq.bz2
RTCC55F_S54_L005_R1_001.fastq.bz2
RTCC55G_S55_L005_R1_001.fastq.bz2
RTCC55H_S56_L005_R1_001.fastq.bz2
RTCC55I_S57_L005_R1_001.fastq.bz2
RTCC55J_S58_L005_R1_001.fastq.bz2
RTCC55K_S59_L005_R1_001.fastq.bz2
RTCC55L_S60_L005_R1_001.fastq.bz2
RTCC56A_S61_L006_R1_001.fastq.bz2
RTCC56B_S62_L006_R1_001.fastq.bz2
RTCC56C_S63_L006_R1_001.fastq.bz2
RTCC56D_S64_L006_R1_001.fastq.bz2
RTCC56E_S65_L006_R1_001.fastq.bz2
RTCC56F_S66_L006_R1_001.fastq.bz2
RTCC56G_S67_L006_R1_001.fastq.bz2
RTCC56H_S68_L006_R1_001.fastq.bz2
RTCC56I_S69_L006_R1_001.fastq.bz2
RTCC56J_S70_L006_R1_001.fastq.bz2
RTCC56K_S71_L006_R1_001.fastq.bz2
RTCC56L_S72_L006_R1_001.fastq.bz2
RTCC57A_S73_L007_R1_001.fastq.bz2
RTCC57B_S74_L007_R1_001.fastq.bz2
RTCC57C_S75_L007_R1_001.fastq.bz2

RTCC57D_S76_L007_R1_001.fastq.bz2
RTCC57E_S77_L007_R1_001.fastq.bz2
RTCC57F_S78_L007_R1_001.fastq.bz2
RTCC57G_S79_L007_R1_001.fastq.bz2
RTCC57H_S80_L007_R1_001.fastq.bz2
RTCC57I_S81_L007_R1_001.fastq.bz2
RTCC57J_S82_L007_R1_001.fastq.bz2
RTCC57K_S83_L007_R1_001.fastq.bz2
RTCC57L_S84_L007_R1_001.fastq.bz2
RTCC58A_S85_L008_R1_001.fastq.bz2
RTCC58B_S86_L008_R1_001.fastq.bz2
RTCC58C_S87_L008_R1_001.fastq.bz2
RTCC58D_S88_L008_R1_001.fastq.bz2
RTCC58E_S89_L008_R1_001.fastq.bz2
RTCC58F_S90_L008_R1_001.fastq.bz2
RTCC58G_S91_L008_R1_001.fastq.bz2
RTCC58H_S92_L008_R1_001.fastq.bz2
RTCC58I_S93_L008_R1_001.fastq.bz2
RTCC58J_S94_L008_R1_001.fastq.bz2
RTCC58K_S95_L008_R1_001.fastq.bz2
RTCC58L_S96_L008_R1_001.fastq.bz2
RTCC66A_S39_L004_R1_001.fastq.bz2
RTCC66E_S43_L004_R1_001.fastq.bz2
RTCC66B_S40_L004_R1_001.fastq.bz2
RTCC66F_S44_L004_R1_001.fastq.bz2
RTCC67A_S49_L005_R1_001.fastq.bz2
RTCC67B_S50_L005_R1_001.fastq.bz2
RTCC66C_S41_L004_R1_001.fastq.bz2
RTCC66G_S45_L004_R1_001.fastq.bz2
RTCC66D_S42_L004_R1_001.fastq.bz2
RTCC66H_S46_L004_R1_001.fastq.bz2
RTCC67C_S51_L005_R1_001.fastq.bz2
RTCC67D_S52_L005_R1_001.fastq.bz2
CTCF_degron_ut_input_merged_coverage.bw
CTCF_degron_ut_CTCF_IP_merged_coverage.bw
CTCF_degron_ut_RAD21_IP_merged_coverage.bw
CTCF_degron_ut_SMC1A_IP_merged_coverage.bw
CTCF_degron_ut_SMC3_IP_merged_coverage.bw
CTCF_degron_ut_YY1_IP_merged_coverage.bw
CTCF_degron_auxin_input_merged_coverage.bw
CTCF_degron_auxin_CTCF_IP_merged_coverage.bw
CTCF_degron_auxin_RAD21_IP_merged_coverage.bw
CTCF_degron_auxin_SMC1A_IP_merged_coverage.bw
CTCF_degron_auxin_SMC3_IP_merged_coverage.bw
CTCF_degron_auxin_YY1_IP_merged_coverage.bw
RAD21_degron_ut_input_merged_coverage.bw
RAD21_degron_ut_CTCF_IP_merged_coverage.bw
RAD21_degron_ut_RAD21_IP_merged_coverage.bw
RAD21_degron_ut_SMC1A_IP_merged_coverage.bw
RAD21_degron_ut_SMC3_IP_merged_coverage.bw
RAD21_degron_ut_YY1_IP_merged_coverage.bw
RAD21_degron_auxin_input_merged_coverage.bw
RAD21_degron_auxin_CTCF_IP_merged_coverage.bw
RAD21_degron_auxin_RAD21_IP_merged_coverage.bw
RAD21_degron_auxin_SMC1A_IP_merged_coverage.bw
RAD21_degron_auxin_SMC3_IP_merged_coverage.bw
RAD21_degron_auxin_YY1_IP_merged_coverage.bw
WAPL_degron_ut_input_merged_coverage.bw
WAPL_degron_ut_CTCF_IP_merged_coverage.bw
WAPL_degron_ut_RAD21_IP_merged_coverage.bw
WAPL_degron_ut_SMC1A_IP_merged_coverage.bw
WAPL_degron_ut_SMC3_IP_merged_coverage.bw
WAPL_degron_ut_YY1_IP_merged_coverage.bw
WAPL_degron_auxin_input_merged_coverage.bw
WAPL_degron_auxin_CTCF_IP_merged_coverage.bw
WAPL_degron_auxin_RAD21_IP_merged_coverage.bw
WAPL_degron_auxin_SMC1A_IP_merged_coverage.bw
WAPL_degron_auxin_SMC3_IP_merged_coverage.bw
WAPL_degron_auxin_YY1_IP_merged_coverage.bw
YY1_degron_ut_input_merged_coverage.bw
YY1_degron_ut_CTCF_IP_merged_coverage.bw
YY1_degron_ut_RAD21_IP_merged_coverage.bw
YY1_degron_ut_SMC1A_IP_merged_coverage.bw
YY1_degron_ut_SMC3_IP_merged_coverage.bw
YY1_degron_ut_YY1_IP_merged_coverage.bw
YY1_degron_auxin_input_merged_coverage.bw

```
YY1_degron_auxin_CTCF_IP_merged_coverage.bw
YY1_degron_auxin_RAD21_IP_merged_coverage.bw
YY1_degron_auxin_SMC1A_IP_merged_coverage.bw
YY1_degron_auxin_SMC3_IP_merged_coverage.bw
YY1_degron_auxin_YY1_IP_merged_coverage.bw
YY1_degron_ut_input_rep8_coverage_spikeIn.bw
YY1_degron_ut_input_rep9_coverage_spikeIn.bw
YY1_degron_ut_RFP_IP_rep8_coverage_spikeIn.bw
YY1_degron_ut_RFP_rep9_coverage_spikeIn.bw
YY1_degron_ut_YY1_abcam_IP_coverage_spikeIn.bw
YY1_degron_ut_YY1_bethyl_IP_coverage_spikeIn.bw
YY1_degron_auxin_input_rep8_coverage_spikeIn.bw
YY1_degron_auxin_input_rep9_coverage_spikeIn.bw
YY1_degron_auxin_RFP_IP_rep8_coverage_spikeIn.bw
YY1_degron_auxin_RFP_IP_rep9_coverage_spikeIn.bw
YY1_degron_auxin_YY1_abcam_IP_coverage_spikeIn.bw
YY1_degron_auxin_YY1_bethyl_IP_coverage_spikeIn.bw
```

Genome browser session
(e.g. UCSC)

no longer applicable

## Methodology

Replicates

We performed Spearman's correlation and Jaccard index to assess the reproducibility between samples. Please see Extended Figure 3a-d for the full result and description.

Sequencing depth

Read length: 51 bp
Single-end
File name Replicate Total reads Uniquely mapped reads
RTCC51A_S1_L001_R1_001.fastq.bz2 DCTCF_UT_CTCF_input_Rep1 28390977 21231461
RTCC51B_S2_L001_R1_001.fastq.bz2 DCTCF_UT_CTCF_ChIP_Rep1 32931783 27107390
RTCC51C_S3_L001_R1_001.fastq.bz2 DCTCF_UT_RAD21_ChIP_Rep1 29663143 22152005
RTCC51D_S4_L001_R1_001.fastq.bz2 DCTCF_UT_SMC1A_ChIP_Rep1 31920364 23816575
RTCC51E_S5_L001_R1_001.fastq.bz2 DCTCF_UT_SMC3_ChIP_Rep1 33634052 25358448
RTCC51F_S6_L001_R1_001.fastq.bz2 DCTCF_UT_YY1_ChIP_Rep1 33959459 24136167
RTCC51G_S7_L001_R1_001.fastq.bz2 DCTCF_IAA_YY1_ChIP_Rep1 35069865 26193557
RTCC51H_S8_L001_R1_001.fastq.bz2 DCTCF_IAA_CTCF_input_Rep1 31335397 23944570
RTCC51I_S9_L001_R1_001.fastq.bz2 DCTCF_IAA_CTCF_ChIP_Rep1 29485544 21870596
RTCC51J_S10_L001_R1_001.fastq.bz2 DCTCF_IAA_RAD21_ChIP_Rep1 36571058 26903685
RTCC51K_S11_L001_R1_001.fastq.bz2 DCTCF_IAA_SMC1A_ChIP_Rep1 34239220 25580983
RTCC51L_S12_L001_R1_001.fastq.bz2 DCTCF_IAA_SMC3_ChIP_Rep1 34809582 24676494
RTCC52A_S13_L002_R1_001.fastq.bz2 DRAD21_UT_CTCF_input_Rep1 25144424 18553371
RTCC52B_S14_L002_R1_001.fastq.bz2 DRAD21_UT_CTCF_ChIP_Rep1 35955736 29096124
RTCC52C_S15_L002_R1_001.fastq.bz2 DRAD21_UT_RAD21_ChIP_Rep1 27333608 20517789
RTCC52D_S16_L002_R1_001.fastq.bz2 DRAD21_UT_SMC1A_ChIP_Rep1 29481356 21800492
RTCC52E_S17_L002_R1_001.fastq.bz2 DRAD21_UT_SMC3_ChIP_Rep1 29484547 22039721
RTCC52F_S18_L002_R1_001.fastq.bz2 DRAD21_UT_YY1_ChIP_Rep1 36126503 25657564
RTCC52G_S19_L002_R1_001.fastq.bz2 DRAD21_IAA_YY1_ChIP_Rep1 30035710 22320231
RTCC52H_S20_L002_R1_001.fastq.bz2 DRAD21_IAA_CTCF_input_Rep1 29583069 23915783
RTCC52I_S21_L002_R1_001.fastq.bz2 DRAD21_IAA_CTCF_ChIP_Rep1 40806074 30018724
RTCC52J_S22_L002_R1_001.fastq.bz2 DRAD21_IAA_RAD21_ChIP_Rep1 37048981 27159280
RTCC52K_S23_L002_R1_001.fastq.bz2 DRAD21_IAA_SMC1A_ChIP_Rep1 32725809 24278048
RTCC52L_S24_L002_R1_001.fastq.bz2 DRAD21_IAA_SMC3_ChIP_Rep1 34037754 23713201
RTCC53A_S25_L003_R1_001.fastq.bz2 DWAPL_UT_CTCF_input_Rep1 24384191 18237846
RTCC53B_S26_L003_R1_001.fastq.bz2 DWAPL_UT_CTCF_ChIP_Rep1 33538857 27889963
RTCC53C_S27_L003_R1_001.fastq.bz2 DWAPL_UT_RAD21_ChIP_Rep1 25117348 18995694
RTCC53D_S28_L003_R1_001.fastq.bz2 DWAPL_UT_SMC1A_ChIP_Rep1 32636618 24552472
RTCC53E_S29_L003_R1_001.fastq.bz2 DWAPL_UT_SMC3_ChIP_Rep1 31072690 23659750
RTCC53F_S30_L003_R1_001.fastq.bz2 DWAPL_UT_YY1_ChIP_Rep1 33231275 23748753
RTCC53G_S31_L003_R1_001.fastq.bz2 DWAPL_IAA_YY1_ChIP_Rep1 29212730 21926504
RTCC53H_S32_L003_R1_001.fastq.bz2 DWAPL_IAA_CTCF_input_Rep1 27408935 22403634
RTCC53I_S33_L003_R1_001.fastq.bz2 DWAPL_IAA_CTCF_ChIP_Rep1 24545163 18722085
RTCC53J_S34_L003_R1_001.fastq.bz2 DWAPL_IAA_RAD21_ChIP_Rep1 39372702 29730206
RTCC53K_S35_L003_R1_001.fastq.bz2 DWAPL_IAA_SMC1A_ChIP_Rep1 33064459 25125531
RTCC53L_S36_L003_R1_001.fastq.bz2 DWAPL_IAA_SMC3_ChIP_Rep1 31227728 22462804
RTCC54A_S37_L004_R1_001.fastq.bz2 DYY1_UT_CTCF_input_Rep1 26011907 20213101
RTCC54B_S38_L004_R1_001.fastq.bz2 DYY1_UT_CTCF_ChIP_Rep1 33422785 27745568
RTCC54C_S39_L004_R1_001.fastq.bz2 DYY1_UT_RAD21_ChIP_Rep1 30087634 23277925
RTCC54D_S40_L004_R1_001.fastq.bz2 DYY1_UT_SMC1A_ChIP_Rep1 31801107 24652683
RTCC54E_S41_L004_R1_001.fastq.bz2 DYY1_UT_SMC3_ChIP_Rep1 38307498 30011393
RTCC54F_S42_L004_R1_001.fastq.bz2 DYY1_UT_YY1_ChIP_Rep1 33404639 23732272
RTCC54G_S43_L004_R1_001.fastq.bz2 DYY1_IAA_YY1_ChIP_Rep1 32150295 25007522
RTCC54H_S44_L004_R1_001.fastq.bz2 DYY1_IAA_CTCF_input_Rep1 30418758 25300258
RTCC54I_S45_L004_R1_001.fastq.bz2 DYY1_IAA_CTCF_ChIP_Rep1 29006914 22613689
RTCC54J_S46_L004_R1_001.fastq.bz2 DYY1_IAA_RAD21_ChIP_Rep1 36127905 28137822
RTCC54K_S47_L004_R1_001.fastq.bz2 DYY1_IAA_SMC1A_ChIP_Rep1 34775334 27165767

```
RTCC54L_S48_L004_R1_001.fastq.bz2 DYY1_IAA_SMC3_ChIP_Rep1 30250760 22575789
RTCC55A_S49_L005_R1_001.fastq.bz2 DCTCF_UT_CTCF_input_Rep2 29064053 21308974
RTCC55B_S50_L005_R1_001.fastq.bz2 DCTCF_UT_CTCF_ChIP_Rep2 28622933 23414213
RTCC55C_S51_L005_R1_001.fastq.bz2 DCTCF_UT_RAD21_ChIP_Rep2 28352119 20707341
RTCC55D_S52_L005_R1_001.fastq.bz2 DCTCF_UT_SMC1A_ChIP_Rep2 31567380 22900366
RTCC55E_S53_L005_R1_001.fastq.bz2 DCTCF_UT_SMC3_ChIP_Rep2 30499910 22361169
RTCC55F_S54_L005_R1_001.fastq.bz2 DCTCF_UT_YY1_ChIP_Rep2 32034114 22436868
RTCC55G_S55_L005_R1_001.fastq.bz2 DCTCF_IAA_YY1_ChIP_Rep2 30759222 23115704
RTCC55H_S56_L005_R1_001.fastq.bz2 DCTCF_IAA_CTCF_input_Rep2 29970750 22561877
RTCC55I_S57_L005_R1_001.fastq.bz2 DCTCF_IAA_CTCF_ChIP_Rep2 26922404 20134454
RTCC55J_S58_L005_R1_001.fastq.bz2 DCTCF_IAA_RAD21_ChIP_Rep2 30922446 22614630
RTCC55K_S59_L005_R1_001.fastq.bz2 DCTCF_IAA_SMC1A_ChIP_Rep2 36670078 27396409
RTCC55L_S60_L005_R1_001.fastq.bz2 DCTCF_IAA_SMC3_ChIP_Rep2 28327056 20318060
RTCC56A_S61_L006_R1_001.fastq.bz2 DRAD21_UT_CTCF_input_Rep2 28533417 21316335
RTCC56B_S62_L006_R1_001.fastq.bz2 DRAD21_UT_CTCF_ChIP_Rep2 39864412 32409731
RTCC56C_S63_L006_R1_001.fastq.bz2 DRAD21_UT_RAD21_ChIP_Rep2 24040588 18207715
RTCC56D_S64_L006_R1_001.fastq.bz2 DRAD21_UT_SMC1A_ChIP_Rep2 33326643 24543818
RTCC56E_S65_L006_R1_001.fastq.bz2 DRAD21_UT_SMC3_ChIP_Rep2 31213208 23450850
RTCC56F_S66_L006_R1_001.fastq.bz2 DRAD21_UT_YY1_ChIP_Rep2 31832642 22608147
RTCC56G_S67_L006_R1_001.fastq.bz2 DRAD21_IAA_YY1_ChIP_Rep2 34373025 25342380
RTCC56H_S68_L006_R1_001.fastq.bz2 DRAD21_IAA_CTCF_input_Rep2 29809808 23571235
RTCC56I_S69_L006_R1_001.fastq.bz2 DRAD21_IAA_CTCF_ChIP_Rep2 25932689 18848066
RTCC56J_S70_L006_R1_001.fastq.bz2 DRAD21_IAA_RAD21_ChIP_Rep2 36267390 26475367
RTCC56K_S71_L006_R1_001.fastq.bz2 DRAD21_IAA_SMC1A_ChIP_Rep2 33420982 24340999
RTCC56L_S72_L006_R1_001.fastq.bz2 DRAD21_IAA_SMC3_ChIP_Rep2 29109019 20424453
RTCC57A_S73_L007_R1_001.fastq.bz2 DWAPL_UT_CTCF_input_Rep2 29365053 21832681
RTCC57B_S74_L007_R1_001.fastq.bz2 DWAPL_UT_CTCF_ChIP_Rep2 31443884 25947147
RTCC57C_S75_L007_R1_001.fastq.bz2 DWAPL_UT_RAD21_ChIP_Rep2 27448853 20709397
RTCC57D_S76_L007_R1_001.fastq.bz2 DWAPL_UT_SMC1A_ChIP_Rep2 33625771 24696151
RTCC57E_S77_L007_R1_001.fastq.bz2 DWAPL_UT_SMC3_ChIP_Rep2 33589486 25104295
RTCC57F_S78_L007_R1_001.fastq.bz2 DWAPL_UT_YY1_ChIP_Rep2 33367843 23747764
RTCC57G_S79_L007_R1_001.fastq.bz2 DWAPL_IAA_YY1_ChIP_Rep2 31873279 23709130
RTCC57H_S80_L007_R1_001.fastq.bz2 DWAPL_IAA_CTCF_input_Rep2 29217470 24308581
RTCC57I_S81_L007_R1_001.fastq.bz2 DWAPL_IAA_CTCF_ChIP_Rep2 27092281 20386332
RTCC57J_S82_L007_R1_001.fastq.bz2 DWAPL_IAA_RAD21_ChIP_Rep2 38029125 28068355
RTCC57K_S83_L007_R1_001.fastq.bz2 DWAPL_IAA_SMC1A_ChIP_Rep2 34460598 25880556
RTCC57L_S84_L007_R1_001.fastq.bz2 DWAPL_IAA_SMC3_ChIP_Rep2 35324429 25450721
RTCC58A_S85_L008_R1_001.fastq.bz2 DYY1_UT_CTCF_input_Rep2 29925152 22627993
RTCC58B_S86_L008_R1_001.fastq.bz2 DYY1_UT_CTCF_ChIP_Rep2 33373385 27186327
RTCC58C_S87_L008_R1_001.fastq.bz2 DYY1_UT_RAD21_ChIP_Rep2 27369432 20907281
RTCC58D_S88_L008_R1_001.fastq.bz2 DYY1_UT_SMC1A_ChIP_Rep2 34935069 26316230
RTCC58E_S89_L008_R1_001.fastq.bz2 DYY1_UT_SMC3_ChIP_Rep2 29912875 22696556
RTCC58F_S90_L008_R1_001.fastq.bz2 DYY1_UT_YY1_ChIP_Rep2 37055177 26986523
RTCC58G_S91_L008_R1_001.fastq.bz2 DYY1_IAA_YY1_ChIP_Rep2 34477481 25774376
RTCC58H_S92_L008_R1_001.fastq.bz2 DYY1_IAA_CTCF_input_Rep2 31577762 25685691
RTCC58I_S93_L008_R1_001.fastq.bz2 DYY1_IAA_CTCF_ChIP_Rep2 28500684 21247823
RTCC58J_S94_L008_R1_001.fastq.bz2 DYY1_IAA_RAD21_ChIP_Rep2 38559687 28706476
RTCC58K_S95_L008_R1_001.fastq.bz2 DYY1_IAA_SMC1A_ChIP_Rep2 34072879 24484758
RTCC58L_S96_L008_R1_001.fastq.bz2 DYY1_IAA_SMC3_ChIP_Rep2 29588719 21873748
RTCC66A_S39_L004_R1_001.fastq.bz2 DYY1_UT_YY1_input_Rep3 31927388 21740506
RTCC66E_S43_L004_R1_001.fastq.bz2 DYY1_UT_YY1_input_Rep4 35563739 23903736
RTCC66B_S40_L004_R1_001.fastq.bz2 DYY1_UT_YY1_ChIP_anti-RFP_Rep1 42139234 26749526
RTCC66F_S44_L004_R1_001.fastq.bz2 DYY1_UT_YY1_ChIP_anti-RFP_Rep2 38365398 24266302
RTCC67A_S39_L004_R1_001.fastq.bz2 DYY1_UT_YY1_ChIP_Rep3 23956428 15852867
RTCC67B_S40_L004_R1_001.fastq.bz2 DYY1_UT_YY1_ChIP_Rep4 31129254 20309199
RTCC66C_S41_L004_R1_001.fastq.bz2 DYY1_IAA_YY1_inplAA_Rep3 29954279 20441546
RTCC66G_S45_L004_R1_001.fastq.bz2 DYY1_IAA_YY1_inplAA_Rep4 34928439 23263109
RTCC66D_S42_L004_R1_001.fastq.bz2 DYY1_IAA_YY1_ChIP_anti-RFP_Rep1 34351731 22521528
RTCC66H_S46_L004_R1_001.fastq.bz2 DYY1_IAA_YY1_ChIP_anti-RFP_Rep2 32396014 21297719
RTCC67C_S41_L004_R1_001.fastq.bz2 DYY1_IAA_YY1_ChIP_Rep3 25209642 16904397
RTCC67D_S42_L004_R1_001.fastq.bz2 DYY1_IAA_YY1_ChIP_Rep4 33004641 22032095
```

**Antibodies**

```
anti-CTCF Abcam ab128873
anti-CTCF Abcam ab128873
anti-YY1 Abcam ab38422
anti-YY1 Abcam ab109237
anti-SMC1A Bethyl laboratories A300-055A
anti-SMC3 Bethyl laboratories A300-060A
```

**Peak calling parameters**

```
ChIP-Seq read alignment: Bowtie (-n 2, -m 1)
ChIP-Seq peak calling: MACS2 (--nomodel --extsize 300)
```

**Data quality**

```
Sample_name Peak_number_FDR5% Peak_number_5-fold
CTCF_degron_auxin_CTCF_IP 26931 18438
CTCF_degron_auxin_Rad21_IP 14989 1129
CTCF_degron_auxin_Smc1a_IP 39093 3210
```

```
CTCF_degron_auxin_Smc3_IP 15393 777
CTCF_degron_auxin_YY1_IP 56799 16758
CTCF_degron_ut_CTCF_IP 81432 61404
CTCF_degron_ut_Rad21_IP 38762 24735
CTCF_degron_ut_Smc1a_IP 43472 17216
CTCF_degron_ut_Smc3_IP 20194 9213
CTCF_degron_ut_YY1_IP 46354 14594
Rad21_degron_auxin_CTCF_IP 90823 65960
Rad21_degron_auxin_Rad21_IP 11347 2803
Rad21_degron_auxin_Smc1a_IP  9511 762
Rad21_degron_auxin_Smc3_IP  5464 502
Rad21_degron_auxin_YY1_IP 42273 12447
Rad21_degron_ut_CTCF_IP 95139 72386
Rad21_degron_ut_Rad21_IP 43113 28993
Rad21_degron_ut_Smc1a_IP 41091 15957
Rad21_degron_ut_Smc3_IP 38509 21803
Rad21_degron_ut_YY1_IP 48329 16732
WAPL_degron_auxin_CTCF_IP 97574 74792
WAPL_degron_auxin_Rad21_IP 49451 33319
WAPL_degron_auxin_Smc1a_IP 38650 17317
WAPL_degron_auxin_Smc3_IP 45257 26165
WAPL_degron_auxin_YY1_IP 40640 12558
WAPL_degron_ut_CTCF_IP 98200 75332
WAPL_degron_ut_Rad21_IP 43445 29722
WAPL_degron_ut_Smc1a_IP 51296 21539
WAPL_degron_ut_Smc3_IP 36245 20304
WAPL_degron_ut_YY1_IP 40093 13294
YY1_degron_auxin_CTCF_IP 101925 75420
YY1_degron_auxin_Rad21_IP 45114 29887
YY1_degron_auxin_Smc1a_IP 47950 20330
YY1_degron_auxin_Smc3_IP 47098 25773
YY1_degron_auxin_YY1_IP 14627 3756
YY1_degron_ut_CTCF_IP 98775 74506
YY1_degron_ut_Rad21_IP 48854 32391
YY1_degron_ut_Smc1a_IP 51886 24617
YY1_degron_ut_Smc3_IP 46209 28173
YY1_degron_ut_YY1_IP 32203 10210
```

| Software | Bowtie2 (v 2.3.5.1), samtools (v 1.9), deepTools (v 3.5.0), MASC2 (v 2.2.6), MAnorm2 (v 1.0), |
| --- | --- |

# Flow Cytometry

## Plots

Confirm that:

☒ The axis labels state the marker and fluorochrome used (e.g. CD4-FITC).

☒ The axis scales are clearly visible. Include numbers along axes only for bottom left plot of group (a 'group' is an analysis of identical markers).

☒ All plots are contour plots with outliers or pseudocolor plots.

☒ A numerical value for number of cells or percentage (with statistics) is provided.

## Methodology

| Sample preparation | We measured RFP (mScarletI) intensity for checking YY1 degradation efficiency. YY1-miniIAA7 cells (clone YD39) were treated with 500 uM IAA for time points at 0, 1, 2, 3 hr. Ethanol-treated cells (negative control) were processed with the same procedure. |
| --- | --- |
| Instrument | BD Bioscience LSR Fortessa |
| Software | FlowJo (v10.3), FlowJo LL |
| Cell population abundance | Cells were detached and dissociated into single cells by trypsin, washed once by culture media, and resuspended into 1 mL of culture media. We typically have >95% of viable cells. |
| Gating strategy | We gated the main population on the FSC/SSC plot by excluding the apparent populations of cell debris and cell doublets. For RFP (mScarletI) gating, we defined ~99.5% of cells in the untreated cell as RFP-positive cells. |

☒ Tick this box to confirm that a figure exemplifying the gating strategy is provided in the Supplementary Information.

