## [Peer Review File · Nature Genetics]

Peer Review Information

Manuscript Title: Enhancer-promoter interactions and transcription are largely maintained upon acute loss of CTCF, cohesin, WAPL, or YY1

Corresponding author name(s): Robert Tjian, Xavier Darzacq

Reviewer Comments & Decisions:

Decision Letter, initial version:
--

Dear Dr. Tjian,

Your Article entitled "Enhancer-promoter interactions and transcription are maintained upon acute loss of CTCF, cohesin, WAPL, and YY1" has now been seen by 4 referees, whose comments are attached. While they find your work of potential interest, they have all raised serious concerns which in our view are sufficiently important that they preclude publication of the work in Nature Genetics, at least in its present form.

Overall, the reviewers have substantial concerns about the analysis and interpretation of the micro-C and mNET-seq data, about the dynamics of protein degradation, and thus about the strength of the novel conclusions that can be drawn at this stage.

Should further experimental and analytical data allow you to fully address these criticisms we would be willing to consider an appeal of our decision (unless, of course, something similar has by then been accepted at Nature Genetics or appeared elsewhere). This includes submission or publication of a portion of this work someplace else.

We hope you understand that until we have read the revised manuscript in its entirety we cannot promise that it will be sent back for peer review.

If you are interested in attempting to revise this manuscript for submission to Nature Genetics in the future, please contact me to discuss a potential appeal. Otherwise, we hope that you find our referees' comments helpful when preparing your manuscript for resubmission elsewhere.

Sincerely,

Tiago

Tiago Faial, PhD
Senior Editor

Nature Genetics

<https://orcid.org/0000-0003-0864-1200>

Referee expertise:

Referee #1: 3D genomics, CTCF

Referee #2: 3D genomics, imaging

Referee #3: 3D genomics, CTCF

Referee #4: chromatin dynamics, nascent transcription

Reviewers' Comments:

Reviewer #1:

Remarks to the Author:

[A pdf copy of this document is attached with figures]

Hsieh et al. perform micro-C, ChIP-seq, mNET-seq, and live single molecular imaging on cell lines with acutely degraded components of genome organization. The effect on 3D folding of depleting CTCF, RAD21 and WAPL has been previously studied by Hi-C and DNA imaging, yet the role of YY1 remains more mysterious. The primary versus secondary transcriptional consequences of disrupted 3D folding have remained challenging to disentangle. Therefore, the presented approaches could potentially shed light on the connection between genome organization and transcription. It was greatly appreciated that authors made their data available for review. However, at this time there are serious issues with cell line generation (WAPL and YY1 specifically), data processing, and interpretation of the micro-C and mNET-seq.

Concerns regarding micro-C analysis

1.1. Serious concern that peak/cot calling picks up accessible/visible regions rather than reflecting 3D folding at enhancers and promoters. Central conclusions of the study build on observing micro-C dots that persist after CTCF/RAD21/WAPL/YY1 degradation. However it is not demonstrated that the high micro-C signal between enhancers and promoters reflects 3D folding, as opposed to 1D biases, e.g. due to the high accessibility of these elements. The fact that E/Ps appear to give strong micro-C signal with their entire surrounding, sometimes tens of kbs away (e.g. figure 1f) is surprising. It remains unclear to what extent does this pattern reflect the sensitivity of micro-C to the technical accessibility of these regions (as opposed to reflected 3D organization) - see below.

1.1.1 Issues with high-resolution dot calling.

The authors rely on published computational methods for calling TADs and loops/dots. However, these methods were not developed for or validated at the resolutions the authors use. For example, the highest resolutions considered were 1 kb and 10 kb in the mustache and chromosight publications for human data. The dots called here are highly biased for increased coverage relative to the genome-wide average at high resolutions. The plot below shows genome-wide average coverage at 400-bp resolution as a black line, and average coverage under the called dot/loop anchors as a function of

map resolution (inferred as end-start) in blue:

[See attached pdf for figures]

Calling dots at very high resolution (< 1 kb) requires an approach that statistically models 1D coverage and verification that the dots called do not display a coverage bias. An example of such an approach is HiC-DC+ (PMID: 34099725), but even this approach would require extension and validation below 1 kb. If obtaining calls without coverage biases at sub-kb resolution is not feasible, 3D genomic analysis would benefit from being restricted to resolutions ≥ 2 kb, where dot calls show a less pronounced coverage bias, and existing computational tools have been validated. Crucially, if dot calls have a coverage bias, they cannot be interpreted as strictly reflecting 3D proximity.

1.1.2 Issues for interpretation related to possible visibility biases of enhancers and promoters. Key contributions of the manuscript would not stand if these dots were called due to technical biases instead of actual 3D folding:

"Specifically, the strength of E-P and P-P interactions positively correlates with the level of gene expression, while cohesin loops show no such correlation" This is expected if the micro-C signal at sub-kilobase resolution is sensitive to E-P accessibility.

"Remarkably, acute depletion of CTCF and cohesin had a negligible impact on the E-P and P-P loops, with ~80% of E-P contacts (Fig. 3c) and 90% of P-P contacts (Fig. 3d) remaining unaltered." Due to the coverage bias at dots called at higher resolutions, this likely reflects the visibility/accessibility of these regions, rather than their physical interaction in 3D. It sounds like the nature of the micro-C dots called between E-P (3D contacts vs. accessibility/technical biases) will remain uncertain until authors can profile a condition or treatment where the micro-C signal between E-P is affected genome-wide while accessibility remains unchanged.

1.1.3 Orthogonal experimental validation. If technically possible, an orthogonal method such as DNA FISH (e.g. ORCA?) based assays could be deployed to establish the nature of the E-P interactions called at high-resolutions in micro-C and address concerns of point 1.

1.2.1 Mapping & filtering, and genotypes. There appear to be issues with either mapping or cell line differences from the reference assembly leading to off-diagonal artifacts. It would be helpful to demonstrate that such features do not impact downstream analyses.

[See attached pdf for figures]

1.3 Concern that degron leakiness alters the perceived effect of degrading target proteins. Substantial variation exists between untreated (UT) conditions, which is not accounted for in the analysis or interpretation. For example, dots readily visible in YY1-UT appear absent in all other UT samples in the genomic region displayed in Figure 1.

Differential analyses relative to untagged reference cells, in addition to untreated cells, should be reported.

Concerns regarding transcription analysis by mNET-seq

2.1 Lack of positive control to conclude about the absence of change. Authors make strong and thought-provoking claims about the lack of widespread changes detected by their mNET-seq approach. This raises the question: can authors detect widespread transcriptional changes with their approach? The conclusions as stated require reporting experiments where a greater number of changes are detected, as to set the basal expectation.

2.1.1 It is essential that authors establish the time resolution of mNET-seq in their hands. In other words how quickly can authors expect to detect pervasive transcription changes? Assaying the CTCF/RAD21/WAPL-AID cells at later time points could establish that transcriptional dysregulation can eventually be detected in these cells with the mNET-seq approach. Given that CTCF and WAPL depleted cells do not have cell cycle defects it may make most sense to assay one or both of these lines at later time points (e.g. 6h, 12h or 24h).

2.1.2 It is essential that authors establish expectations after short-term degradation of factors more directly involved in transcription. Without this comparison it is difficult to conclude much about the small number of differentially transcribed genes after only 3h. YY1 depletion dysregulated ~325 genes, which is the most compared to CTCF/RAD21/WAPL. It is difficult to estimate what is the upper expectation for the number of dysregulated genes after acute depletion of a transcription factor. This could easily be achieved using mESCs where transcription factors more directly linked to promoter or enhancer activity can be depleted and performing the same mNET-seq experiments after 3hrs (e.g. existing OCT4, NANOG or SOX2-degron cells, or using BRD inhibitors).

2.2 Authors must confirm that the dysregulated genes are changing because of CTCF/RAD21/WAPL rather than auxin treatment.

Authors state that "This suggests that while CTCF and cohesin are required for the transcriptional maintenance of only a small subset of genes, those genes tend to require the presence of both factors." An alternative explanation is that auxin treatment in Tir-expressing cells affects those genes, irrespective of CTCF/RAD21 degradation (as reported by Liu et al. 2021 Nature Genetics PMID 33318687 by mRNA-seq at later timepoints). For example it is surprising that *cyp1b1*, a member of the cytochrome P450 family, is the highest upregulated gene across conditions. It would be important that authors perform mNET-seq in such their Tir1-only expressing cells treated with auxin, or rephrase their statements and interpretations to account for this possibility.

2.3 In the absence of internal calibration, is it possible that authors are missing global down- or up-regulation of transcription (e.g. see Arnold et al. 2020 Mol Cell PMID 34324863 for how internal calibration of mNET-seq can alter interpretations)? Please comment on the ability of the current protocol to detect global homogeneous changes in the main text, assuming that performing calibrated mNET-seq experiments is out of the question.

2.4 The authors do not report the technical reproducibility of their mNET-seq and RNA seq data. Please include pairwise correlations between all replicates in all conditions (using either gene body or promoter for mNET-seq).

2.5 Please include a suppl. table with the differentially transcribed genes detected in each mutant.

Further confirmation that these new AID cells undergo near-complete degradation would be helpful.

3.1 Controls for all AID targets. Western blot is unlikely to pick up 5-10% remaining protein. Given that authors introduced fluorescent tags together with their degrons it would be helpful to show flow-cytometry data of untreated and 3h-treated cells together with untagged cells, separately for CTCF, RAD21, WAPL (they already present the YY1 data in Ext. Data Fig. 5a). It would also be especially helpful if authors could report whether long-term depletion leads to the cellular defects previously reported for these factors in mouse ES cells (CTCF: stop in proliferation after 4 days; RAD21: block in mitosis and cell death within 2 days ; WAPL: exit from pluripotency after 2-4 days - See Liu et al. 2020 Nature Genetics PMID 33318687, Nora et al. 2017 Cell PMID 28525758, Kubo et al. 2021 NSMB PMID 28525758 etc...). Can authors comment on what happens to YY1-depleted cells, do they survive?

3.2 Controls for WAPL deficiency. Authors assume near-complete depletion of WAPL based on Western blot analysis with the HA tag, brought together with the AID. However given the micro-C signal it seems unlikely that the Wapl depletion triggered maximum cohesin retention. Extended Data Figure 3D indicates that the micro-C signal at pre-existing dots remains unchanged after WAPL depletion. It looks like Wapl loss is much less complete than other studies in the field, e.g. Haarhuis et al. 2017 (Cell PMID 28475897) and Wutz et al. (PMID: 29217591) where extended grids of peaks emerge and the derivative of contact density shifts strongly rightward. This makes the author's reports of little transcriptional change somewhat inconclusive, as it may simply be due to technical issues of their cell line generation. It would be helpful if authors could characterize the effect of their WAPL degradation: How much more cohesin is there on chromatin after 3hr of WAPL depletion in these cells? It would be informative if authors could perform calibrated ChIP-seq for core cohesin subunits after 3hrs of WAPL degradation (compared to either untreated and the WAPL untagged cells with all the other tags also present in the WAPL-AID line [CTCF-halo, RAD21-SNAP]). Alternatively authors could perform careful quantifications of chromatin-bound core cohesin subunits by western-blot, but this might be difficult as one would only expect a maximum 2-fold increase of cohesin, given that around 50% of cohesin is already bound before WAPL depletion. The authors should also update their reported conclusions in light of the potential technical confounders.

3.3 Issues with YY1 degradation.

The majority of YY1 peaks were diminished, but only ~50% of peaks (n = 15075) were called significantly changed by differential peak analysis. Does that mean the depletion is not complete? How do we interpret the seemingly complete depletion base of Western and flow cytometry then - are these assays not sensitive enough? Or is it because the YY1 ChIP-seq was not calibrated? Could it be because authors use the miniIAA7 tag with osTir1, instead of atAFB2 as recommended by citation 64, Li et al. Nat Methods PMID 31451765?

Conflating time elapsed with primary versus secondary effects on transcription. Author conclusions assume that the larger transcriptional consequences reported by others in similar cell lines likely reflects secondary consequences from a small subset of affected genes. While this could very well be true, it is also possible that direct primary transcriptional consequences of RAD21/CTCF/WAPL/YY1 depletion are slow-manifesting. Some evidence suggest that some of the slow-manifesting transcriptional picked up by previous studies could be consequences of direct cis-regulatory impacts: for example, over 80% of the genes down-regulated at mRNA level after one day of CTCF degradation by Nora et al. 2017 Cell PMID 28525758 had CTCF binding in their promoter. While it is challenging to disentangle slow primary defects from rapid secondary ones with the approaches used by the authors, the possibility that some primary consequences may be slow-manifesting should be discussed off the bat in the text, as it is an obvious expectation, rather than brought up only in the discussion under a

"time-buffering" model.

5. Please include annotated .gb files for all targeting constructs mentioned in Table S1, rather than stating "sequence available upon request". Please either make the vectors available on addgene or by simply add annotated .gb sequence files as supplementary data.

Minor points:

Captions are not always complete. e.g. it is not clear what microC data are plotted in Fig. 1: untreated YY1? something else?

Authors mention identifying 75,000 dots in the text but the 4-category breakdown in figure 1a-b only adds up to 42,000 (assuming the random and promoter-random categories are shuffled controls?). Please clarify and include the missing "other" category in the figure caption.

A heatmap showing correlations between micro-C replicates, or a table or pairwise correlations, should be added to the supplement.

It would be helpful if authors could refrain from using interchangeably "dots" and "loops". "Dots" seems the most accurate description of the peak-calling result, without making assumptions about the underlying conformation which can only be resolved through modeling or imaging.

Is there anything special about the genes immediately dysregulated after CTCF degradation? E.g. they have CTCF binding in their promoter? Are they sitting next to any relevant micro-C feature (stripe, dot, boundary...)? It would be helpful if authors confirmed their data showed these aspects of previous studies, or discuss why it might not.

"These studies have shown that while CTCF and cohesin play only a minor role in compartmentalization" While this is true for CTCF depletion, it was reported by many group that cohesin antagonizes compartmentalization, so that compartmentalization patterns are greatly affected in RAD21, NIPBL or WAPL depletion (e.g. Rao et al. 2017, Schwarzer et al. 2017, Haarhuis et al. 2017). Authors may want to reword to reflect this.

The following sentence makes it sound like measuring 3D interactions between E-P by micro-C at sub-kilobase resolution in mammalian genomes is already established. "We recently reported that Micro-C can effectively resolve ultra-fine 3D genome folding at nucleosome resolution, including E-P and P-P interactions, thus overcoming this limitation, 24–27." Ref 27 (Hansen et al. 2017) is cited to support that micro-C can effectively detect E-P and P-P interactions at nucleosome resolution. The sequencing depth in that study did not allow for nucleosome resolution. Indeed Hansen et al. 2017 reported around 14,000 loops vs 75,000 here. Similarly Ref. 24-25 are previous publications from the present author this time in *S. cerevisiae*, where there was no mention of E-P or P-P dots at nucleosome resolution. Please remove these citations here or rephrase the sentence to distinguish technology development (micro-C) from biological interpretation (E-P interactions).

"We note that Micro- C has much higher sensitivity for detecting E-P and P-P contacts compared to Hi-C, establishing Micro-C as a more suitable assay to study genome organization relevant to transcription regulation genome-wide in an unbiased manner." It is not fair for authors to state micro-

C is unbiased, it simply has milder biases than Hi-C, so this should be reworded.

"Newly transcribed RNAs generally have a higher correlation with E-P contacts than with compartments and TADs (Extended Data Fig. 1f)" what does correlation with TADs mean? Which score is used? Strength of insulation? Averaged over the two boundaries? Please clarify.

In paragraph "Acute depletion of CTCF, cohesin, and WAPL perturbs structural loops," please include a sentence in the main text explaining that these experiments were done without internal spike-in calibration, so authors cannot conclude about possible differences in overall binding (it is likely that the loss of CTCF/RAD21 signal is under-estimated without calibration). Without calibration, authors should temper the statement "While cohesin peaks are generally lost upon CTCF degradation, CTCF binding is unaffected by altering the level of cohesin on chromosomes" - it is possible that CTCF levels are overall higher or lower. "RAD21 or WAPL depletion caused only a 10 to 20% reduction in CTCF peaks" - please clarify "number of peaks" (current sentence could be interpreted as referring to signal intensity). Maybe worth rephrasing to also account for that in ref. 11 (Rao et al. Cell 2017) the CTCF ChIP-seq after RAD21 depletion was not calibrated either.

"Nevertheless, upon rapid induction (1 hour) of erythroid differentiation, YY1 triggers little or no change in H3K27ac and H3K27ac-anchored HiChIP interactions" - phrasing is confusing. Maybe authors could instead write that "YY1 redistribution triggers little or no change...", as this was not an inducible-degradation approach.

YY1 was suggested to play a more prominent role in differentiated cells compared to mouse ES cells. It would be fair to acknowledge this when citing ref. 61. Beagan et al. Genome Research PMID 28536180.

Please rephrase the following sentence to reflect that the authors only investigated the situation after 3hrs. "In summary, we find that, while CTCF, cohesin, and WAPL may regulate some gene expression, their acute depletion affects the transcription of only a handful of genes in mESCs, which largely encode pluripotency and differentiation factors."

Tir1 transgenes are missing in cartoons of Extended Fig. 2 for all the cell lines.

Reviewer #2:

Remarks to the Author:

In their manuscript, Enhancer-promoter interactions and transcription are maintained upon acute loss of CTCF, cohesin, WAPL, and YY1, Hsieh et al. explore the role of architectural proteins on E-P interactions and gene expression by a combination of micro-C and nascent transcriptomics in mESCs. Their findings are as follows:

- 1- Using micro-C in combination with a new loop caller, Mustache, they identify many new looping interactions between enhancer and promoters.
- 2- Acute (3hr) loss of CTCF, Rad21, WAPL, and YY1 does not perturb the frequency, location, or strength of most newly identified E-P interactions.
- 3- Acute (3hr) loss of CTCF, Rad21, WAPL, and YY1 does not affect global gene expression.
- 4- Cohesin loss alters YY1 binding dynamics.

The authors propose a time-buffering model, which I quite like and do think that it reconciles their and others' largely negative results related to acute loss of architectural proteins and gene regulation. However, I do have some concerns outlined below:

The main criticism of this paper pertains to the validity of the newly identified loops, which is the basis of their largely negative results and major conclusions. The authors utilize a new genome-wide, 3C-based method known as Micro-C along with a very new loop caller they recently developed (Mustache) and claim to identify new looping interactions not previously identified by Hi-C. Surprisingly, the loop caller alone adds a significant amount of new loops even from the authors' previous Micro-C analyses. While this makes sense in theory given the higher resolution MNase digestion would generate over convention, some of the features associated with these new loops does raise a few red flags.

That is, the authors find that these new loops correlate with transcription, frequently traverse TAD boundaries, and are robust to acute depletion of CTCF, cohesin, WAPL, and YY1. While these interactions may indeed be a novel and important discovery, the authors' conclusions are almost entirely based on these negative results. Importantly, they do not completely rule out the possibility that these new interactions are instead an artifact of the approach. For instance, transcriptional sites would of course also be associated with more open chromatin and higher MNase accessibility in general. It was not clear from the methods how the authors normalize their data to MNase digestion biases similar to how typical Hi-C data is normalized to the density and location of site-specific restriction digestion. It would also be good to know if frequent exceptions to this correlation exist wherein Micro-C is detecting new loops not associated with high levels of open chromatin or transcription. If so, do these also share the same features as mentioned above. Are the new loops dependent on transcription as could be determined via polII inhibition or mutational analysis? Surely, these resources are available for mESCs?

But most concerning is that they find that many (29-30%) of these loops traverse a TAD boundary. While it is true that boundaries have been found to be variable and TAD insulation is not strict, it is hard to imagine that stable or frequent E-P interactions traversing a boundary would represent a functional interaction. So far, a large body of literature point to the fact that cognate promoters and enhancers tend to be within the same TAD. For instance, this reviewer is not aware of any developmental gene whose functionally validated enhancer has been mapped outside of the same TAD and if so this is an extremely rare event. And recent estimates from eQTL studies suggest that >95% of variants are also located within the same TAD as the associated gene (Delaneau et al., 2019). Similar conclusions have been made from perturbation studies in mESCs (Sun et al., 2019). How do the authors reconcile their results with these findings?

Lastly, many studies have shown, from sequencing and imaging-based methods, that loss of cohesin or WAPL changes chromatin compaction both within and between TADs. Even indirectly, this would be expected to alter promoter-enhancer proximity. And yet, the authors show no change in number or strength of their newly identified interactions in the absence of several architectural proteins. Given the above, it seems extremely important that the authors validate their new looping interactions through orthogonal methodology such as 3C, Capture-C, or 5C that each would result in comparable resolutions to that of Micro-C. In addition, if the authors were to bin their data to artificially reduce their resolution to that comparable to Hi-C, would these interactions disappear upon reanalysis? At the very least, this would show that they are dependent on the unprecedented high resolution achieved by Micro-C.

Other comments:

1. It has recently become apparent that AID tagged versions of proteins, including Rad21, can be 'leaky' and show reduced activity or levels compared to their wildtype versions. How does the expression profile of their tagged cell lines compare to that of parental mESCs? Is it possible the small changes upon auxin treatment are due to an already perturbed system prior to treatment?
2. The conclusion that cohesin might act to recruit transcription factors in general based solely on YY1 behavior seems like a bit of a leap. The authors might want to tone down this conclusion.

Reviewer #3:

Remarks to the Author:

CTCF, cohesin complex, and YY1 have been proposed to be key players shaping the chromatin architecture in mammalian cells, with CTCF and the cohesin complex involved in mediating long-range chromatin loops between convergent CTCF binding sites through a loop extrusion process, while YY1 shown to mediate enhancer-promoter contacts by virtue of its DNA binding to both classes of elements and its dimerization potential. Additionally, cohesin has also been suggested to mediate enhancer-promoter interactions due to its association with mediator complex and loading by Nipbl protein, which binds to both promoters and enhancers. While abundant literature supports a role for CTCF and cohesin in chromatin organization especially in defining topologically associating domains (TADs) and long-distance chromatin loops, their function in chromatin contacts between enhancers and promoters is less clear and has been the subject of intense debate in recent years. Also under intense debate is whether chromatin interactions between enhancers and promoters actively contribute to enhancer dependent gene activation. Recent studies using auxin inducible degrons to acutely deplete CTCF or cohesin subunits showed that CTCF and cohesin are dispensable for transcription of most genes in a variety of cell systems tested. By contrast, acute depletion of YY1 has been shown to lead to dramatic changes in gene expression in mouse ES cells. These previous studies raised questions about what exactly is the role that CTCF and cohesin play in enhancer-promoter communication and gene activation, and whether different proteins (such as YY1) are involved enhancer-promoter contacts and enhancer dependent transcription from target genes.

The manuscript by Hsieh et al. attempts to address these two questions and to provide clarity on the role of CTCF, cohesin and YY1 in enhancer-promoter contacts and the role of enhancer-promoter loops in gene activation. Like previous studies, the authors used the auxin inducible degron system to deplete CTCF, Rad21 subunit of cohesin, cohesin unloader protein WAPL and YY1 protein, in a mouse embryonic stem cell line. But in contrast to the previous work, the experimental design in the current report involved a shorter protein depletion time (3 hours vs. 24hr or longer), which could avoid the secondary effects during long-term protein depletion. Additionally, the authors used Micro-C, a variant of Hi-C technique, to achieve higher resolution (a few hundred base pairs versus 5- or 10-kb resolution) and sensitivity of loop detection especially between enhancers and promoters. Furthermore, the authors employed mNET-seq to measure changes in nascent transcript levels genome-wide after protein depletion. Their main findings are: (1) depletion of these proteins did not cause significant changes in enhancer-promoter interactions, even for YY1; and (2) depletion of these proteins resulted in minimal changes in gene expression. The authors also used live cell single particle imaging techniques to study the dynamics of YY1 binding to chromatin upon depletion of cohesin, and found an unexpected but modest role for cohesin to retain YY1 on chromatin. Based on these observations, the author concluded that "neither transcriptional condensates, CTCF, cohesin, WAPL, nor YY1 are generally required for the short-term maintenance of E-P interactions and the subsequent expression of most genes after acute depletion and loss of function" (Line 494). To reconcile with

previous genetic evidence implicating CTCF and cohesin's role in regulating promoter-enhancer interactions and transcription at some genes, the authors further proposed a "time-buffering model", "where architectural proteins generally contribute to the establishment, but not the short-term maintenance, of E-P interactions and gene expression" (Line 548).

General assessment:

Overall, the current work reports several very interesting observations related to the role of CTCF, cohesin and YY1 in enhancer promoter interactions and transcription, with some in direct contradiction with previous studies, and others confirmatory. An important contribution of this work lies in the use of a well-controlled experimental system and high precision technical approaches. The genetically engineered cell lines to enable acute depletion of CTCF, YY1, and cohesin complex in cells are valuable tools to the field. The micro-C and mNET-seq datasets along with other genomic datasets in this experimental model system also provide a valuable resource for the community to investigate the role of these proteins in chromatin organization and gene regulation. Surprisingly, the authors found that cohesin depletion did not disrupt enhancer-promoter contacts in general, and that YY1 loss had no significant impact on both enhancer promoter contacts and gene expression, which is in direct contradiction to Weintraub et al. (Cell 2017, ref #62). These new results are interesting and certainly raise new questions.

On the other hand, the experimental evidence provided in the current work does not fully support the sweeping conclusion that CTCF and cohesin play no role in the maintenance of enhancer promoter interactions (as the title implies), nor the statement about transcriptional condensates (see specific comments below for details). Further, some of the findings described in the current work are not new. For example, the observations that CTCF was largely dispensable for enhancer-promoter interactions and general transcription are consistent with previous studies (for example, Nora et al. Cell 2017; Kubo et al. NSMB 2021, which the authors cited as ref #10 and #37, respectively). The observation that cohesin loss resulted in minimal changes in gene expression was also reported before (for example, Rao et al. Cell 2017, ref #11 in the manuscript). Therefore, the degree by which this work advances our understanding of the maintenance of enhancer-promoter interactions and their function is currently unclear.

Main comments:

1. The experimental evidence presented does not fully support the authors' assertion that enhancer-promoter (E-P) interactions are unaffected after depletion of CTCF or cohesin. A closer inspection of Figure 3C suggests that E-P interactions were generally weakened after depletion of Rad21 and to a lesser degree CTCF - the center of the data cloud representing E-P interactions is below the diagonal line, and this trend is not obvious for P-P interactions. Therefore, one could say that there is a general and weak effect for cohesin loss on E-P interactions, consistent with residual levels of cohesin retained on chromatin after short term depletion.

2. Similarly, the assessment of YY1's effect on chromatin interactions was also problematic. The authors in Figure 4i examined changes of chromatin interactions on cohesin mediated loops (strong loops), and found little change. But what about E-P interactions? It would be important for the authors to provide a similar plot for the YY1 depleted cells as in Figure 4C for CTCF and cohesin depletion. The authors did report ~800 loops changed after YY1 depletion (Line 320), but Figure 4e indicated only

436 as “down” loops. So it is unclear how the authors reached the conclusion that loss of YY1 did not affect chromatin interactions between promoters and enhancers, especially when only 50% of YY1 binding sites showed significantly reduced binding after YY1 depletion (Line 308).

3. A major feature of the experimental design is the short duration of protein depletion (3 hours). This is also a potential weakness, as it is apparent that 3-hour IAA treatment did not fully deplete the proteins. Residual ChIP-seq signal for CTCF and cohesin were readily detectable on chromatin at CTCF peaks (Ext Fig 2f) and E-P & P-P loop anchors (Ext Fig 4e). Same can be said for YY1 (Figure 4b, Ext Fig 5C). It is therefore hard to rule out the possibility that residual proteins bound to chromatin might contribute to maintenance of chromatin interactions in the short-term depletion.

4. The authors concluded that “transcriptional condensates” are not generally required for “short-term maintenance of E-P interactions” (Line 493), by drawing on previous literature (Ref #57, 58 and 88), without providing any new evidence in the current work. This statement therefore needs to be revised to better reflect this fact.

5. It is interesting that cohesin loss results in reduced population of immobile and presumably chromatin bound YY1, but this evidence alone is insufficient to lead to the model that cohesin facilitates target search by transcription factors.

Minor comments:

1. With multiple loop callers, authors show that micro-C can identify more fine range E-P and P-P interactions compared to Hi-C. What percent of the E-P, P-P loops called by previous Hi-C data are also detected by Micro-C and vice versa? What is the overlap? What about with HiChIP /Plac-Seq data? This analysis is important to allow the readers to assess the specificity and sensitivity of the data.

2. Titles for the scatterplots missing (e.g., Ext Fig 4c)

3. Extended Fig 4d. A subset of downregulated loop anchors in WAPL depletion cells seem to show more bias towards enhancer and promoter regions. Do these regions show any differences in Rad21 binding?

4. Upon depletion of YY1 itself, there is a modest decrease of Rad21 binding. Which Rad21-bound regions show reduction upon YY1 depletion, what are their characteristics? Do they correspond to loop anchors, or show any change in loop strength?

Reviewer #4:

Remarks to the Author:

In this paper, Hsieh et al. leverage high resolution Micro-C and nascent RNA (mNET-seq) data to understand the roles played by CTCF, cohesin, and YY1 in the maintenance of enhancer-promoter interactions. The authors use mESC cell lines in which CTCF, WAPL, cohesin, and YY1 are tagged for rapid (3h) degradation. The authors argue that the marginal effect of CTCF and cohesin on overall gene expression level, as observed previously in the literature, is explained by cis-regulatory loops between enhancers and promoters being relatively insensitive to the depletion of these architectural

proteins. The authors then show through similar experiments that the depletion of YY1, previously suggested as an important factor for enhancer-promoter loops, also had only marginal effects on enhancer-promoter loops and transcription. Finally, using a combination of confocal microscopy and single molecule imaging, the authors characterize the dynamics of YY1 binding to chromatin and show that the depletion of RAD21, but not CTCF significantly reduces YY1-chromatin binding by increasing its searching time. This paper examines an important aspect of the relationship between chromatin folding and transcription. It is a well written paper; experiments are well justified and expertly executed. The results are timely. However, while the data is comprehensive and solid, I do have some major concerns about data analysis and interpretation that need to be addressed prior to publication.

Major comments:

(1)* The authors argue that 80-90% of loop interactions between enhancers and promoters (or promoters and promoters) are 'unaltered' by depletion of cohesin or CTCF. I am not convinced that the author's data supports this conclusion. Setting aside the ~20% of enhancer-promoter interactions that the author's analysis turns up as significantly reduced, there does appear to be a small reduction in contact frequency more broadly. Carefully examining the scatterplots in Fig. 3C shows that the bulk of enhancer promoter contacts are comfortably below the 0, 1 line, indicating lower contact frequency in AID than control cells. If this shift is not indeed explained by normalization issues (see points below), I think the paper would improve if the authors provided a more nuanced presentation of these results, acknowledging the shift that is clearly present in their data.

(2)* All of the Micro-C analysis in this manuscript compares contact frequency between different conditions with very large, global differences in contact frequency. As with any other sequencing assay, contacts that are lost from (say) CTCF-CTCF loop anchors or TADs upon CTCF or Rad21 depletion will be redistributed to other loci. In 1D data (e.g., mNET-seq/ ChIP-seq/ etc.), spike-in controls can be used to normalize between different samples in absolute terms, allowing the discovery of truly global changes. While I recognize there is no easy analog of a spike-in for a Micro-C experiment, I would like the authors to discuss the importance of normalization, and the ways in which normalization issues could impact the author's interpretations.

(3)* MNase, depending on the digestion conditions, does not cut uniformly across the entire genome, but rather it has specific biases for accessible regions. Any change in the distribution of MNase cut frequency near promoter/ enhancer regions between wild-type and CTCF, YY1, or Rad21 depletion could alter the contact frequency observed in contact maps. The authors should evaluate whether there are differences in MNase cutting efficiency between different conditions by examining the 1D signal in their Micro-C libraries.

(4)* It generally is not clear whether the authors measure transcription after removing the pause peak, focusing on expression in the gene body (which most directly correlates with mRNA), or whether paused Pol II is included. As the different stages of transcription initiation and pause might have different relationships with contact frequency, it is crucial to distinguish clearly between them in their presentation.

(5)* Data showing that RAD21 serves as a "platform" for YY1, or possibly other transcription factors, is extremely weak. I am convinced by the data showing that it increases search times (although I will defer to an expert on the rigour of the microscopy experiments). How do the authors feel about an alternative model in which the act of loop extrusion acts to stir the nucleoplasm, resulting in higher

diffusion and faster search times? There are likely other indirect explanations as well, and I would encourage the authors to consider these mechanisms before jumping right to IDR-IDR interactions. Related to this point - unless the authors can directly show that protein-protein interactions are necessary for the effect of RAD21 depletion on YY1 search time, I would encourage them to drop the use of the word "platform" and other terms that imply a direct interaction.

Minor comments:

- * In figure 1d, the axes should be better described. Additionally, in P-P contacts, clarify whether the data points represent an average of the two promoters.
- * Throughout the manuscript, add axes description to the APA heatmaps explaining, for example, which of the rows/columns represent promoters/enhancers.
- * Please add transcriptional orientation (i.e. what is the direction of the transcription unit) to the APA heatmaps.
- * A clearer explanation on how anchors were chosen and defined as promoters and enhancers would also help readers interpret the data. What dataset was used to define TSSs or whether it was accomplished by the MNase data should be clarified.
- * Please use the acronym polyadenylation cleavage site (PAS), rather than TES, in figure 1C. I assume TES stands for transcription end site, but presumably the point used is the end of a gene annotation. Transcription continues past the end of the PAS, as is clearly shown in the authors' plots.
- * Some interesting differences are shown between P-P and E-P pairs (Fig. 1D and 3H) but are ignored in the text. Are these associated with the different levels in initiation and pausing?
- * Figure 3b shows that loops from up/ down/ unchanged groups are enriched in functional elements involved in transcriptional regulation. Taken along with the pattern shown in 3a, this further suggest that the effect of CTCF and RAD21 depletions on cis-regulatory elements looping is underestimated by the authors.

Although we cannot publish your paper, it may be appropriate for another journal in the Nature Portfolio. If you wish to explore the journals and transfer your manuscript please use [redacted] manuscript transfer portal. If you transfer to Nature journals or the Communications journals, you will not have to re-supply manuscript metadata and files. This link can only be used once and remains active until used.

All Nature Portfolio journals are editorially independent, and the decision on your manuscript will be taken by their editors. For more information, please see our manuscript transfer FAQ page.

Note that any decision to opt in to In Review at the original journal is not sent to the receiving journal on transfer. You can opt in to In Review at receiving journals that support this service by choosing to modify your manuscript on transfer. In Review is available for primary research manuscript types only.

Author Rebuttal to Initial comments

Reviewer #1:

Hsieh et al. perform micro-C, ChIP-seq, mNET-seq, and live single molecular imaging on cell lines with acutely degraded components of genome organization. The effect on 3D folding of depleting CTCF, RAD21 and WAPL has been previously studied by Hi-C and DNA imaging, yet the role of YY1 remains more mysterious. The primary versus secondary transcriptional consequences of disrupted 3D folding have remained challenging to disentangle. Therefore, the presented approaches could potentially shed light on the connection between genome organization and transcription. It was greatly appreciated that authors made their data available for review. However, at this time there are serious issues with cell line generation (WAPL and YY1 specifically), data processing, and interpretation of the micro-C and mNET-seq.

Concerns regarding micro-C analysis

1.1. Serious concern that peak/cot calling picks up accessible/visible regions rather than reflecting 3D folding at enhancers and promoters. Central conclusions of the study build on observing micro-C dots that persist after CTCF/RAD21/WAPL/YY1 degradation. However it is not demonstrated that the high micro-C signal between enhancers and promoters reflects 3D folding, as opposed to 1D biases, e.g. due to the high accessibility of these elements. The fact that E/Ps appear to give strong micro-C signal with their entire surrounding, sometimes tens of kbs away (e.g. figure 1f) is surprising. It remains unclear to what extent does this pattern reflect the sensitivity of micro-C to the technical accessibility of these regions (as opposed to reflected 3D organization) - see below.

Previously, we and others have independently demonstrated that Micro-C contacts are not biased toward highly accessible chromatin regions (Hsieh et al., 2020; Hua et al., 2021; Krietenstein et al., 2020). Notably, a recent Micro-C-based technique, Micro-Capture-C (MCC), that focuses on the interactions around promoter regions also did not find substantial bias toward DNaseI hypersensitive sites (Hua et al., 2021: Extended data Fig. 2). To further support this point, we provide key results below demonstrating that Micro-C data is not affected by chromatin accessibility levels. The E-P interactions captured by Micro-C thus represent *bona fide* proximal chromatin contacts.

1. When we analyzed Micro-C reads as single-end data (e.g., ChIP-seq, MNase-seq), we do not observe any apparent correlation with chromatin accessibility. In **Fig. 1** (left panel), we show that neither balanced (KR/ICE) nor coverage-normalized Micro-C data shows any correlation with ATAC-seq signal ($r^2 \ll 0.1$). Even the raw Micro-C reads have nearly no correlation with ATAC-seq signals ($r^2 = 0.15$). We note that all the Micro-C data in this manuscript were normalized either by matrix balancing or by coverage. Moreover, regions with low ATAC-seq signal (low accessibility) have slightly higher Micro-C signal than regions with high ATAC-seq signal (**Fig. 1**, right panel), arguing against Micro-C simply reflecting chromatin accessibility.

Fig. 1

2. To further demonstrate that Micro-C is not biased toward capturing highly accessible regions such as promoters, we performed a genome-wide pile-up analysis of Micro-C at base-pair resolution surrounding active promoter regions (**Fig. 2**). Coverage-normalized Micro-C shows nearly no bias toward the +1/-1 nucleosome region (more accessible) compared to the signal within the gene body (less accessible).

Fig. 2

3. In addition, genome-wide $P(s)$ analysis shows no evidence of significant differences in contact probability for different types of chromatin regions, including accessible promoters and enhancers, Polycomb-bound and heterochromatic domains, within a 500-kb range (**Fig. 3**).

Fig. 3

4. As we showed in our previous work (Hsieh et al., 2020), very deeply sequenced Hi-C data (Bonev et al., 2017: ~3.3B reads) can recapitulate some high-resolution E-P loops detected by Micro-C (**Fig. 4**). Although these dots in Hi-C are much fuzzier and their precise locations are often indiscernible, the example below clearly indicates that these dots appear in both Micro-C and Hi-C data. It is widely accepted in the 3D genome field that Hi-C has negligible bias toward highly accessible chromatin, thus the contact enrichment around high-resolution E-P interactions shared between two methods cannot be artifacts that are only present in Micro-C data. Furthermore, the example below shows that Micro-C can resolve what appears as a large blurry dot in Hi-C into multiple finer-scale dots and pinpoint the precise locations of the interactions (**Fig. 4**, zoomed-in box). Micro-C is thus a superior method to Hi-C in identifying fine-scale interactions.

Fig. 4

Together, our results and those of others strongly support that Micro-C does not exhibit any systematic bias toward highly accessible chromatin regions compared to other regions. From here on, we will refer back to these initial points every time a reviewer's concern assumes a Micro-C accessibility bias.

1.1.1 Issues with high-resolution dot calling.

The authors rely on published computational methods for calling TADs and loops/dots. However, these methods were not developed for or validated at the resolutions the authors use. For example, the highest resolutions considered were 1 kb and 10 kb in the mustache and chromosight publications for human data. The dots called here are highly biased for increased coverage relative to the genome-wide average at high resolutions. The plot below shows genome-wide average coverage at 400-bp resolution as a black line, and average coverage under the called dot/loop anchors as a function of map resolution (inferred as end-start) in blue:

Calling dots at very high resolution (< 1 kb) requires an approach that statistically models 1D coverage and verification that the dots called do not display a coverage bias. An example of such an approach is HiC-DC+ (PMID: 34099725), but even this approach would require extension and validation below 1 kb. If obtaining calls without coverage biases at sub-kb resolution is not feasible, 3D genomic analysis would benefit from being restricted to resolutions ≥ 2 kb, where dot calls show a less pronounced coverage bias, and existing computational tools have been validated. Crucially, if dot calls have a coverage bias, they cannot be interpreted as strictly reflecting 3D proximity.

Before using Mustache and Chromosight to call dots at high resolutions, we were well aware of this potential caveat. However, in our communications with the authors of Mustache (Dr. Ferhat Ay) and Chromosight (Dr. Romain Koszul), they do not see any problem with using their algorithms to identify high-resolution dots in Micro-C data. They emphasized that the maximal spatial resolution for dot calling largely depends on the sequencing depth. There should be no theoretical limitation to applying the algorithms to identify dots with our deepest sequenced Micro-C dataset. Dr. Ay did not apply Mustache to data with < 1 -kb resolution simply because such data was not available in the 4DN database at that time. The Koszul group actually applied Chromosight to ChIA-PET data at 500-bp resolution (Matthey-Doret et al., 2020: Fig. 3), and Chromosight has been routinely applied to yeast Micro-C data using 200- or 400-bp resolution. Importantly, Micro-C matrices are balanced using KR/ICE before loop detection. Thus, the potential coverage biases are largely corrected (as shown in Fig. 1-3 above). To test whether loop calling is affected by coverage in our data, we analyzed 1D coverage vs. the number of significant loops per locus. Fig. 5 below shows a similar trend across all resolutions, suggesting that 1D coverage does not explicitly affect dot calling in the high-resolution data.

Fig. 5

If the reviewer is still concerned, we are happy to use the loop calling results from data at ≥ 1 -kb resolution in the main text. Preliminary tests with this approach show no change in our results and conclusions. We also note that, except for the de novo loop calling in Fig. 1 and Fig. 3a, all other E-P quantifications presented throughout the manuscript were performed with data at 2- or 4-kb resolution. The latest Micro-C and Hi-C benchmark study by the Dekker lab recommends using Micro-C for loop detection (e.g., cis-regulatory elements interactions) due to its supreme sensitivity (Oksuz et al., 2021).

1.1.2 Issues for interpretation related to possible visibility biases of enhancers and promoters. Key contributions of the manuscript would not stand if these dots were called due to technical biases instead of actual 3D folding:

“Specifically, the strength of E-P and P-P interactions positively correlates with the level of gene expression, while cohesin loops show no such correlation” This is expected if the micro-C signal at sub-kilobase resolution is sensitive to E-P accessibility.

“Remarkably, acute depletion of CTCF and cohesin had a negligible impact on the E-P and P-P loops, with $\sim 80\%$ of E-P contacts (Fig. 3c) and 90% of P-P contacts (Fig. 3d) remaining unaltered.” Due to the coverage bias at dots called at higher resolutions, this likely reflects the visibility/accessibility of these regions, rather than their physical interaction in 3D. It sounds like the nature of the micro-C dots called between E-P (3D contacts vs. accessibility/technical biases) will remain uncertain until authors can profile a condition or treatment where the micro-C signal between E-P is affected genome-wide while accessibility remains unchanged.

Since we have demonstrated that the Micro-C signal is not biased toward accessible chromatin (please see the response to **1.1**) and we have provided strong evidence to dispute the problem of calling dots at high resolutions (please see the response to **1.1.1**), we assert that all the main conclusions in the manuscript remain unchanged. Importantly, we here emphasize that the E-P interactions were quantified with Micro-C data at 2- or 4-kb resolution, which is clearly not affected by possible visibility biases as the reviewer stated.

In addition, a recently published paper indicated that CTCF loss significantly changes $\sim 70\%$ of ATAC-seq peaks located around promoters, enhancers, and CTCF binding sites (Xu et al., 2021). If the changes in chromatin accessibility were to bias the Micro-C signal, we would expect massive changes of E-P interactions in Micro-C. However, we only find that $< 15\%$ of total E-P/P-P interactions are affected by CTCF depletion.

1.1.3 Orthogonal experimental validation. If technically possible, an orthogonal method such as DNA FISH (e.g. ORCA?) based assays could be deployed to establish the nature of the E-P interactions called at high-resolutions in micro-C and address concerns of point 1.

The average length of high-resolution E-P loops is ~ 20 kb in mammals (Gasperini et al., 2019: CRISPRi screening assay; Hsieh et al., 2020: Micro-C). However, the published ORCA method only resolved a ~ 130 -kb region with a 2-kb resolution at best (Mateo et al., 2019). Given the localization error derived from fluorescence-based imaging (Brandão et al., 2021), we doubt that DNA FISH or ORCA would provide conclusive results to validate the high-resolution Micro-C signal. In addition, perfecting a robust experimental system and analysis framework could take months or longer, while we believe this burdensome effort would at best add tangentially to the primary focus of this manuscript. We thus think validation using ORCA is not an ideal option.

Instead, we think that analyzing the Micro-C signal surrounding well-characterized and functionally-validated E-P contacts (e.g. by a CRISPR-based assay) (Moorthy et al., 2017) is more likely to give us a definitive conclusion. Indeed, in our previous mammalian Micro-C paper (Fig. 6) we confirmed that these E-P interactions can be clearly detected as dots or stripes in Micro-C, suggesting that Micro-C can capture the most genuine and functional 3D chromatin contacts.

Fig. 6

1.2.1 Mapping & filtering, and genotypes. There appear to be issues with either mapping or cell line differences from the reference assembly leading to off-diagonal artifacts. It would be helpful to demonstrate that such features do not impact downstream analyses.

After rechecking our Micro-C contact matrices, we noticed that the off-diagonal signals are rare and even more widespread in the standard 4D nucleome Hi-C data (see examples shown in the same region in **Fig. 7**). It therefore seems clear that these signals are not artifacts derived from our cell genotype or data mapping and filtering. We also note that their effects on our E-P loop analysis are minimal, if any. In fact, they are removed at early steps of our analysis pipeline because they lack promoter and/or enhancer marks (most are repeat elements) and span a much larger distance than

most E-P interactions.

Fig. 7

Micro-C

Hi-C (Bonev et al, 2017)

1.3 Concern that degron leakiness alters the perceived effect of degrading target proteins. Substantial variation exists between untreated (UT) conditions, which is not accounted for in the analysis or interpretation. For example, dots readily visible in YY1-UT appear absent in all other UT samples in the genomic region displayed in Figure 1.

Differential analyses relative to untagged reference cells, in addition to untreated cells, should be reported.

We will include a reproducibility and differential analysis (e.g., genome-wide matrix correlation and Jaccard index of loop anchors) comparing the pairwise correlation of all untreated Micro-C data. We will quantify the degree of the degron leakiness by Western blot and add nascent-RNA seq data to compare untagged cells to untreated, AID-tagged cell lines.

Concerns regarding transcription analysis by mNET-seq

To address comments 2.1, 2.1.1, 2.1.2, and 2.3 below, we will include a positive control (e.g., Pol II or Brd4 inhibition) and a time-course mNET-seq experiment for CTCF, WAPL, and YY1 depletion with a calibration control in the revised manuscript.

2.1 Lack of positive control to conclude about the absence of change. Authors make strong and thought-provoking claims about the lack of widespread changes detected by their mNET-seq approach. This raises the question: can authors detect widespread transcriptional changes with their approach? The conclusions as stated require reporting experiments where a greater number of changes are detected, as to set the basal expectation.

2.1.1 It is essential that authors establish the time resolution of mNET-seq in their hands. In other words how quickly can authors expect to detect pervasive transcription changes? Assaying the CTCF/RAD21/WAPL-AID cells at later time points could establish that transcriptional dysregulation can eventually be detected in these cells with the mNET-seq approach. Given that CTCF and WAPL depleted cells do not have cell cycle defects it may make most sense to assay one or both of these lines at later time points (e.g. 6h, 12h or 24h).

2.1.2 It is essential that authors establish expectations after short-term degradation of factors more directly involved in transcription. Without this comparison it is difficult to conclude much about the small number of differentially transcribed genes after only 3h. YY1 depletion dysregulated ~325 genes, which is the most compared to CTCF/RAD21/WAPL. It is difficult to estimate what is the upper expectation for the number of dysregulated genes after acute depletion of a transcription factor. This could easily be achieved using mESCs where transcription factors more directly linked to promoter or enhancer activity can be depleted and performing the same mNET-seq experiments after 3hrs (e.g. existing OCT4, NANOG or SOX2-degron cells, or using BRD inhibitors).

2.2 Authors must confirm that the dysregulated genes are changing because of CTCF/RAD21/WAPL rather than auxin treatment.

Authors state that “This suggests that while CTCF and cohesin are required for the transcriptional maintenance of only a small subset of genes, those genes tend to require the presence of both factors.” An alternative explanation is that auxin treatment in Tir-expressing cells affects those genes, irrespective of CTCF/RAD21 degradation (as reported by Liu et al. 2021 Nature Genetics PMID 33318687 by mRNA-seq at later timepoints). For example it is surprising that *cyp1b1*, a member of the cytochrome P450 family, is the highest upregulated gene across conditions. It would be important that authors perform mNET-seq in such their Tir1-only expressing cells treated with auxin, or rephrase their statements and interpretations to account for this possibility.

We are aware that the changes of the cytochrome P450 genes are likely due to the response to ethanol treatment (auxin is dissolved in ethanol). We will exclude these genes after confirming with the expression profile of parental cells.

2.3 In the absence of internal calibration, is it possible that authors are missing global down- or up-regulation of transcription (e.g. see Arnold et al. 2020 Mol Cell PMID 34324863 for how internal calibration of mNET-seq can alter interpretations)? Please comment on the ability of the current protocol to detect global homogeneous changes in the main text, assuming that performing calibrated mNET-seq experiments is out of the question.

2.4 The authors do not report the technical reproducibility of their mNET-seq and RNA seq data. Please include pairwise correlations between all replicates in all conditions (using either gene body or promoter for mNET-seq).

We will include the analysis.

2.5 Please include a suppl. table with the differentially transcribed genes detected in each mutant.

We will include the table.

Further confirmation that these new AID cells undergo near-complete degradation would be helpful.

3.1 Controls for all AID targets. Western blot is unlikely to pick up 5-10% remaining protein. Given that authors introduced fluorescent tags together with their degrons it would be helpful to show flow-cytometry data of untreated and 3h-treated cells together with untagged cells, separately for CTCF, RAD21, WAPL (they already present the YY1 data in Ext. Data Fig. 5a). It would also be especially helpful if authors could report whether long-term depletion leads to the cellular defects previously reported for these factors in mouse ES cells (CTCF: stop in proliferation after 4 days; RAD21: block in mitosis and cell death within 2 days ; WAPL: exit from pluripotency after 2-4 days - See Liu et al. 2020 Nature Genetics PMID 33318687, Nora et al. 2017 Cell PMID 28525758, Kubo et al. 2021 NSMB PMID 28525758 etc...). Can authors comment on what happens to YY1-depleted cells, do they survive?

We will add the flow cytometry analysis to monitor the degradation efficiency and the cell cycle profile for CTCF, RAD21, WAPL, and YY1 depletion across a time course.

3.2 Controls for WAPL deficiency. Authors assume near-complete depletion of WAPL based on Western blot analysis with the HA tag, brought together with the AID. However given the micro-C signal it seems unlikely that the Wapl depletion triggered maximum cohesin retention. Extended Data Figure 3D indicates that the micro-C signal at pre-existing dots remains unchanged after WAPL depletion. It looks like Wapl loss is much less complete than other studies in the field, e.g. Haarhuis et al. 2017 (Cell PMID 28475897) and Wutz et al. (PMID: 29217591) where extended grids of peaks emerge and the derivative of contact density shifts strongly rightward. This makes the author's reports of little transcriptional change somewhat inconclusive, as it may simply be due to technical issues of their cell line generation. It would be helpful if authors could characterize the effect of their WAPL degradation: How much more cohesin is there on chromatin after 3hr of WAPL depletion in these cells? It would be informative if authors could perform calibrated ChIP-seq for core cohesin subunits after 3hrs of WAPL degradation (compared to either untreated and the WAPL untagged cells with all the other tags also present in the WAPL-AID line [CTCF-halo, RAD21-SNAP]). Alternatively authors could perform careful quantifications of chromatin-bound core cohesin subunits by western-blot, but this might be difficult as one would only expect a maximum 2-fold increase of cohesin, given that around 50% of cohesin is already bound before WAPL depletion. The authors should also update their reported conclusions in light of the potential technical confounders.

We believe the reviewer may be misinterpreting some of the results reported in those two WAPL perturbation papers. Haarhuis et al. used the HAP1 cells with WAPL knockout, and Wutz et al. used RNAi to knock down WAPL in HeLa or MEFs for 3 days. Their experimental conditions and the questions of interest are very different from ours. First, since the residence time of cohesin is longer than 20 min in mESCs, we do not necessarily expect that cohesin can reach the maximal retention after only 3 hrs of WAPL depletion (and we did not claim it in the manuscript). Our primary focus is the immediate short-term effect of WAPL depletion on the 3D genome and transcription, rather than studying the endpoint result after cells undergo unknown processes. Second, we do not expect WAPL depletion to drastically increase the contact intensity of pre-existing loops. As shown in Haarhuis et al. and the latest pre-print in Liu et al. by the de Wit lab, WAPL depletion does not substantially increase the contact intensity of the pre-existing dots (**Fig. 8**, top panels). By comparing the $P(s)$ curve from Haarhuis et al. and differential $P(s)$ curve (6 hr, green line) from Liu et al. with our Micro-C data, we find a similarly right-shifted $P(s)$ curve (**Fig. 8**). Critically, aggregate peak analysis across multiple ranges of genomic distance shows substantially stronger extended cohesin loops (> 250 kb and beyond) in WAPL depletion (**Fig. 8**, bottom panel). Thus, we provide strong evidence that a 3-hr WAPL depletion is sufficient to increase cohesin retention on chromatin and extend the grids of cohesin loops. However, these additional long-range loops have no immediate effect on transcription.

We are willing to add quantitative biochemical assays and flow cytometry analysis to confirm the fraction of cohesin on chromatin 3 hrs after WAPL depletion, even though the reviewer's concerns here are based mainly on some misguided references to previous studies.

Fig. 8

Micro-C data in this study

3.3 Issues with YY1 degradation.

The majority of YY1 peaks were diminished, but only ~50% of peaks (n = 15075) were called significantly changed by differential peak analysis. Does that mean the depletion is not complete? How do we interpret the seemingly complete depletion base of Western and flow cytometry then - are these assays not sensitive enough? Or is it because the YY1 ChIP-seq was not calibrated? Could it be because authors use the miniIAA7 tag with osTir1, instead of atAFB2 as recommended by citation 64, Li et al. Nat Methods PMID 31451765?

We also confirmed by confocal imaging that YY1 degradation is near complete. We now plan to repeat YY1 ChIP-seq with different antibodies to confirm that the residual signal after acute degradation of YY1 is not due to antibody detection of off-target proteins. We did use the parental cell line with stably expressing atAFB2 for YY1 degradation.

Conflating time elapsed with primary versus secondary effects on transcription. Author conclusions assume that the larger transcriptional consequences reported by others in similar cell lines likely reflect secondary consequences from a small subset of affected genes. While this could very well be true, it is also possible that direct primary transcriptional consequences of RAD21/CTCF/WAPL/YY1 depletion are slow-manifesting. Some evidence suggests that some of the slow-manifesting transcriptional changes picked up by previous studies could be consequences of direct cis-regulatory impacts: for example, over 80% of the genes down-regulated at mRNA level after one day of CTCF degradation by Nora et al. 2017 Cell PMID 28525758 had CTCF binding in their promoter. While it is challenging to disentangle slow primary defects from rapid secondary ones with the approaches used by the authors, the possibility that some primary consequences may be slow-manifesting should be discussed off the bat in the text, as it is an obvious expectation, rather than brought up only in the discussion under a “time-buffering” model.

The reviewer cited Nora et al. to describe the “slow-manifesting” transcriptional effect 1 day after CTCF depletion. However, the biological definition of a “slow-manifesting” effect is unclear to us. First, promoter bound CTCF can act as a transcription factor or a mitotic bookmarking factor. In this case the definition of “slow-manifesting” is in line with our “time-buffering” model. Second, this study used bulk RNA-seq analysis to interpret the transcriptional effects of CTCF depletion. This means their observations are comprised of mixed results of 1) 3D genome reorganization upon CTCF depletion; 2) secondary transcriptional effects; 3) total pool of mRNA before and after CTCF depletion; and 4) secondary effects of post-transcriptional RNA processing and buffering. Thus, we do not plan to discuss the “slow-manifesting” transcription in our manuscript without a definitive clarification and example.

5. Please include annotated .gb files for all targeting constructs mentioned in Table S1, rather than stating “sequence available upon request”. Please either make the vectors available on addgene or by simply add annotated .gb sequence files as supplementary data.

We will provide the files.

Minor points:

We will correct all the minor points in the revised manuscript.

Captions are not always complete. e.g. it is not clear what microC data are plotted in Fig. 1: untreated YY1? something else?

Authors mention identifying 75,000 dots in the text but the 4-category breakdown in figure 1a-b only adds up to 42,000 (assuming the random and promoter-random categories are shuffled controls?). Please clarify and include the missing “other” category in the figure caption.

A heatmap showing correlations between micro-C replicates, or a table or pairwise correlations, should be added to the supplement.

It would be helpful if authors could refrain from using interchangeably “dots” and “loops”. “Dots” seems the most accurate description of the peak-calling result, without making assumptions about the underlying conformation which can only be resolved through modeling or imaging.

Is there anything special about the genes immediately dysregulated after CTCF degradation? E.g. they have CTCF binding in their promoter? Are they sitting next to any relevant micro-C feature (stripe, dot, boundary...)? It would be helpful if authors confirmed their data showed these aspects of previous studies, or discuss why it might not.

“These studies have shown that while CTCF and cohesin play only a minor role in compartmentalization” While this is true for CTCF depletion, it was reported by many group that cohesin antagonizes compartmentalization, so that compartmentalization patterns are greatly affected in RAD21, NIPBL or WAPL depletion (e.g. Rao et al. 2017, Schwarzer et al. 2017, Haarhuis et al. 2017). Authors may want to reword to reflect this.

The following sentence makes it sound like measuring 3D interactions between E-P by micro-C at sub-kilobase resolution in mammalian genomes is already established. “We recently reported that Micro-C can effectively resolve ultra-fine 3D genome folding at nucleosome resolution, including E-P and P-P interactions, thus overcoming this limitation, 24–27.” Ref 27 (Hansen et al. 2017) is cited to support that micro-C can effectively detect E-P and P-P interactions at nucleosome resolution. The sequencing depth in that study did not allow for nucleosome resolution. Indeed Hansen et al. 2017 reported around 14,000 loops vs 75,000 here. Similarly Ref. 24-25 are previous publications from the present author this time in *S. cerevisiae*, where there was no mention of E-P or P-P dots at nucleosome resolution. Please remove these citations here or rephrase the sentence to distinguish technology development (micro-C) from biological interpretation (E-P interactions).

“We note that Micro- C has much higher sensitivity for detecting E-P and P-P contacts compared to Hi-C, establishing Micro-C as a more suitable assay to study genome organization relevant to transcription regulation genome-wide in an unbiased manner.” It is not fair for authors to state micro-C is unbiased, it simply has milder biases than Hi-C, so this should be reworded.

“Newly transcribed RNAs generally have a higher correlation with E-P contacts than with compartments and TADs (Extended Data Fig. 1f)” what does correlation with TADs mean? Which score is used? Strength of insulation? Averaged over the two boundaries? Please clarify.

In paragraph “Acute depletion of CTCF, cohesin, and WAPL perturbs structural loops,” please include a sentence in the main text explaining that these experiments were done without internal spike-in calibration, so authors cannot conclude about possible differences in overall binding (it is likely that the loss of CTCF/RAD21 signal is under-estimated without calibration). Without calibration, authors should temper the statement “While cohesin peaks are generally lost upon CTCF degradation, CTCF binding is unaffected by altering the level of cohesin on chromosomes” - it is possible that CTCF levels are overall higher or lower. “RAD21 or WAPL depletion caused only a 10 to 20% reduction in CTCF peaks” - please clarify “number of peaks” (current sentence could be interpreted as referring to signal intensity). Maybe worth rephrasing to also account for that in ref. 11 (Rao et al. Cell 2017) the CTCF ChIP-seq after RAD21 depletion was not calibrated either.

“Nevertheless, upon rapid induction (1 hour) of erythroid differentiation, YY1 triggers little or no change in H3K27ac and H3K27ac-anchored HiChIP interactions” - phrasing is confusing. Maybe authors could instead write that “YY1 redistribution triggers little or no change...”, as this was not an inducible-degradation approach.

YY1 was suggested to play a more prominent role in differentiated cells compared to mouse ES cells. It would be fair to acknowledge this when citing ref. 61. Beagan et al. Genome Research PMID 28536180.

Please rephrase the following sentence to reflect that the authors only investigated the situation after 3hrs. “In summary, we find that, while CTCF, cohesin, and WAPL may regulate some gene expression, their acute depletion affects the transcription of only a handful of genes in mESCs, which largely encode pluripotency and differentiation factors.”

Tir1 transgenes are missing in cartoons of Extended Fig. 2 for all the cell lines.

Reviewer #2:

In their manuscript, Enhancer-promoter interactions and transcription are maintained upon acute loss of CTCF, cohesin, WAPL, and YY1, Hsieh et al. explore the role of architectural proteins on E-P interactions and gene expression by a combination of micro-C and nascent transcriptomics in mESCs. Their findings are as follows:

- 1- Using micro-C in combination with a new loop caller, Mustache, they identify many new looping interactions between enhancer and promoters.
- 2- Acute (3hr) loss of CTCF, Rad21, WAPL, and YY1 does not perturb the frequency, location, or strength of most newly identified E-P interactions.
- 3- Acute (3hr) loss of CTCF, Rad21, WAPL, and YY1 does not affect global gene expression.
- 4- Cohesin loss alters YY1 binding dynamics.

The authors propose a time-buffering model, which I quite like and do think that it reconciles their and others' largely negative results related to acute loss of architectural proteins and gene regulation. However, I do have some concerns outlined below:

The main criticism of this paper pertains to the validity of the newly identified loops, which is the basis of their largely negative results and major conclusions. The authors utilize a new genome-wide, 3C-based method known as Micro-C along with a very new loop caller they recently developed (Mustache) and claim to identify new looping interactions not previously identified by Hi-C. Surprisingly, the loop caller alone adds a significant amount of new loops even from the authors' previous Micro-C analyses. While this makes sense in theory given the higher resolution MNase digestion would generate over convention, some of the features associated with these new loops does raise a few red flags.

That is, the authors find that these new loops correlate with transcription, frequently traverse TAD boundaries, and are robust to acute depletion of CTCF, cohesin, WAPL, and YY1. While these interactions may indeed be a novel and important discovery, the authors' conclusions are almost entirely based on these negative results. Importantly, they do not completely rule out the possibility that these new interactions are instead an artifact of the approach. For instance, transcriptional sites would of course also be associated with more open chromatin and higher MNase accessibility in general. It was not clear from the methods how the authors normalize their data to MNase digestion biases similar to how typical Hi-C data is normalized to the density and location of site-specific restriction digestion. It would also be good to know if frequent exceptions to this correlation exist wherein Micro-C is detecting new loops not associated with high levels of open chromatin or transcription. If so, do these also share the same features as mentioned above. Are the new loops dependent on transcription as could be determined via polIII inhibition or mutational analysis? Surely, these resources are available for mESCs?

1. Please see our response to the reviewer #1's comments 1.1 and 1.1.1 that fully address the concern of Micro-C analysis. We also provide a very detailed description of our Micro-C analysis and normalization in the methods section. In addition, in our previous study, we demonstrated that Micro-C detects ~5-fold more loops than Hi-C by using a well-established loop caller, hiccups. We confirmed that the results remain unchanged regardless of which loop callers we used or excluding these newly-identified loops. More importantly, we did not use the de novo loops called by Mustache or Chromosight for the analysis of TAD boundary crossing. Instead, we quantified the paired E-P interactions (by chromHMM states) with the Micro-C data at 2-kb or 4-kb resolution.
2. We will provide some examples to show Micro-C loops not associated with highly accessible regions.
3. For a detailed analysis of Pol II inhibition on E-P interactions, please see our previous study in Hsieh et al. 2020 and **Fig. 9** below. In brief, treatment with Pol II inhibitors triptolide or flavopiridol moderately reduces E-P or P-P interactions and substantially attenuates chromatin stripes.

Fig. 9

But most concerning is that they find that many (29-30%) of these loops traverse a TAD boundary. While it is true that boundaries have been found to be variable and TAD insulation is not strict, it is hard to imagine that stable or frequent E-P interactions traversing a boundary would represent a functional interaction. So far, a large body of literature point to the fact that cognate promoters and enhancers tend to be within the same TAD. For instance, this reviewer is not aware of any developmental gene whose functionally validated enhancer has been mapped outside of the same TAD and if so this is an extremely rare event. And recent estimates from eQTL studies suggest that >95% of variants are also located within the same TAD as the associated gene (Delaneau et al., 2019). Similar conclusions have been made from perturbation studies in mESCs (Sun et al., 2019). How do the authors reconcile their results with these findings?

First, a recent CRISPRi-based enhancer screening reported that ~30% of E-P interactions do not fall into the same TAD (Gasperini et al., 2019). Second, after checking the promoter Capture Hi-C result in Sun et al. 2019, we highlight that the authors clearly indicated that ~24% of promoter-centered interactions are not constrained within the same TAD, which aligns with our findings. Third, emerging evidence by super-resolution imaging indicates that TADs are dynamic chromatin structures that constantly form and dissolve (Bintu et al., 2018). Cohesin can traverse across boundaries frequently and mediate domain intermingling (Luppino et al., 2020; Szabo et al., 2020). We think enhancers' communication with their cognate genes in another domain is not a rare event. Finally, since the eQTL analysis in Delaneau et al., is based on the correlation of a set of ChIP-seq data for histone modifications. Thus, the cis-regulatory domains (CRDs) identified in this study most likely represent local chromatin domains segregated by chromatin states (e.g., active or inactive regions) rather than E-P loops. It is also unclear whether this computational structure can fully reflect *in vivo* 3D genome structures, and even more uncertain is the comparison to other computationally defined structures (such as TADs) that are susceptible to various computational parameters. A recent study indicates that analysis of chromatin loops outperforms eQTL in explaining neurological GWAS results, suggesting there may be much ampler information in high-resolution 3D genome maps than in eQTL analyses (Lu et al., 2020). Together, our findings of cross-TAD interactions are largely consistent with other studies. We are confident reporting numbers close to those that are functionally validated.

Lastly, many studies have shown, from sequencing and imaging-based methods, that loss of cohesin or WAPL changes chromatin compaction both within and between TADs. Even indirectly, this would be expected to alter promoter-enhancer proximity. And yet, the authors show no change in number or strength of their newly identified interactions in the absence of several architectural proteins.

Given the above, it seems extremely important that the authors validate their new looping interactions through orthogonal methodology such as 3C, Capture-C, or 5C that each would result in comparable resolutions to that of Micro-C. In addition, if the authors were to bin their data to artificially reduce their resolution to that comparable to Hi-C, would these interactions disappear upon reanalysis? At the very least, this would show that

they are dependent on the unprecedented high resolution achieved by Micro-C.

1. We will compare Micro-C data with promoter HiChIP and Capture Hi-C if available. However, we note that recent studies using HiChIP (Kubo et al., 2021) and Capture Hi-C (Thiecke et al., 2020) have reported very consistent results with ours.
2. Please see our response to the reviewer #1's 1.1. These loops are uniquely detected in Micro-C.

Other comments:

1. It has recently become apparent that AID tagged versions of proteins, including Rad21, can be 'leaky' and show reduced activity or levels compared to their wildtype versions. How does the expression profile of their tagged cell lines compare to that of parental mESCs? Is it possible the small changes upon auxin treatment are due to an already perturbed system prior to treatment?

We will quantify the degree of the degron leakiness by Western blot and flow cytometry analysis and add nascent RNA-seq analysis of untagged cells.

2. The conclusion that cohesin might act to recruit transcription factors in general based solely on YY1 behavior seems like a bit of a leap. The authors might want to tone down this conclusion.

We are willing to tone it down and perhaps include one or two more transcription factors to strengthen this model.

Reviewer #3:

CTCF, cohesin complex, and YY1 have been proposed to be key players shaping the chromatin architecture in mammalian cells, with CTCF and the cohesin complex involved in mediating long-range chromatin loops between convergent CTCF binding sites through a loop extrusion process, while YY1 shown to mediate enhancer-promoter contacts by virtue of its DNA binding to both classes of elements and its dimerization potential. Additionally, cohesin has also been suggested to mediate enhancer-promoter interactions due to its association with mediator complex and loading by Nipbl protein, which binds to both promoters and enhancers. While abundant literature supports a role for CTCF and cohesin in chromatin organization especially in defining topologically associating domains (TADs) and long-distance chromatin loops, their function in chromatin contacts between enhancers and promoters is less clear and has been the subject of intense debate in recent years. Also under intense debate is whether chromatin interactions between enhancers and promoters actively contribute to enhancer dependent gene activation. Recent studies using auxin inducible degrons to acutely deplete CTCF or cohesin subunits showed that CTCF and cohesin are dispensable for transcription of most genes in a variety of cell systems tested. By contrast, acute depletion of YY1 has been shown to lead to dramatic changes in gene expression in mouse ES cells. These previous studies raised questions about what exactly is the role that CTCF and cohesin play in enhancer-promoter communication and gene activation, and whether different proteins (such as YY1) are involved enhancer-promoter contacts and enhancer dependent transcription from target genes.

The manuscript by Hsieh et al. attempts to address these two questions and to provide clarity on the role of CTCF, cohesin and YY1 in enhancer-promoter contacts and the role of enhancer-promoter loops in gene activation. Like previous studies, the authors used the auxin inducible degron system to deplete CTCF, Rad21 subunit of cohesin, cohesin unloader protein WAPL and YY1 protein, in a mouse embryonic stem cell line. But in contrast to the previous work, the experimental design in the current report involved a shorter protein depletion time (3 hours vs. 24hr or longer), which could avoid the secondary effects during long-term protein depletion. Additionally, the authors used Micro-C, a variant of Hi-C technique, to achieve higher resolution (a few hundred base pairs versus 5- or 10-kb resolution) and sensitivity of loop detection especially between enhancers and promoters. Furthermore, the authors employed mNET-seq to measure changes in nascent transcript levels genome-wide after protein depletion. Their main findings are: (1) depletion of these proteins did not cause significant changes in enhancer-promoter interactions, even for YY1; and (2) depletion of these proteins resulted in minimal changes in gene expression. The authors also used live cell single particle imaging techniques to study the dynamics of YY1 binding to chromatin upon depletion of cohesin, and found an unexpected but modest role for cohesin to retain YY1 on chromatin. Based on these observations, the author concluded that “neither transcriptional condensates, CTCF, cohesin, WAPL, nor YY1 are generally required for the short-term maintenance of E-P interactions and the subsequent expression of most genes after acute depletion and loss of function” (Line 494). To reconcile with previous genetic evidence implicating CTCF and cohesin’s role in regulating promoter-enhancer interactions and transcription at some genes, the authors further proposed a “time-buffering model”, “where architectural proteins generally contribute to the establishment, but not the short-term maintenance, of E-P interactions and gene expression” (Line 548).

General assessment:

Overall, the current work reports several very interesting observations related to the role of CTCF, cohesin and YY1 in enhancer promoter interactions and transcription, with some in direct contradiction with previous studies, and others confirmatory. An important contribution of this work lies in the use of a well-controlled experimental system and high precision technical approaches. The genetically engineered cell lines to enable acute depletion of CTCF, YY1, and cohesin complex in cells are valuable tools to the field. The micro-C and mNET-seq datasets along with other genomic datasets in this experimental model system also provide a valuable resource for the community to investigate the role of these proteins in chromatin organization and gene regulation. Surprisingly, the authors found that cohesin depletion did not disrupt enhancer-promoter contacts in general, and that YY1 loss had no significant impact on both enhancer promoter contacts and gene expression, which is in direct contradiction to Weintraub et al. (Cell 2017, ref #62). These new results are interesting and certainly raise new questions.

On the other hand, the experimental evidence provided in the current work does not fully support the sweeping conclusion that CTCF and cohesin play no role in the maintenance of enhancer promoter interactions (as the title implies), nor the statement about transcriptional condensates (see specific comments below for details). Further, some of the findings described in the current work are not new. For example, the observations that CTCF was largely dispensable for enhancer-promoter interactions and general transcription are consistent with previous studies (for example, Nora et al. Cell 2017; Kubo et al. NSMB 2021, which the authors cited as ref #10 and #37, respectively). The observation that cohesin loss resulted in minimal changes in gene expression was also reported before (for example, Rao et al. Cell 2017, ref #11 in the manuscript). Therefore, the degree by which this work advances our understanding of the maintenance of enhancer-promoter interactions and their function is currently unclear.

In brief, we think our study will significantly contribute multiple new discoveries to the field:

1. We used Micro-C to identify many more details of E-P interactions that were technically impossible in previous studies. Many of these previously undetectable loops are functionally validated by CRISPR-based studies (Gasperini et al., 2019; Moorthy et al., 2017).
2. We resolved the immediate short-term effect of CTCF, cohesin, WAPL, and YY1 on E-P interactions and nascent transcription. All the prior studies focused on later time points that may suffer from secondary effects.
3. These new findings allow us to establish a time buffering model that reconciles our and previous studies and explain a possible mechanism of 3D genome and transcriptional regulation.
4. We comprehensively dissect YY1 dynamics by single-particle imaging and uncover a putative role of cohesin in regulating transcription factor binding distinct from its known function in genome folding.

Main comments:

1. The experimental evidence presented does not fully support the authors' assertion that enhancer-promoter (E-P) interactions are unaffected after depletion of CTCF or cohesin. A closer inspection of Figure 3C suggests that E-P interactions were generally weakened after depletion of Rad21 and to a lesser degree CTCF - the center of the data cloud representing E-P interactions is below the diagonal line, and this trend is not obvious for P-P interactions. Therefore, one could say that there is a general and weak effect for cohesin loss on E-P interactions, consistent with residual levels of cohesin retained on chromatin after short term depletion.

We will inspect this result carefully and perhaps add a note of caution in our interpretation.

2. Similarly, the assessment of YY1's effect on chromatin interactions was also problematic. The authors in Figure 4i examined changes of chromatin interactions on cohesin mediated loops (strong loops), and found little change. But what about E-P interactions? It would be important for the authors to provide a similar plot for the YY1 depleted cells as in Figure 4C for CTCF and cohesin depletion. The authors did report ~800 loops changed after YY1 depletion (Line 320), but Figure 4e indicated only 436 as "down" loops. So it is unclear how the authors reached the conclusion that loss of YY1 did not affect chromatin interactions between promoters and enhancers, especially when only 50% of YY1 binding sites showed significantly reduced binding after YY1 depletion (Line 308).

1. We will add the same E-P analysis for YY1.
2. We note that we only analyzed loops with significant change in YY1 binding. We were also baffled by the discrepancy between the ChIP-seq results and the protein analysis, flow cytometry and imaging experiments, all showing near-complete loss of YY1 3-hrs upon degradation, and. We now plan to repeat YY1 ChIP-seq with different antibodies to confirm that the residual signal after acute degradation of YY1 is not due to antibody detection of off-target proteins.

3. A major feature of the experimental design is the short duration of protein depletion (3 hours). This is also a potential weakness, as it is apparent that 3-hour IAA treatment did not fully deplete the proteins. Residual ChIP-

seq signal for CTCF and cohesin were readily detectable on chromatin at CTCF peaks (Ext Fig 2f) and E-P & P-P loop anchors (Ext Fig 4e). Same can be said for YY1 (Figure 4b, Ext Fig 5C). It is therefore hard to rule out the possibility that residual proteins bound to chromatin might contribute to maintenance of chromatin interactions in the short-term depletion.

1. We will add biochemical assays and flow cytometry analysis to confirm the fraction of proteins on chromatin after 3 hrs of depletion.
2. Since all CTCF and cohesin loops are nearly completely abolished 3 hrs after depletion, it is unclear what the basis is to assume that the residual proteins will specifically maintain E-P interactions but not CTCF/cohesin loops.
3. In addition, we have previously measured the average number of CTCF, RAD21, and YY1 molecules in a single cell (Cattoglio et al., 2019; Holzmann et al., 2019). We estimate that, if our degradation can reach ~95% depletion after 3 hrs, there are roughly only ~300 CTCF, ~200 RAD21, and ~700 YY1 proteins remaining bound on chromatin. Thus, we do not think that hundreds of residual architectural proteins are sufficient to maintain >25,000 E-P loops and transcription. We thus expect the effect of residual proteins on our results is minimal.

4. The authors concluded that “transcriptional condensates” are not generally required for “short-term maintenance of E-P interactions” (Line 493), by drawing on previous literature (Ref #57, 58 and 88), without providing any new evidence in the current work. This statement therefore needs to be revised to better reflect this fact.

We will revise this part.

5. It is interesting that cohesin loss results in reduced population of immobile and presumably chromatin bound YY1, but this evidence alone is insufficient to lead to the model that cohesin facilitates target search by transcription factors.

We are willing to tone it down and perhaps include one or two more transcription factors to strengthen this model.

Minor comments:

We will address all the minor comments.

1. With multiple loop callers, authors show that micro-C can identify more fine range E-P and P-P interactions compared to Hi-C. What percent of the E-P, P-P loops called by previous Hi-C data are also detected by Micro-C and vice versa? What is the overlap? What about with HiChIP /Plac-Seq data? This analysis is important to allow the readers to assess the specificity and sensitivity of the data.

2. Titles for the scatterplots missing (e.g., Ext Fig 4c)

3. Extended Fig 4d. A subset of downregulated loop anchors in WAPL depletion cells seem to show more bias towards enhancer and promoter regions. Do these regions show any differences in Rad21 binding?

4. Upon depletion of YY1 itself, there is a modest decrease of Rad21 binding. Which Rad21-bound regions show reduction upon YY1 depletion, what are their characteristics? Do they correspond to loop anchors, or show any change in loop strength?

Reviewer #4:

In this paper, Hsieh et al. leverage high resolution Micro-C and nascent RNA (mNET-seq) data to understand the roles played by CTCF, cohesin, and YY1 in the maintenance of enhancer-promoter interactions. The authors use mESC cell lines in which CTCF, WAPL, cohesin, and YY1 are tagged for rapid (3h) degradation. The authors argue that the marginal effect of CTCF and cohesin on overall gene expression level, as observed previously in the literature, is explained by cis-regulatory loops between enhancers and promoters being relatively insensitive to the depletion of these architectural proteins. The authors then show through similar experiments that the depletion of YY1, previously suggested as an important factor for enhancer-promoter loops, also had only marginal effects on enhancer-promoter loops and transcription. Finally, using a combination of confocal microscopy and single molecule imaging, the authors characterize the dynamics of YY1 binding to chromatin and show that the depletion of RAD21, but not CTCF significantly reduces YY1-chromatin binding by increasing its searching time. This paper examines an important aspect of the relationship between chromatin folding and transcription. It is a well written paper; experiments are well justified and expertly executed. The results are timely. However, while the data is comprehensive and solid, I do have some major concerns about data analysis and interpretation that need to be addressed prior to publication.

Major comments:

(1)* The authors argue that 80-90% of loop interactions between enhancers and promoters (or promoters and promoters) are ‘unaltered’ by depletion of cohesin or CTCF. I am not convinced that the author’s data supports this conclusion. Setting aside the ~20% of enhancer-promoter interactions that the author’s analysis turns up as significantly reduced, there does appear to be a small reduction in contact frequency more broadly. Carefully examining the scatterplots in Fig. 3C shows that the bulk of enhancer promoter contacts are comfortably below the 0, 1 line, indicating lower contact frequency in AID than control cells. If this shift is not indeed explained by normalization issues (see points below), I think the paper would improve if the authors provided a more nuanced presentation of these results, acknowledging the shift that is clearly present in their data.

We will inspect this result carefully and perhaps add a note of caution in our interpretation.

(2)* All of the Micro-C analysis in this manuscript compares contact frequency between different conditions with very large, global differences in contact frequency. As with any other sequencing assay, contacts that are lost from (say) CTCF-CTCF loop anchors or TADs upon CTCF or Rad21 depletion will be redistributed to other loci. In 1D data (e.g., mNET-seq/ ChIP-seq/ etc.), spike-in controls can be used to normalize between different samples in absolute terms, allowing the discovery of truly global changes. While I recognize there is no easy analog of a spike-in for a Micro-C experiment, I would like the authors to discuss the importance of normalization, and the ways in which normalization issues could impact the author’s interpretations.

Please see our response to reviewer #1’s comments 1.1 and 2.1 that fully address Micro-C and mNET-seq analysis. We also provide a very detailed description of our Micro-C analysis and normalization in the methods section and will discuss it further in the revised manuscript. In addition, we will include a new set of mNET-seq with a calibration control in the revised manuscript.

(3)* MNase, depending on the digestion conditions, does not cut uniformly across the entire genome, but rather it has specific biases for accessible regions. Any change in the distribution of MNase cut frequency near promoter/ enhancer regions between wild-type and CTCF, YY1, or Rad21 depletion could alter the contact frequency observed in contact maps. The authors should evaluate whether there are differences in MNase cutting efficiency between different conditions by examining the 1D signal in their Micro-C libraries.

Please see our response to reviewer #1’s comments 1.1.

(4)* It generally is not clear whether the authors measure transcription after removing the pause peak, focusing on expression in the gene body (which most directly correlates with mRNA), or whether paused Pol II is included. As the different stages of transcription initiation and pause might have different relationships with contact frequency, it is crucial to distinguish clearly between them in their presentation.

We only include the mNET-seq signal at the gene body. We will add a detailed description in the text.

(5)* Data showing that RAD21 serves as a “platform” for YY1, or possibly other transcription factors, is extremely weak. I am convinced by the data showing that it increases search times (although I will defer to an expert on the rigour of the microscopy experiments). How do the authors feel about an alternative model in which the act of loop extrusion acts to stir the nucleoplasm, resulting in higher diffusion and faster search times? There are likely other indirect explanations as well, and I would encourage the authors to consider these mechanisms before jumping right to IDR-IDR interactions. Related to this point - unless the authors can directly show that protein-protein interactions are necessary for the effect of RAD21 depletion on YY1 search time, I would encourage them to drop the use of the word “platform” and other terms that imply a direct interaction.

We will discuss other alternative models and modify our discussion about the TF binding platform.

Minor comments:

We will address all the minor comments.

* In figure 1d, the axes should be better described. Additionally, in P-P contacts, clarify whether the data points represent an average of the two promoters.

* Throughout the manuscript, add axes description to the APA heatmaps explaining, for example, which of the rows\columns represent promoters\enhancers.

* Please add transcriptional orientation (i.e. what is the direction of the transcription unit) to the APA heatmaps.

* A clearer explanation on how anchors were chosen and defined as promoters and enhancers would also help readers interpret the data. What dataset was used to define TSSs or whether it was accomplished by the MNase data should be clarified.

* Please use the acronym polyadenylation cleavage site (PAS), rather than TES, in figure 1C. I assume TES stands for transcription end site, but presumably the point used is the end of a gene annotation. Transcription continues past the end of the PAS, as is clearly shown in the authors' plots.

* Some interesting differences are shown between P-P and E-P pairs (Fig. 1D and 3H) but are ignored in the text. Are these associated with the different levels in initiation and pausing?

* Figure 3b shows that loops from up/ down/ unchanged groups are enriched in functional elements involved in transcriptional regulation. Taken along with the pattern shown in 3a, this further suggest that the effect of CTCF and RAD21 depletions on cis-regulatory elements looping is underestimated by the authors.

Reference

- Bintu, B., Mateo, L.J., Su, J.-H., Sinnott-Armstrong, N.A., Parker, M., Kinrot, S., Yamaya, K., Boettiger, A.N., and Zhuang, X. (2018). Super-resolution chromatin tracing reveals domains and cooperative interactions in single cells. *Science* 362, eaau1783.
- Bonev, B., Cohen, N.M., Szabo, Q., Fritsch, L., Papadopoulos, G.L., Lubling, Y., Xu, X., Lv, X., Hugnot, J.-P., Tanay, A., et al. (2017). Multiscale 3D Genome Rewiring during Mouse Neural Development. *Cell* 171, 557-572.e24.
- Brandão, H.B., Gabriele, M., and Hansen, A.S. (2021). Tracking and interpreting long-range chromatin interactions with super-resolution live-cell imaging. *Curr Opin Cell Biol* 70, 18–26.
- Cattoglio, C., Pustova, I., Walther, N., Ho, J.J., Hantsche-Grininger, M., Inouye, C.J., Hossain, M.J., Dailey, G.M., Ellenberg, J., Darzacq, X., et al. (2019). Determining cellular CTCF and cohesin abundances to constrain 3D genome models. *Elife* 8, e40164.
- Gasperini, M., Hill, A.J., McFaline-Figueroa, J.L., Martin, B., Kim, S., Zhang, M.D., Jackson, D., Leith, A., Schreiber, J., Noble, W.S., et al. (2019). A Genome-wide Framework for Mapping Gene Regulation via Cellular Genetic Screens. *Cell* 176, 377-390.e19.
- Holzmann, J., Politi, A.Z., Nagasaka, K., Hantsche-Grininger, M., Walther, N., Koch, B., Fuchs, J., Dürnberger, G., Tang, W., Ladurner, R., et al. (2019). Absolute quantification of cohesin, CTCF and their regulators in human cells. *Elife* 8, e46269.
- Hsieh, T.-H.S., Cattoglio, C., Slobodyanyuk, E., Hansen, A.S., Rando, O.J., Tjian, R., and Darzacq, X. (2020). Resolving the 3D Landscape of Transcription-Linked Mammalian Chromatin Folding. *Mol Cell* 78, 539-553.e8.
- Hua, P., Badat, M., Hanssen, L.L.P., Hentges, L.D., Crump, N., Downes, D.J., Jeziorska, D.M., Oudelaar, A.M., Schwessinger, R., Taylor, S., et al. (2021). Defining genome architecture at base-pair resolution. *Nature* 1–5.
- Krietenstein, N., Abraham, S., Venev, S.V., Abdennur, N., Gibcus, J., Hsieh, T.-H.S., Parsi, K.M., Yang, L., Maehr, R., Mirny, L.A., et al. (2020). Ultrastructural Details of Mammalian Chromosome Architecture. *Mol Cell* 78, 554-565.e7.
- Kubo, N., Ishii, H., Xiong, X., Bianco, S., Meitinger, F., Hu, R., Hocker, J.D., Conte, M., Gorkin, D., Yu, M., et al. (2021). Promoter-proximal CTCF binding promotes distal enhancer-dependent gene activation. *Nat Struct Mol Biol* 28, 152–161.
- Lu, L., Liu, X., Huang, W.-K., Giusti-Rodríguez, P., Cui, J., Zhang, S., Xu, W., Wen, Z., Ma, S., Rosen, J.D., et al. (2020). Robust Hi-C Maps of Enhancer-Promoter Interactions Reveal the Function of Non-coding Genome in Neural Development and Diseases. *Mol Cell*.
- Luppino, J.M., Park, D.S., Nguyen, S.C., Lan, Y., Xu, Z., Yunker, R., and Joyce, E.F. (2020). Cohesin promotes stochastic domain intermingling to ensure proper regulation of boundary-proximal genes. *Nat Genet* 52, 840–848.
- Mateo, L.J., Murphy, S.E., Hafner, A., Cinquini, I.S., Walker, C.A., and Boettiger, A.N. (2019). Visualizing DNA folding and RNA in embryos at single-cell resolution. *Nature* 568, 49–54.

Matthey-Doret, C., Baudry, L., Breuer, A., Montagne, R., Guiguelmoni, N., Scolari, V., Jean, E., Campeas, A., Chanut, P.-H., Oriol, E., et al. (2020). Chromosight: A computer vision program for pattern detection in chromosome contact maps. *Biorxiv* 2020.03.08.981910.

Moorthy, S.D., Davidson, S., Shchuka, V.M., Singh, G., Malek-Gilani, N., Langroudi, L., Martchenko, A., So, V., Macpherson, N.N., and Mitchell, J.A. (2017). Enhancers and super-enhancers have an equivalent regulatory role in embryonic stem cells through regulation of single or multiple genes. *Genome Res* 27, 246–258.

Oksuz, B.A., Yang, L., Abraham, S., Venev, S.V., Krietenstein, N., Parsi, K.M., Ozadam, H., Oomen, M.E., Nand, A., Mao, H., et al. (2021). Systematic evaluation of chromosome conformation capture assays. *Nat Methods* 1–10.

Szabo, Q., Donjon, A., Jerković, I., Papadopoulos, G.L., Cheutin, T., Bonev, B., Nora, E.P., Bruneau, B.G., Bantignies, F., and Cavalli, G. (2020). Regulation of single-cell genome organization into TADs and chromatin nanodomains. *Nat Genet* 52, 1151–1157.

Thiecke, M.J., Wutz, G., Muhar, M., Tang, W., Bevan, S., Malysheva, V., Stocsits, R., Neumann, T., Zuber, J., Fraser, P., et al. (2020). Cohesin-Dependent and -Independent Mechanisms Mediate Chromosomal Contacts between Promoters and Enhancers. *Cell Reports* 32, 107929.

Xu, B., Wang, H., Wright, S., Hyle, J., Zhang, Y., Shao, Y., Niu, M., Fan, Y., Rosikiewicz, W., Djekidel, M.N., et al. (2021). Acute depletion of CTCF rewires genome-wide chromatin accessibility. *Genome Biol* 22, 244.

Decision Letter, Appeal:

Dear Dr. Tjian,

Thank you for asking us to reconsider our decision on your manuscript "Enhancer-promoter interactions and transcription are maintained upon acute loss of CTCF, cohesin, WAPL, and YY1".

I have now discussed the points of your preliminary rebuttal letter with my colleagues, and we think that the revision plan sounds encouraging. We therefore invite you to revise your manuscript along the lines that you propose. It'll be particularly important to carefully monitor protein degradation temporal dynamics (free and chromatin-bound), ideally using a side-by-side comparison between untagged and tagged (treated and untreated) lines to also control for the potential effects of the tag itself and auxin treatment. Residual protein levels at the 3-h time point may significantly impact the conclusions drawn so it's crucial that this point is investigated thoroughly.

When preparing a revision, please ensure that it generally complies with our editorial requirements for format and style; details can be found in the Guide to Authors on our website (<http://www.nature.com/ng/>).

Please be sure that your manuscript is accompanied by a separate letter detailing the changes you have made and your response to the points raised. At this stage we will need you to upload:

1) a copy of the manuscript in MS Word .docx format.

2) The Editorial Policy Checklist:

<https://www.nature.com/documents/nr-editorial-policy-checklist.pdf>

3) The Reporting Summary:

(Here you can read about the role of the Reporting Summary in reproducible science:

<https://www.nature.com/news/announcement-towards-greater-reproducibility-for-life-sciences-research-in-nature-1.22062>)

Please use the link below to be taken directly to the site and view and revise your manuscript:

[redacted]

With kind wishes,

Tiago

Tiago Faial, PhD

Senior Editor

Nature Genetics

<https://orcid.org/0000-0003-0864-1200>

Author Rebuttal, first revision:

Reviewer #1:

Hsieh et al. perform micro-C, ChIP-seq, mNET-seq, and live single molecular imaging on cell lines with acutely degraded components of genome organization. The effect on 3D folding of depleting CTCF, RAD21 and WAPL has been previously studied by Hi-C and DNA imaging, yet the role of YY1 remains more mysterious. The primary versus secondary transcriptional consequences of disrupted 3D folding have remained challenging to disentangle. Therefore, the presented approaches could potentially shed light on the connection between genome organization and transcription. It was greatly appreciated that authors made their data available for review. However, at this time there are serious issues with cell line generation (WAPL and YY1 specifically), data processing, and interpretation of the micro-C and mNET-seq.

Concerns regarding micro-C analysis

1.1. Serious concern that peak/cot calling picks up accessible/visible regions rather than reflecting 3D folding at enhancers and promoters. Central conclusions of the study build on observing micro-C dots that persist after CTCF/RAD21/WAPL/YY1 degradation. However it is not demonstrated that the high micro-C signal between enhancers and promoters reflects 3D folding, as opposed to 1D biases, e.g. due to the high accessibility of these elements. The fact that E/Ps appear to give strong micro-C signal with their entire surrounding, sometimes tens of kbs away (e.g. figure 1f) is surprising. It remains unclear to what extent does this pattern reflect the sensitivity of micro-C to the technical accessibility of these regions (as opposed to reflected 3D organization) - see below.

Response: The reviewer asked if the fine-scale chromatin dots detected in Micro-C are due to higher accessibility of micrococcal nuclease (MNase) to promoters and enhancers, which may bias the Micro-C signal towards these regions. Below we cite other studies and present results of additional analyses on our Micro-C data that demonstrate accessibility biases are largely negligible.

MNase digests chromatin uniformly

Recent studies from the Henikoff and Längst labs have experimentally and computationally addressed this issue (Chereji et al., 2019; Schwartz et al., 2018), indicating that MNase equally accesses DNaseI-hypersensitive and non-hypersensitive sites, sites of active and inactive transcription, and euchromatin as well as heterochromatin (see Fig.9 in Chereji et al, Fig.4 in Schwartz et al, and a snapshot from Chereji et al in **Fig. 1** below). They also state that “*although heterochromatin and euchromatin appear different when observed cytologically at low resolution, at the molecular level, MNase and other proteins can access heterochromatin regions at rates similar to those of accessing euchromatin*”. We thus conclude that MNase can digest chromatin uniformly across the genome regardless of its accessibility level.

(Fig. 1)

Nucleosome occupancy mapped in *Drosophila* S2 cells by MNase-seq. Rightmost panel: green (HP1-bound), yellow (active), red (active), blue (Polycomb-bound) and black (repressive) chromatin states defined in (Filion et al., 2010)

Micro-C is not biased toward chromatin accessible regions

Previously, we and others have independently demonstrated that Micro-C contacts are not biased toward highly accessible chromatin regions (Hsieh et al., 2020; Hua et al., 2021; Krietenstein et al., 2020). Notably, a recent Micro-C-based technique, Micro-Capture-C (MCC), that focuses on the interactions around promoter regions also did not find substantial bias toward DNaseI hypersensitive sites (Hua et al., 2021: Extended data Fig. 2). To further support this point, we provide key results below demonstrating that Micro-C data is not affected by chromatin accessibility levels. We thus conclude that the E-P interactions captured by Micro-C represent *bona fide* proximal chromatin contacts.

1. When we analyzed Micro-C reads as single-end data like ChIP-seq or MNase-seq, we do not observe any apparent correlation with chromatin accessibility. In **Fig. 2** (left panel), we show that neither balanced (KR/ICE) nor coverage-normalized Micro-C data show substantial correlation with ATAC-seq signal ($r^2 \ll 0.1$). Even the raw Micro-C reads have nearly no correlation with ATAC-seq signals ($r^2 = 0.15$). We note that all the Micro-C data in this manuscript were normalized either by matrix balancing or by coverage (detailed in the Methods section). In fact, regions with low ATAC-seq signal (low accessibility) have slightly higher Micro-C signal than regions with high ATAC-seq signal (**Fig. 2**, right panel), arguing against Micro-C simply reflecting chromatin accessibility.

(Fig. 2)

2. To further demonstrate that Micro-C is not biased toward capturing highly accessible regions such as promoters, we performed a genome-wide pile-up analysis of Micro-C at base-pair resolution surrounding active promoter regions (**Fig. 3**). Coverage-normalized Micro-C shows nearly no bias toward the +1/-1 nucleosome region (more accessible) compared to the signal within the gene body (less accessible).

(Fig. 3)

3. If contacts at accessible promoters/enhancers were artifactually amplified by Micro-C, we should expect them to be far more prevalent than contacts at polycomb/heterochromatin regions. However, genome-wide $P(s)$ analysis shows no evidence of significant differences in contact probability for different types of chromatin regions, including accessible promoters and enhancers, Polycomb-bound and heterochromatic domains, within a 500-kb range (**Fig. 4**).

(Fig. 4)

4. As we showed in our previous work (Hsieh et al., 2020), very deeply sequenced Hi-C data (Bonev et al., 2017: ~3.3B reads) can recapitulate some E-P loops detected by Micro-C (**Fig. 5**). While these dots in Hi-C are much fuzzier and their precise locations are often indiscernible, the example below clearly indicates that they are present in both data. It is widely accepted in the 3D-genome field that Hi-C has minimal bias towards highly accessible chromatin, so contact enrichment around E-P interactions present in both methods cannot be an artifact of Micro-C. Critically, Micro-C can further resolve what appears as a large blurry dot in Hi-C into multiple finer-scale dots and pinpoint the precise location of their interactions (**Fig. 5**, zoomed-in box), indicating that Micro-C is a superior method to Hi-C in identifying fine-scale interactions. Additionally, the latest Micro-C and Hi-C benchmark study by the Dekker lab at UMMS recommends using Micro-C for loop detection (e.g., cis-regulatory elements interactions) due to its supreme sensitivity (Oksuz et al., 2021).

(Fig. 5)

Together, our results and those of others strongly support that Micro-C does not exhibit any systematic bias toward highly accessible chromatin regions. From here on, we will refer to these initial points whenever a reviewer's concern assumes a Micro-C accessibility bias. We included the Micro-C quality control for chromatin accessibility in the manuscript **Extended Data Fig. 1a**.

1.1.1 Issues with high-resolution dot calling.

The authors rely on published computational methods for calling TADs and loops/dots. However, these methods were not developed for or validated at the resolutions the authors use. For example, the highest resolutions considered were 1 kb and 10 kb in the mustache and chromosight publications for human data. The dots called here are highly biased for increased coverage relative to the genome-wide average at high resolutions. The plot below shows genome-wide average coverage at 400-bp resolution as a black line, and average coverage under the called dot/loop anchors as a function of map resolution (inferred as end-start) in blue:

Calling dots at very high resolution (< 1 kb) requires an approach that statistically models 1D coverage and verification that the dots called do not display a coverage bias. An example of such an approach is HiC-DC+ (PMID: 34099725), but even this approach would require extension and validation below 1 kb. If obtaining calls without coverage biases at sub-kb resolution is not feasible, 3D genomic analysis would benefit from being restricted to resolutions >= 2 kb, where dot calls show a less pronounced coverage bias, and existing computational tools have been validated. Crucially, if dot calls have a coverage bias, they cannot be interpreted as strictly reflecting 3D proximity.

**Response:** We were aware of this potential caveat before using Mustache and Chromosight for high-resolution dot calling. In our communications with the authors of Mustache (Dr. Ferhat Ay, La Jolla Institute) and Chromosight (Dr. Romain Koszul, Institute Pasteur), they do not see any issue with using their algorithms to identify high-resolution dots in Micro-C data. They emphasized that the maximal spatial resolution for dot calling is largely dependent on the sequencing depth. There should be no theoretical limitations to applying the algorithms to identify dots in our deepest sequenced Micro-C dataset. Dr. Ay did not apply Mustache to data with < 1-kb resolution simply because such data was not available in the 4DN database at that time. The Koszul group did apply Chromosight to ChIA-PET data at 500-bp resolution (Matthey-Doret et al., 2020a: Fig. 3), and Chromosight has been routinely applied to yeast Micro-C data using 200- or 400-bp resolution. Importantly, Micro-C matrices are normalized using KR/ICE before loop detection. Thus, the potential coverage biases are greatly reduced (as shown in **Fig. 2-4** above).

To test whether loop calling is affected by coverage in our data, we analyzed “1D coverage vs. the number of significant loops per locus”. **Fig. 6** below shows a similar trend across all resolutions, suggesting that 1D coverage does not explicitly affect dot calling in the high-resolution data (Wilcoxon Rank Sum and Signed Rank Tests).

**(Fig. 6)**

We also note that only the initial *de novo* loop calling for wild type mESC (Hsieh et al., 2020) by Mustache or Chromosight was done with resolutions < 1 kb. All the subsequent dot quantifications presented throughout the manuscript were analyzed with data at 2- or 4-kb resolution. We revised the figure legends and Method section to clarify our analysis details. In summary, our loop analysis is as follows.

1. Micro-C data	Wild-type mESCs (Hsieh et al 2020, 2.6B reads)	Δ CTCF, Δ RAD21, Δ WAPL, Δ YY1 (This study, ~600M reads)
2. De novo loop calling	Mustache / Chromosight at resolutions of 400bp, 600bp, 800bp, 1kb, 2kb, 4kb, 10kb, 20kb	Mustache / Chromosight at resolutions of 1kb, 2kb, 4kb, 10kb, 20kb
3. Subtype classification	ChromHMM Cohesin, E-P, P-P, Polycomb	ChromHMM Cohesin, E-P, P-P, Polycomb
4. Quantification	Chromosight / Hiccups at resolution of 2kb	Chromosight / Hiccups at resolution of 2kb
5. Filtering	Loop strength q-value	Loop strength q-value
6. Downstream analysis		

Finally, to determine whether dots identified with < 1-kb resolution data may possibly affect our results, we kept dots only from > 1-kb resolution data and repeated our analysis. We obtained ~56,377 dots that show no change from our previous results of ~75,190 dots in terms of loop strength, loop size, and correlation with transcription (**Fig. 7**; compare with Fig. 1b in the main text). Thus, according to our results and those of other groups, Mustache and Chromosight can be applied to identify dots in high-resolution Micro-C data without exhibiting a bias that could compromise our conclusions.

(Fig. 7)

1.1.2 Issues for interpretation related to possible visibility biases of enhancers and promoters. Key contributions of the manuscript would not stand if these dots were called due to technical biases instead of actual 3D folding:

“Specifically, the strength of E-P and P-P interactions positively correlates with the level of gene expression, while cohesin loops show no such correlation” This is expected if the micro-C signal at sub-kilobase resolution is sensitive to E-P accessibility.

Response: We show that these results hold true when using Hi-C data, albeit the degree of correlation between E-P/P-P strength and gene expression is much more evident in Micro-C data (**Extended Data Fig. 1j**).

“Remarkably, acute depletion of CTCF and cohesin had a negligible impact on the E-P and P-P loops, with ~80% of E-P contacts (Fig. 3c) and 90% of P-P contacts (Fig. 3d) remaining unaltered.” Due to the coverage bias at dots called at higher resolutions, this likely reflects the visibility/accessibility of these regions, rather than their physical interaction in 3D. It sounds like the nature of the micro-C dots called between E-P (3D contacts vs. accessibility/technical biases) will remain uncertain until authors can profile a condition or treatment where the micro-C signal between E-P is affected genome-wide while accessibility remains unchanged.

Response: Since we have demonstrated that the Micro-C signal is not biased toward accessible chromatin (please see the response to **1.1**) and we have provided strong evidence to dispute the problem of calling dots at high resolutions (please see the response to **1.1.1**), we assert that all the main conclusions in the manuscript remain unchanged. Importantly, we here emphasize that the E-P interactions were quantified with Micro-C data at 2- or 4-kb resolution, which is clearly not affected by possible visibility biases as the reviewer stated.

In addition, a recently published paper indicated that CTCF loss significantly changes ~70% of ATAC-seq peaks located around promoters, enhancers, and CTCF binding sites (Xu et al., 2021). If the changes in chromatin accessibility were to bias the Micro-C signal, we would expect massive changes in E-P interactions in our results. However, we find that only < 15% of total E-P/P-P interactions are affected by CTCF depletion, further ruling out the possibility of an accessibility bias in Micro-C data.

1.1.3 Orthogonal experimental validation. If technically possible, an orthogonal method such as DNA FISH (e.g. ORCA?) based assays could be deployed to establish the nature of the E-P interactions called at high-resolutions in micro-C and address concerns of point 1.

Response: The average length of high-resolution E-P loops is ~20 kb in mammals (Gasperini et al., 2019: CRISPRi screening assay; Hsieh et al., 2020: Micro-C). However, the published ORCA method only resolved a ~130-kb region with a 2-kb resolution at best (Mateo et al., 2019). Given the localization error derived from fluorescence-based imaging (Brandão et al., 2021), we doubt that DNA FISH or ORCA would provide conclusive results to validate the high-resolution Micro-C signal. In addition, perfecting a robust experimental system and analysis framework could take months or longer, while we believe this burdensome effort would at best add tangentially to the primary focus of this manuscript. We thus think validation using ORCA is not an ideal option.

Instead, we think that analyzing the Micro-C signal surrounding well-characterized and functionally validated E-P contacts (e.g., by a CRISPR-based assay) (Moorthy et al., 2017) is more likely to give us a definitive conclusion. Indeed, in our previous mammalian Micro-C paper (**Fig. 8**) we confirmed that these E-P interactions can be clearly detected as dots or stripes in Micro-C, suggesting that Micro-C can capture the most genuine and functional 3D chromatin contacts.

(**Fig. 8**)

1.2.1 Mapping & filtering, and genotypes. There appear to be issues with either mapping or cell line differences from the reference assembly leading to off-diagonal artifacts. It would be helpful to demonstrate that such features do not impact downstream analyses.

Response: We thank the reviewer for pointing this out. After rechecking our Micro-C contact matrices, we found that the off-diagonal signals are rare and appear even more widespread in the standard 4D nucleome Hi-C data (see examples shown in the same region in **Fig. 9**). These signals are thus not artifacts derived from our cell genotype or data mapping and filtering. Additionally, these off-diagonal noises are removed at the beginning of our analysis pipeline due to the lack of promoters and enhancers (most of them consist of repeat elements) and span a much larger distance than most E-P interactions. We thus argue that off-diagonal artifacts have minimal, if any, effect on our E-P loop analysis.

(Fig. 9)

1.3 Concern that degron leakiness alters the perceived effect of degrading target proteins. Substantial variation exists between untreated (UT) conditions, which is not accounted for in the analysis or interpretation. For example, dots readily visible in YY1-UT appear absent in all other UT samples in the genomic region displayed in Figure 1.

Differential analyses relative to untagged reference cells, in addition to untreated cells, should be reported.

Response: To determine to what extent degron leakiness may affect our “untreated” cells, we quantified CTCF, RAD21, WAPL, and YY1 protein levels in each degron clone and compared them to wild-type cells by Western blot (**Extended Data Fig. 2b-c** and **Fig. 17** below). We also conducted correlation test, reproducibility tests and differential analyses of all untreated samples in ChIP-seq (**Extended Data Fig. 3a-c**), Micro-C (**Extended Data Fig. 4a-d**), and RNA-seq data (**Extended Data Fig. 5a**). Although CTCF-AID, RAD21-AID, and YY1-AID cell lines show some basal degradation (~40%, ~30%, ~25% reduction of protein levels compared to wild type cells, respectively), we find no significant changes in chromatin association, 3D genome organization, or transcriptome in these untreated cells. The results of our quality controls are summarized below in **Fig. 10** (please refer to the manuscript for full details).

(Fig. 10)

The results of the pairwise correlation analysis indicate that ChIP-seq, RNA-seq, and mNET-seq data are highly reproducible between cell lines in the untreated condition.

The results of the pairwise correlation and decaying curve analysis indicate that Micro-C data are highly reproducible between cell lines in the untreated condition.

Concerns regarding transcription analysis by mNET-seq

2.1 Lack of positive control to conclude about the absence of change. Authors make strong and thought-provoking claims about the lack of widespread changes detected by their mNET-seq approach. This raises the question: can authors detect widespread transcriptional changes with their approach? The conclusions as stated require reporting experiments where a greater number of changes are detected, as to set the basal expectation.

2.1.1 It is essential that authors establish the time resolution of mNET-seq in their hands. In other words how quickly can authors expect to detect pervasive transcription changes? Assaying the CTCF/RAD21/WAPL-AID cells at later time points could establish that transcriptional dysregulation can eventually be detected in these cells with the mNET-seq approach. Given that CTCF and WAPL depleted cells do not have cell cycle defects it may make most sense to assay one or both of these lines at later time points (e.g. 6h, 12h or 24h).

2.1.2 It is essential that authors establish expectations after short-term degradation of factors more directly involved in transcription. Without this comparison it is difficult to conclude much about the small number of differentially transcribed genes after only 3h. YY1 depletion dysregulated ~325 genes, which is the most compared to CTCF/RAD21/WAPL. It is difficult to estimate what is the upper expectation for the number of dysregulated genes after acute depletion of a transcription factor. This could easily be achieved using mESCs where transcription factors more directly linked to promoter or enhancer activity can be depleted and performing

the same mNET-seq experiments after 3hrs (e.g. existing OCT4, NANOG or SOX2-degron cells, or using BRD inhibitors).

Response: To address comments **2.1**, **2.1.1**, **2.1.2**, we now include a positive control (dBET6 BRD inhibition) and a time-course mNET-seq experiment (0, 3, 12 and 24 hrs of IAA treatment) for CTCF, RAD21, WAPL, and YY1 depletion with a calibration control in the revised manuscript.

Consistent with a previous study that identified ~7,400 differentially expressed genes (DEGs, $p_{adj} < 0.05$) (Arnold et al., 2021), we detected ~6,410 DEGs ($p_{adj} < 0.01$ & > 2 -fold change) after a 2-hr treatment with dBET6 (**Extended Data Fig. 5b**). The result indicates that our mNET-seq method can detect massive transcriptional changes.

Our degradation time course revealed that CTCF depletion leads to immediate transcriptional deregulation of ~60 genes ($p_{adj} < 0.01$ & > 2 -fold change) (**Fig. 2h** and **Extended Data Fig. 5c-d**). The result is consistent with a PRO-seq analysis (Luan et al., 2021) that reported ~122 DEGs ($p_{adj} < 0.05$ & > 2 -fold change) after 4 hours of CTCF depletion. While RAD21 depletion for 3 hours did not cause significant transcriptional changes, extended cohesin depletion led to drastic transcriptional deregulation (over thousands of DEGs), possibly because of the secondary effects such as cell cycle defects (**Fig. 2h** and **Extended Data Fig. 5c-d**). While our WAPL depletion clearly increases the fraction of cohesin retention on chromatin (~35%) (**Extended Data Fig. 2h**) and expands the grid of cohesin loops in Micro-C maps (**Extended Data Fig. 4h**), we do not detect a global effect on transcription across the time course (~10 DEGs, $p_{adj} < 0.01$ & > 2 -fold change) (**Extended Data Fig. 5c-d**), contradicting the previous study that found ~779 DEGs ($p_{adj} < 0.05$) after 6-hour WAPL depletion using TT-seq (Liu et al., 2021). We noticed that we have used a more stringent cut-off to call DEGs than other studies. By adjusting the threshold to $p_{adj} < 0.05$, we can detect more DEGs in the range comparable to previous studies (DEGs=~394).

Together, we conclude that 1) our mNET-seq method is sensitive to massive transcriptional alterations; 2) our mNET-seq results largely agree with previous reports; and 3) many of the DEGs detected using “ $p_{adj} < 0.05$ ” fail to survive with “ $p_{adj} < 0.01$ & 2-fold change”, indicating that these changes to gene expression are mild (see Table below).

mNET-seq or similar nascent RNA profiling methods					
Depletion	Time after +IAA / previous studies	$p_{adj} < 0.01$ shrinkage(2x FC)	$p_{adj} < 0.01$ 2x FC	$p_{adj} < 0.01$	$p_{adj} < 0.05$
CTCF	3h	50	73	212	295
	12h	69	117	327	556
	24h	76	194	646	1183
	Luan et al, G1E-ER4, 4h, PRO-seq				122
RAD21	3h	5	6	94	196
	12h	941	1313	3754	5240
	24h	2274	2727	5353	7680
	Thiecke et al, HeLa, 1h, SLAM-seq				687
Rao et al, HCT-116, 6h, PRO-seq				4196	
WAPL	3h	2	2	5	6
	12h	9	39	147	394
	24h	11	44	164	437
	Liu et al 2021, 6h, TT-seq				779
Liu et al 2021, 24h, TT-seq				2733	

* Data in this study are colored with darker shade.

* This manuscript reports the results obtained with the most stringent cut-off ($p_{adj} < 0.01$ & > 2 -fold change after LFC shrinkage).

2.2 Authors must confirm that the dysregulated genes are changing because of CTCF/RAD21/WAPL rather than auxin treatment.

Authors state that “This suggests that while CTCF and cohesin are required for the transcriptional maintenance of only a small subset of genes, those genes tend to require the presence of both factors.” An alternative explanation is that auxin treatment in Tir-expressing cells affects those genes, irrespective of CTCF/RAD21 degradation (as reported by Liu et al. 2021 Nature Genetics PMID 33318687 by mRNA-seq at later timepoints). For example it is surprising that *cyp1b1*, a member of the cytochrome P450 family, is the highest upregulated gene across conditions. It would be important that authors perform mNET-seq in such their Tir1-only expressing cells treated with auxin, or rephrase their statements and interpretations to account for this possibility.

Response: In our latest analysis of the new data and Tir1 parental cells treated with IAA we do not observe a significant change in those cytochrome-related genes even after 12-hr treatment (**Extended Data Fig. 5c-d**). The full list of RNA-seq and mNET-seq results are available in **Supplemental Table 11-12**.

2.3 In the absence of internal calibration, is it possible that authors are missing global down- or up-regulation of transcription (e.g. see Arnold et al. 2020 Mol Cell PMID 34324863 for how internal calibration of mNET-seq can alter interpretations)? Please comment on the ability of the current protocol to detect global homogeneous changes in the main text, assuming that performing calibrated mNET-seq experiments is out of the question.

Response: We included an internal calibration for our latest RNA-seq (ERCC) and mNET-seq (*Drosophila* S2 cells) experiments. Consistent with previous reports claiming no global transcription change after CTCF and RAD21 depletion (Luan et al., 2021; Rao et al., 2017), our calibration also indicates that there is no apparent global effect on gene expression after loss of CTCF, RAD21, WAPL, and YY1 (**Extended Data Fig. 5c-e**).

2.4 The authors do not report the technical reproducibility of their mNET-seq and RNA seq data. Please include pairwise correlations between all replicates in all conditions (using either gene body or promoter for mNET-seq).

Response: We included the reproducibility analysis in **Extended Data Fig. 5a**. Please also see a brief summary in **Fig. 10** above.

2.5 Please include a suppl. table with the differentially transcribed genes detected in each mutant.

Response: We included the tables reporting the results of RNA-seq and mNET-seq in **Supplemental table 11-12**.

Further confirmation that these new AID cells undergo near-complete degradation would be helpful.

3.1 Controls for all AID targets. Western blot is unlikely to pick up 5-10% remaining protein. Given that authors introduced fluorescent tags together with their degrons it would be helpful to show flow-cytometry data of untreated and 3h-treated cells together with untagged cells, separately for CTCF, RAD21, WAPL (they already present the YY1 data in Ext. Data Fig. 5a). It would also be especially helpful if authors could report whether long-term depletion leads to the cellular defects previously reported for these factors in mouse ES cells (CTCF: stop in proliferation after 4 days; RAD21: block in mitosis and cell death within 2 days ; WAPL: exit from pluripotency after 2-4 days - See Liu et al. 2020 Nature Genetics PMID 33318687, Nora et al. 2017 Cell PMID 28525758, Kubo et al. 2021 NSMB PMID 28525758 etc...). Can authors comment on what happens to YY1-depleted cells, do they survive?

Response:

Validation of target protein degradation

Since excessive amount of BFP or auto-fluorescence accumulate in the cytoplasm after IAA treatment, we were unable to accurately quantify relative protein levels using flow cytometry (**Fig. 11A**). We thus quantified nuclear BFP intensity using confocal imaging and found that the results are

consistent with our Western blot data (**Fig. 11B**; please see (Gabriele et al., 2021), Fig. S4 for full details). We also used biochemical fractionation to enrich the chromatin-bound fractions and determine how much of the target protein residue remained on chromatin. Upon 3-hr depletion, CTCF (100%), RAD21 (100%), and WAPL (~90%) are nearly completely degraded and free from the chromatin fraction (**Fig. 11C** and **Extended Data Fig. 2h**).

Cell growth/death analysis

Consistent with previous findings the reviewer cited above, we found that the depletion of CTCF and RAD21 led to cell growth defects, while WAPL depletion caused only a mild growth defect (**Fig. 11D**; please see (Gabriele et al., 2021), Fig. S5 for full details). Additionally, we analyzed the ratio of dead cells by staining them with Annexin V and Propidium Iodide throughout the degradation time course. Only RAD21 depletion results in widespread cell death after 12 hours (**Fig. 11E**).

Together, we confirmed that proteins are nearly completely removed from chromatin within 3-hr of IAA treatment, and cells depleted of these proteins display the same long-term phenotypes as the previous reports.

(Fig. 11)

3.2 Controls for WAPL deficiency. Authors assume near-complete depletion of WAPL based on Western blot analysis with the HA tag, brought together with the AID. However given the micro-C signal it seems unlikely that the Wapl depletion triggered maximum cohesin retention. Extended Data Figure 3D indicates that the micro-C signal at pre-existing dots remains unchanged after WAPL depletion. It looks like Wapl loss is much less complete than other studies in the field, e.g. Haarhuis et al. 2017 (Cell PMID 28475897) and Wutz et al. (PMID: 29217591) where extended grids of peaks emerge and the derivative of contact density shifts strongly rightward. This makes the author's reports of little transcriptional change somewhat inconclusive, as it may simply be due to technical issues of their cell line generation. It would be helpful if authors could characterize the effect of their WAPL degradation: How much more cohesin is there on chromatin after 3hr of WAPL depletion in these cells? It would be informative if authors could perform calibrated ChIP-seq for core cohesin subunits after 3hrs of WAPL degradation (compared to either untreated and the WAPL untagged cells with all

the other tags also present in the WAPL-AID line [CTCF-halo, RAD21-SNAP]). Alternatively authors could perform careful quantifications of chromatin-bound core cohesin subunits by western-blot, but this might be difficult as one would only expect a maximum 2-fold increase of cohesin, given that around 50% of cohesin is already bound before WAPL depletion. The authors should also update their reported conclusions in light of the potential technical confounders.

Response: We believe the reviewer may be misinterpreting some of the results reported in those two WAPL perturbation papers. Haarhuis et al. used the HAP1 cells with WAPL knockout, and Wutz et al. used RNAi to knock down WAPL in HeLa or MEFs for 3 days. Their experimental conditions and the questions of interest are very different from ours. First, since the residence time and target search time of cohesin are longer than 20 min in mESCs, we do not necessarily expect that cohesin can reach maximal retention after only 3 hrs of WAPL depletion (and we did not claim it in the manuscript). We are more interested in the immediate short-term effect of WAPL depletion on the 3D genome and transcription, rather than studying the endpoint result after cells undergo unknown processes. Second, we do not expect WAPL depletion to drastically increase the contact intensity of pre-existing loops. As shown in Haarhuis et al. and the latest pre-print in Liu et al. by the de Wit lab, WAPL depletion does not substantially increase the contact intensity of the pre-existing dots (**Fig. 12**, top panels). By comparing the $P(s)$ curve from Haarhuis et al. and differential $P(s)$ curve (6 hr, green line) from Liu et al. with our Micro-C data, we find a similarly right-shifted $P(s)$ curve (**Fig. 12**). Critically, aggregate peak analysis across multiple ranges of genomic distance shows substantially stronger extended cohesin loops (> 250 kb and beyond) in WAPL depletion (**Fig. 12**, bottom panel), indicating a longer retention of cohesin-mediated extrusion. Third, we observe ~35% increase of cohesin binding on chromatin following 3-hr WAPL depletion in our biochemical fractionation assay (**Extended Data Fig. 2h**). Together, we provide strong evidence that a 3-hr WAPL depletion is sufficient to increase cohesin retention on chromatin and extend the grids of cohesin loops. However, these additional long-range loops have no immediate effect on transcription.

Even though the reviewer's concerns here are based mainly on some misguided references to previous studies, we added a comment in the main text noting that our WAPL cell line does not achieve maximum degradation and cohesin retention within 3 hours, but it is sufficient to determine whether additional cohesin retention and extended loops may affect transcription.

(Fig. 12)

3.3 Issues with YY1 degradation.

The majority of YY1 peaks were diminished, but only ~50% of peaks (n = 15075) were called significantly changed by differential peak analysis. Does that mean the depletion is not complete? How do we interpret the seemingly complete depletion base of Western and flow cytometry then - are these assays not sensitive enough? Or is it because the YY1 ChIP-seq was not calibrated? Could it be because authors use the miniIAA7 tag with osTir1, instead of atAFB2 as recommended by citation 64, Li et al. Nat Methods PMID 31451765?

Response: We performed a new spiked-in ChIP-seq experiment for YY1 using different antibodies to validate that the residual signal was not caused by detection of off-target proteins by the antibody. We confirmed that >85% of peaks (total=34342) were significantly reduced upon loss of YY1 (**Extended Data Fig. 7c-e**). Our differential peak analysis shows consistent results regardless of whether spike-in normalization is included (with spike-in: 94.7% changes, without spike-in: 85.6% changes), suggesting that our analysis using empirical Bayes approach is less biased by global homogenous change. Furthermore, we confirmed by confocal imaging and biochemical fractionation that YY1 degradation is nearly complete (**Extended Data Fig. 7b and h**).

For YY1 degradation, we did use the parental cell line with stably expressing atAFB2 as reported in (Li et al., 2019).

Conflating time elapsed with primary versus secondary effects on transcription. Author conclusions assume that the larger transcriptional consequences reported by others in similar cell lines likely reflects secondary consequences from a small subset of affected genes. While this could very well be true, it is also possible that direct primary transcriptional consequences of RAD21/CTCF/WAPL/YY1 depletion are slow-manifesting. Some evidence suggest that some of the slow-manifesting transcriptional picked up by previous studies could be consequences of direct cis-regulatory impacts: for example, over 80% of the genes down-regulated at mRNA level after one day of CTCF degradation by Nora et al. 2017 Cell PMID 28525758 had CTCF binding in their promoter. While it is challenging to disentangle slow primary defects from rapid secondary ones with the approaches used by the authors, the possibility that some primary consequences may be slow-manifesting should be discussed off the bat in the text, as it is an obvious expectation, rather than brought up only in the discussion under a “time-buffering” model.

Response: The reviewer cited Nora et al. to describe the “slow-manifesting” transcriptional effect 1 day after CTCF depletion. However, the biological definition of a “slow-manifesting” effect is unclear to us. First, promoter bound CTCF can act as a transcription factor or a mitotic bookmarking factor. In this case the definition of “slow-manifesting” is in line with our “time-buffering” model. Second, this study used bulk RNA-seq analysis to interpret the transcriptional effects of CTCF depletion. This means their observations are comprised of mixed results of 1) 3D genome reorganization upon CTCF depletion; 2) secondary transcriptional effects; 3) total pool of mRNA before and after CTCF depletion; and 4) secondary effects of post-transcriptional RNA processing and buffering. Thus, we do not plan to discuss the “slow-manifesting” transcription in our manuscript without a definitive clarification and example.

5. Please include annotated .gb files for all targeting constructs mentioned in Table S1, rather than stating “sequence available upon request”. Please either make the vectors available on addgene or by simply add annotated .gb sequence files as supplementary data.

Response: We now provide the sequence files of all targeting constructs in **Supplementary files**.

Minor points:

Captions are not always complete. e.g. it is not clear what microC data are plotted in Fig. 1: untreated YY1? something else?

Response: Corrected.

Authors mention identifying 75,000 dots in the text but the 4-category breakdown in figure 1a-b only adds up to 42,000 (assuming the random and promoter-random categories are shuffled controls?). Please clarify and include the missing “other” category in the figure caption.

Response: About 25% of loop anchors do not enrich for any specific 1D chromatin pattern. Those loops were excluded from further analysis. Please also see Hsieh et al (2019) for more details and full list in **Supplemental Table 6**.

A heatmap showing correlations between micro-C replicates, or a table or pairwise correlations, should be added to the supplement.

Response: Added in **Extended Data Fig. 4b-d**.

It would be helpful if authors could refrain from using interchangeably “dots” and “loops”. “Dots” seems the most accurate description of the peak-calling result, without making assumptions about the underlying conformation which can only be resolved through modeling or imaging.

Response: The focal enrichment contacts in Hi-C data are referred to as “loop” (e.g., CTCF/Cohesin loops) and enhancer-promoter interactions are historically called E-P loops in most of the literature. While we agree with the reviewer’s point that not all cohesin-mediated contacts and E-P/P-P interactions are formed through physical looping, and some studies may suggest using “dots” instead of “loops,” we chose to use “loops” over “dots” to describe these enhanced focal contacts in this manuscript to match the most commonly used term in the field. We now include a note in the Methods section to motivate our terminology choice.

Is there anything special about the genes immediately dysregulated after CTCF degradation? E.g. they have CTCF binding in their promoter? Are they sitting next to any relevant micro-C feature (stripe, dot, boundary...)? It would be helpful if authors confirmed their data showed these aspects of previous studies, or discuss why it might not.

Response: The genes that immediately respond to 3-hr CTCF depletion show slightly higher CTCF occupancy than other genes (**Fig. 13**, left). These dysregulated genes are also enriched at loop anchors (called by mustache) and TAD boundaries (called by arrowhead) (**Fig. 13**, right), but the enrichment is restricted at the early time point, which again points to indirect transcriptional effects of CTCF depletion at later time points.

(Fig. 13)

“These studies have shown that while CTCF and cohesin play only a minor role in compartmentalization” While this is true for CTCF depletion, it was reported by many group that cohesin antagonizes compartmentalization, so that compartmentalization patterns are greatly affected in RAD21, NIPBL or WAPL depletion (e.g. Rao et

al. 2017, Schwarzer et al. 2017, Haarhuis et al. 2017). Authors may want to reword to reflect this.

Response: We corrected this sentence to *“These studies have shown that while CTCF plays only a minor role in compartmentalization, CTCF and cohesin removal largely eliminates TADs and chromatin loops anchored by these proteins across the genome.”*

The following sentence makes it sound like measuring 3D interactions between E-P by micro-C at sub-kilobase resolution in mammalian genomes is already established. “We recently reported that Micro-C can effectively resolve ultra-fine 3D genome folding at nucleosome resolution, including E-P and P-P interactions, thus overcoming this limitation, 24–27.” Ref 27 (Hansen et al. 2017) is cited to support that micro-C can effectively detect E-P and P-P interactions at nucleosome resolution. The sequencing depth in that study did not allow for nucleosome resolution. Indeed Hansen et al. 2017 reported around 14,000 loops vs 75,000 here. Similarly Ref. 24-25 are previous publications from the present author this time in *S. cerevisiae*, where there was no mention of E-P or P-P dots at nucleosome resolution. Please remove these citations here or rephrase the sentence to distinguish technology development (micro-C) from biological interpretation (E-P interactions).

Response: We corrected the citations in this sentence to *“We recently reported that Micro-C can effectively resolve ultra-fine 3D genome folding at nucleosome resolution^{24,25}, including E-P and P-P interactions^{7,17}, thus overcoming this limitation.”* Ref 24 and 25 are the first two yeast Micro-C studies probing nucleosome-resolution chromatin structure. Ref 7 and 17 are the first two Micro-C studies applied to mouse and human that focus on E-P interactions.

“We note that Micro- C has much higher sensitivity for detecting E-P and P-P contacts compared to Hi-C, establishing Micro-C as a more suitable assay to study genome organization relevant to transcription regulation genome-wide in an unbiased manner.” It is not fair for authors to state micro-C is unbiased, it simply has milder biases than Hi-C, so this should be reworded.

Response: We changed to *“We note that Micro-C has much higher sensitivity for detecting E-P and P-P contacts compared to Hi-C, establishing Micro-C as a more suitable assay to study genome organization relevant to transcription regulation genome-wide in a less biased manner”*

“Newly transcribed RNAs generally have a higher correlation with E-P contacts than with compartments and TADs (Extended Data Fig. 1f)” what does correlation with TADs mean? Which score is used? Strength of insulation? Averaged over the two boundaries? Please clarify.

Response: We used the corner score from the Arrowhead analysis. In general, higher values indicate a greater likelihood of being at the corner of a contact domain, which is largely correlated with the strength of TADs.

In paragraph “Acute depletion of CTCF, cohesin, and WAPL perturbs structural loops,” please include a sentence in the main text explaining that these experiments were done without internal spike-in calibration, so authors cannot conclude about possible differences in overall binding (it is likely that the loss of CTCF/RAD21 signal is under-estimated without calibration). Without calibration, authors should temper the statement “While cohesin peaks are generally lost upon CTCF degradation, CTCF binding is unaffected by altering the level of cohesin on chromosomes” - it is possible that CTCF levels are overall higher or lower. “RAD21 or WAPL depletion caused only a 10 to 20% reduction in CTCF peaks” - please clarify “number of peaks” (current sentence could be interpreted as referring to signal intensity). Maybe worth rephrasing to also account for that in ref. 11 (Rao et al. Cell 2017) the CTCF ChIP-seq after RAD21 depletion was not calibrated either.

Response: 1) We included a sentence indicating our ChIP-seq data for CTCF and Cohesin are not spike-in normalized. 2) However, the description about changes in chromatin association is confirmed by biochemical fractionation analysis. Our conclusion thus remains the same. 3) The sentence is now corrected.

“Nevertheless, upon rapid induction (1 hour) of erythroid differentiation, YY1 triggers little or no change in

H3K27ac and H3K27ac-anchored HiChIP interactions” - phrasing is confusing. Maybe authors could instead write that “YY1 redistribution triggers little or no change...”, as this was not an inducible-degradation approach.

Response: The sentence is rephrased.

YY1 was suggested to play a more prominent role in differentiated cells compared to mouse ES cells. It would be fair to acknowledge this when citing ref. 61. Beagan et al. Genome Research PMID 28536180.

Response: We included the details.

Please rephrase the following sentence to reflect that the authors only investigated the situation after 3hrs. “In summary, we find that, while CTCF, cohesin, and WAPL may regulate some gene expression, their acute depletion affects the transcription of only a handful of genes in mESCs, which largely encode pluripotency and differentiation factors.”

Response: We rephrased to “*In summary, we find that, while CTCF, cohesin, and WAPL may regulate some gene expression, their acute (3 hours) depletion affects the transcription of only a handful of genes in mESCs, which largely encode pluripotency and differentiation factors.*”

Tir1 transgenes are missing in cartoons of Extended Fig. 2 for all the cell lines.

Response: We included the information.

Reviewer #2:

In their manuscript, Enhancer-promoter interactions and transcription are maintained upon acute loss of CTCF, cohesin, WAPL, and YY1, Hsieh et al. explore the role of architectural proteins on E-P interactions and gene expression by a combination of micro-C and nascent transcriptomics in mESCs. Their findings are as follows:

- 1- Using micro-C in combination with a new loop caller, Mustache, they identify many new looping interactions between enhancer and promoters.
- 2- Acute (3hr) loss of CTCF, Rad21, WAPL, and YY1 does not perturb the frequency, location, or strength of most newly identified E-P interactions.
- 3- Acute (3hr) loss of CTCF, Rad21, WAPL, and YY1 does not affect global gene expression.
- 4- Cohesin loss alters YY1 binding dynamics.

The authors propose a time-buffering model, which I quite like and do think that it reconciles their and others' largely negative results related to acute loss of architectural proteins and gene regulation. However, I do have some concerns outlined below:

The main criticism of this paper pertains to the validity of the newly identified loops, which is the basis of their largely negative results and major conclusions. The authors utilize a new genome-wide, 3C-based method known as Micro-C along with a very new loop caller they recently developed (Mustache) and claim to identify new looping interactions not previously identified by Hi-C. Surprisingly, the loop caller alone adds a significant amount of new loops even from the authors' previous Micro-C analyses. While this makes sense in theory given the higher resolution MNase digestion would generate over convention, some of the features associated with these new loops does raise a few red flags.

That is, the authors find that these new loops correlate with transcription, frequently traverse TAD boundaries, and are robust to acute depletion of CTCF, cohesin, WAPL, and YY1. While these interactions may indeed be a novel and important discovery, the authors' conclusions are almost entirely based on these negative results. Importantly, they do not completely rule out the possibility that these new interactions are instead an artifact of the approach. For instance, transcriptional sites would of course also be associated with more open chromatin and higher MNase accessibility in general. It was not clear from the methods how the authors normalize their data to MNase digestion biases similar to how typical Hi-C data is normalized to the density and location of site-specific restriction digestion. It would also be good to know if frequent exceptions to this correlation exist wherein Micro-C is detecting new loops not associated with high levels of open chromatin or transcription. If so, do these also share the same features as mentioned above. Are the new loops dependent on transcription as could be determined via polII inhibition or mutational analysis? Surely, these resources are available for mESCs?

Response: The reviewer questioned 1) if the fine-scale chromatin dots detected in Micro-C are due to higher accessibility of micrococcal nuclease (MNase) to promoters and enhancers, which may bias the Micro-C signal towards these regions; 2) how Micro-C data are normalized; 3) how transcription affects chromatin loops. We here cited studies and performed additional analyses to address these questions below.

MNase digests chromatin uniformly

Please see response to Reviewer #1 above (point 1.1)

Matrix balancing implicitly corrects systemic biases

The 3C-based assays involve a series of biochemical reactions that may introduce noise to the data analysis and the final results. Such noise must be eliminated before interpreting the data. Early normalization methods for Hi-C data focused on explicit factors causing noise. Factors like sequence uniqueness, GC content, uneven distribution of restriction sites across the genome may introduce biases into the 3C-based analysis (Yaffe and Tanay, 2011). As a solution to these issues, the Mirny and Aiden labs proposed using matrix balancing methods to handle all noise sources "implicitly" (Imakaev et al., 2012; Rao et al., 2014). Two fundamental assumptions underlie the methods: 1)

visibility across all genomic regions should be equal, and 2) all Hi-C biases are one-dimensional and factorizable. A balanced matrix should allow equal visibility to any genomic locus regardless of system biases, that is, a normalized matrix whose rows and columns sum to the same amount. In Micro-C/Hi-C analysis, three primary matrix balancing algorithms are commonly used, including "the square root of vanilla coverage (VC)", "Knight-Ruiz (KR)", and "Iterative Correction and Eigenvector decomposition (ICE)". The idea of VC is similar to coverage normalization as it divides each matrix element by its row sum and column sum to eliminate different sequencing coverage of each locus. ICE and KR repeat the VC process until all rows and cols sum to the same value and converge. Results from KR and ICE are typically nearly identical and highly reproducible. Additionally, we often performed distance normalization (e.g., observed/expected) to remove the bias caused by 1D distance-dependent interactions in 3C-based data.

Micro-C is not biased toward chromatin accessible regions

Please see response to Reviewer #1 above (**point 1.1**)

Clarification of Micro-C loop analysis

The new loop callers Mustache (Ardakany et al., 2020) and Chromosight (Matthey-Doret et al., 2020b) reported a technical advance in computer vision that detects blob-shaped organizational features in Hi-C/Micro-C contact maps with a higher sensitivity than previous methods. Even with Hi-C data, the algorithms can identify much more loops than HICCUPS (the method most widely used in the field) (**Fig. 14**, left panel). Both studies also confirmed detecting more loops in Micro-C compared with Hi-C (**Fig. 14**, right panel). Similarly, in our previous study, we demonstrated that Micro-C detects ~5-fold more loops than Hi-C when using HICCUPS. We confirmed that the results remain unchanged regardless of which loop callers are used or excluding these newly identified loops by Mustache/Chromosight. Furthermore, we did not use the *de novo* loops called by Mustache or Chromosight for the analysis of TAD boundary crossing. Instead, we quantified the paired E-P interactions with the Micro-C data at 2-kb or 4-kb resolution. The figure legend is now revised to clarify our analysis.

(**Fig. 14**)

The effect of Pol II inhibition on E-P interactions

For a detailed analysis of Pol II inhibition on E-P interactions, please see our previous study in Hsieh et al. 2020 and **Fig. 15** below. In brief, treatment with Pol II inhibitors triptolide or flavopiridol moderately reduces E-P or P-P interactions and substantially attenuates chromatin stripes associated with promoter and enhancer loci.

(**Fig. 15**)

But most concerning is that they find that many (29-30%) of these loops traverse a TAD boundary. While it is true that boundaries have been found to be variable and TAD insulation is not strict, it is hard to imagine that stable or frequent E-P interactions traversing a boundary would represent a functional interaction. So far, a large body of literature point to the fact that cognate promoters and enhancers tend to be within the same TAD. For instance, this reviewer is not aware of any developmental gene whose functionally validated enhancer has been mapped outside of the same TAD and if so this is an extremely rare event. And recent estimates from eQTL studies suggest that >95% of variants are also located within the same TAD as the associated gene (Delaneau et al., 2019). Similar conclusions have been made from perturbation studies in mESCs (Sun et al., 2019). How do the authors reconcile their results with these findings?

Response: Our data reporting that ~30% of loops cross a TAD boundary are largely consistent with previous studies. First, a recent CRISPRi-based enhancer screening reported that ~30% of E-P interactions do not fall into the same TAD (Gasperini et al., 2019). Second, promoter Capture Hi-C experiments (Sun et al. 2019) also observed that ~24% of promoter-centered interactions are not constrained within the same TAD, which aligns with our findings. Third, emerging evidence by single-cell Hi-C and super-resolution imaging indicates that TADs are dynamic chromatin structures that constantly form and dissolve (Bintu et al., 2018; Cattoni et al., 2017; Giorgetti et al., 2014; Nagano et al., 2013; Stevens et al., 2017). Associations between regions in the same TAD are not necessarily pervasive and interactions are common between neighboring TADs (Finn et al., 2019; Mateo et al., 2019). Cohesin can traverse across boundaries frequently and mediate domain intermingling (Luppino et al., 2020; Szabo et al., 2020). Fourth, the latest evidence indicates that the perturbations of TAD boundaries around the Sox2 and HoxD regions are mostly harmless to embryo development (Chakraborty et al., 2022; Rodríguez-Carballo et al., 2020), suggesting that high affinity enhancer-promoter interactions can overcome structural barriers. Finally, since the eQTL analysis in Delaneau et al., is based on the correlation of a set of CHIP-seq data for histone modifications, the cis-regulatory domains (CRDs) identified in this study most likely represent local chromatin domains segregated by chromatin states (e.g., active or inactive regions) rather than long-range E-P loops. It is uncertain whether this computational structure fully reflects 3D genome structures in situ. A recent study indicates that analysis of chromatin loops outperforms eQTL in explaining neurological GWAS results, suggesting there may be much ampler information in high-resolution 3D genome maps than in eQTL analyses (Lu et al., 2020). Together, our findings of cross-TAD interactions are largely consistent with other studies, arguing that communication between enhancers and their cognate genes in another domain is not uncommon.

Lastly, many studies have shown, from sequencing and imaging-based methods, that loss of cohesin or WAPL changes chromatin compaction both within and between TADs. Even indirectly, this would be expected to alter promoter-enhancer proximity. And yet, the authors show no change in number or strength of their newly identified interactions in the absence of several architectural proteins.

Given the above, it seems extremely important that the authors validate their new looping interactions through

orthogonal methodology such as 3C, Capture-C, or 5C that each would result in comparable resolutions to that of Micro-C. In addition, if the authors were to bin their data to artificially reduce their resolution to that comparable to Hi-C, would these interactions disappear upon reanalysis? At the very least, this would show that they are dependent on the unprecedented high resolution achieved by Micro-C.

Response: We note that recent studies using orthogonal methods such as HiChIP (Kubo et al., 2021) and Capture Hi-C (Thiecke et al., 2020) have reported similar results to ours. Kubo et al found that majority of chromatin interactions between enhancers and promoters remain unchanged (~99.5%) after 48-hour CTCF depletion in mESCs. Thiecke et al also reported ~30 - 40% of promoter-anchored contacts are maintained after RAD21 or CTCF depletion in HeLa cells, albeit to a lesser extent perhaps due to lower resolution by using 6-base cutter or different cell lines. Thus, consistent results from orthogonal methods support our findings that E-P interactions are largely maintained after acute depletion of architectural proteins (see an example in **Fig. 16**, top panel).

As shown in **Fig. 5-6** above and **Fig. 16** below, Micro-C and its variant Micro-Capture-C (MCC) recapitulate the results seen in Hi-C and Capture-C (black arrows) while further resolving finer-scale chromatin interactions (e.g., *Myc-Pvt1* interactions and intragenic interactions around *Nfix* gene, blue arrows). These results indicate that the finer-scale loops are uniquely detected in Micro-C.

(Fig. 16)

Other comments:

1. It has recently become apparent that AID tagged versions of proteins, including Rad21, can be ‘leaky’ and show reduced activity or levels compared to their wildtype versions. How does the expression profile of their tagged cell lines compare to that of parental mESCs? Is it possible the small changes upon auxin treatment are

due to an already perturbed system prior to treatment?

Response: We quantified the levels of basal degradation for the AID tagged proteins and found that CTCF-AID has ~40% of basal degradation and RAD21-AID and YY1-mIAA7 have ~20% degradation (Fig. 17, left panel). However, the basal degradation has very minimal impact on gene expression, with high correlations between wild-type, parental cells, and all the untreated conditions (Pearson's $r > 0.95$ in RNA-seq and > 0.91 in mNET-seq) (Fig. 17, right panel). Only ~5 – 60 genes significantly change ($P_{adj} < 0.01$) among the untreated cells. We thus conclude that although some basal degradation occurs after degenon tagging, it does not affect gene expression profiles of the individual clones significantly. We included the results in the manuscript **Extended Data Fig. 2** and **5**.

(Fig. 17)

Quantification of basal degradation for the AID-tagged proteins

RNA-seq and mNET-seq show the basal degradation has minimal effect on transcription between untreated cells

2. The conclusion that cohesin might act to recruit transcription factors in general based solely on YY1 behavior seems like a bit of a leap. The authors might want to tone down this conclusion.

Response: We probed the nuclear dynamics of two additional transcription factors, KLF4 and SOX2,

by spaSPT and found that their bound fractions were reduced by ~20% 3 hours after cohesin degradation (**Extended Data Fig. 9g**). Furthermore, while our paper was under review an independent study from Gordon Hager's group (Rinaldi et al., 2022) found precisely the same thing: for the TF glucocorticoid receptor, cohesin depletion leads to decreased chromatin binding. Thus, while we agree with the reviewer that our model was too strong a leap based on a single TF (YY1) in the original submission, since we now have evidence for 4 TFs (YY1, KLF4, SOX2, GR) in two different cell types (mESCs, human cancer cells) and from two different labs, we now believe that the evidence is strong enough to invoke a role for cohesin in facilitating TF binding.

Reviewer #3:

CTCF, cohesin complex, and YY1 have been proposed to be key players shaping the chromatin architecture in mammalian cells, with CTCF and the cohesin complex involved in mediating long-range chromatin loops between convergent CTCF binding sites through a loop extrusion process, while YY1 shown to mediate enhancer-promoter contacts by virtue of its DNA binding to both classes of elements and its dimerization potential. Additionally, cohesin has also been suggested to mediate enhancer-promoter interactions due to its association with mediator complex and loading by Nipbl protein, which binds to both promoters and enhancers. While abundant literature supports a role for CTCF and cohesin in chromatin organization especially in defining topologically associating domains (TADs) and long-distance chromatin loops, their function in chromatin contacts between enhancers and promoters is less clear and has been the subject of intense debate in recent years. Also under intense debate is whether chromatin interactions between enhancers and promoters actively contribute to enhancer dependent gene activation. Recent studies using auxin inducible degrons to acutely deplete CTCF or cohesin subunits showed that CTCF and cohesin are dispensable for transcription of most genes in a variety of cell systems tested. By contrast, acute depletion of YY1 has been shown to lead to dramatic changes in gene expression in mouse ES cells. These previous studies raised questions about what exactly is the role that CTCF and cohesin play in enhancer-promoter communication and gene activation, and whether different proteins (such as YY1) are involved enhancer-promoter contacts and enhancer dependent transcription from target genes.

The manuscript by Hsieh et al. attempts to address these two questions and to provide clarity on the role of CTCF, cohesin and YY1 in enhancer-promoter contacts and the role of enhancer-promoter loops in gene activation. Like previous studies, the authors used the auxin inducible degron system to deplete CTCF, Rad21 subunit of cohesin, cohesin unloader protein WAPL and YY1 protein, in a mouse embryonic stem cell line. But in contrast to the previous work, the experimental design in the current report involved a shorter protein depletion time (3 hours vs. 24hr or longer), which could avoid the secondary effects during long-term protein depletion. Additionally, the authors used Micro-C, a variant of Hi-C technique, to achieve higher resolution (a few hundred base pairs versus 5- or 10-kb resolution) and sensitivity of loop detection especially between enhancers and promoters. Furthermore, the authors employed mNET-seq to measure changes in nascent transcript levels genome-wide after protein depletion. Their main findings are: (1) depletion of these proteins did not cause significant changes in enhancer-promoter interactions, even for YY1; and (2) depletion of these proteins resulted in minimal changes in gene expression. The authors also used live cell single particle imaging techniques to study the dynamics of YY1 binding to chromatin upon depletion of cohesin, and found an unexpected but modest role for cohesin to retain YY1 on chromatin. Based on these observations, the author concluded that “neither transcriptional condensates, CTCF, cohesin, WAPL, nor YY1 are generally required for the short-term maintenance of E-P interactions and the subsequent expression of most genes after acute depletion and loss of function” (Line 494). To reconcile with previous genetic evidence implicating CTCF and cohesin’s role in regulating promoter-enhancer interactions and transcription at some genes, the authors further proposed a “time-buffering model”, “where architectural proteins generally contribute to the establishment, but not the short-term maintenance, of E-P interactions and gene expression” (Line 548).

General assessment:

Overall, the current work reports several very interesting observations related to the role of CTCF, cohesin and YY1 in enhancer promoter interactions and transcription, with some in direct contradiction with previous studies, and others confirmatory. An important contribution of this work lies in the use of a well-controlled experimental system and high precision technical approaches. The genetically engineered cell lines to enable acute depletion of CTCF, YY1, and cohesin complex in cells are valuable tools to the field. The micro-C and mNET-seq datasets along with other genomic datasets in this experimental model system also provide a valuable resource for the community to investigate the role of these proteins in chromatin organization and gene regulation. Surprisingly, the authors found that cohesin depletion did not disrupt enhancer-promoter contacts in general, and that YY1 loss had no significant impact on both enhancer promoter contacts and gene expression, which is in direct contradiction to Weintraub et al. (Cell 2017, ref #62). These new results are interesting and certainly raise new questions.

On the other hand, the experimental evidence provided in the current work does not fully support the sweeping conclusion that CTCF and cohesin play no role in the maintenance of enhancer promoter interactions (as the title implies), nor the statement about transcriptional condensates (see specific comments below for details). Further, some of the findings described in the current work are not new. For example, the observations that CTCF was largely dispensable for enhancer-promoter interactions and general transcription are consistent with previous studies (for example, Nora et al. Cell 2017; Kubo et al. NSMB 2021, which the authors cited as ref #10 and #37, respectively). The observation that cohesin loss resulted in minimal changes in gene expression was also reported before (for example, Rao et al. Cell 2017, ref #11 in the manuscript). Therefore, the degree by which this work advances our understanding of the maintenance of enhancer-promoter interactions and their function is currently unclear.

Response: We thank the reviewer for a thoughtful summary. We highlight the key contributions in this manuscript that are distinct from previous studies:

1. **Effects on fine-scale E-P interactions:** We used Micro-C to identify several E-P interactions that were technically impossible to detect in previous studies. In addition, some of these previously undetectable loops have been functionally validated by CRISPR-based studies (Gasperini et al., 2019; Moorthy et al., 2017).
2. **Immediate effects on nascent transcription:** We determined the immediate short-term effect of CTCF, cohesin, WAPL, and YY1 on E-P interactions and nascent transcription. Most prior studies focused on either later time points or total RNA quantification, both prone to reflect secondary effects. Specifically, previous observations were a mix of 1) 3D genome reorganization in response to protein depletion; 2) secondary transcriptional effects; 3) total mRNA pool before and after protein depletion; and 4) secondary effects of post-transcriptional RNA processing and buffering.
3. **Transcriptional establishment vs. maintenance:** We comprehensively dissect YY1 dynamics using single-particle imaging and uncover a putative role of cohesin in regulating transcription factor binding for transcriptional establishment, which is distinct from its known function in genome folding.
4. **Time-buffering model:** By combining high-resolution Micro-C maps with nascent transcription profiles, we establish a time-buffering model that reconciles our and previous studies and explains a possible mechanism of 3D genome and transcriptional regulation.

Main comments:

1. The experimental evidence presented does not fully support the authors' assertion that enhancer-promoter (E-P) interactions are unaffected after depletion of CTCF or cohesin. A closer inspection of Figure 3C suggests that E-P interactions were generally weakened after depletion of Rad21 and to a lesser degree CTCF - the center of the data cloud representing E-P interactions is below the diagonal line, and this trend is not obvious for P-P interactions. Therefore, one could say that there is a general and weak effect for cohesin loss on E-P interactions, consistent with residual levels of cohesin retained on chromatin after short term depletion.

Response: We revised the section with a quantitative description as follows "*To further validate this, we specifically quantified the strength of loops that are anchored by E-P and P-P 5 kb to 2 Mb apart. Remarkably, acute depletion of CTCF and cohesin has only a limited impact on the E-P and P-P loops, with ~80% of E-P contacts and 90% of P-P contacts remaining unaltered (Fig. 3d). Despite being less drastic than for cohesin loops (Fig. 2f), E-P interactions appear to be slightly weakened globally, deviating from the midpoint line (i.e., the average contact intensity of unchanged loops decreased by ~2.4% and ~8.0% upon CTCF and cohesin depletion, respectively). This trend is in accordance with the mild reduction of E-P interaction strength observed upon cohesin depletion by tiled Micro-C at selected loci in mESCs (Aljahani et al., 2022). On the other hand, P-P loops are largely insensitive to CTCF or cohesin depletion (Fig. 3d). Similarly, a pile-up analysis shows that when CTCF or cohesin is depleted, the contact intensity is decreased by ~22.2% or ~29.1% for E-P loops but only ~4.1% or ~8.4% for P-P loops (Fig. 3e and Extended Data Fig. 6c-d).*"

2. Similarly, the assessment of YY1's effect on chromatin interactions was also problematic. The authors in Figure 4i examined changes of chromatin interactions on cohesin mediated loops (strong loops), and found little

change. But what about E-P interactions? It would be important for the authors to provide a similar plot for the YY1 depleted cells as in Figure 4C for CTCF and cohesin depletion. The authors did report ~800 loops changed after YY1 depletion (Line 320), but Figure 4e indicated only 436 as “down” loops. So it is unclear how the authors reached the conclusion that loss of YY1 did not affect chromatin interactions between promoters and enhancers, especially when only 50% of YY1 binding sites showed significantly reduced binding after YY1 depletion (Line 308).

Response: We would like to clarify a few points below:

1. In **Fig. 4d** and **Extended Data Fig. 7m** (previous **Fig. 4i**), we already showed that YY1 depletion does not have a global effect on YY1-associated loops, E-P, and P-P interactions. We also showed the minimal effect of YY1 degradation on compartments, TADs, and cohesin loops.
2. We included a scatter plot in **Fig. 4e** showing that ~374 up-regulated loops and ~436 down-regulated loops (changed loops = ~810/75190; ~1.08%) are detected after 3 hours of YY1 depletion. We conclude that, since ~99% of chromatin loops are insensitive to YY1 loss, YY1 does not have a major impact on chromatin loops.
3. In our new analysis of YY1 degradation, we found that YY1 is nearly completely degraded within 3 hours, as confirmed by Western blot, flow cytometry, and imaging, and that 86% of peaks were significantly reduced by ChIP-seq (lost peaks = 29379/34342) (**Extended Data Fig. 7**).

3. A major feature of the experimental design is the short duration of protein depletion (3 hours). This is also a potential weakness, as it is apparent that 3-hour IAA treatment did not fully deplete the proteins. Residual ChIP-seq signal for CTCF and cohesin were readily detectable on chromatin at CTCF peaks (Ext Fig 2f) and E-P & P-P loop anchors (Ext Fig 4e). Same can be said for YY1 (Figure 4b, Ext Fig 5C). It is therefore hard to rule out the possibility that residual proteins bound to chromatin might contribute to maintenance of chromatin interactions in the short-term depletion.

Response: We performed biochemical fractionation and flow cytometry analysis to determine the fraction of proteins remaining on chromatin after 3 hours of depletion (**Fig. 18**; please see (Gabriele et al., 2021), Fig. S4 and **Extended Data Fig. 2** and **7** for full details). In brief, our results showed that CTCF and RAD21 are nearly completely degraded 3 hours after depletion and that only ~8% and 13% of residual WAPL and YY1 remain associated with chromatin, respectively. Furthermore, we have previously measured the average number of CTCF, RAD21, and YY1 molecules in a single cell (Cattoglio et al., 2019; Holzmänn et al., 2019). Our analysis suggests that if our degradation reaches ~95% depletion for CTCF and RAD21 (due to the detection limit of western blot) and ~87% depletion for YY1, at most only ~787 CTCF, ~722 RAD21, and ~700 YY1 proteins would remain bound to chromatin (per chromosome). A few hundreds of residual architectural proteins are very unlikely sufficient to maintain >50,000 E-P loops and transcription. Also, since only hundreds of proteins remain after depletion and all CTCF and cohesin loops are nearly completely abolished, it is hard to conceive that the residual proteins will specifically maintain E-P interactions but not CTCF/cohesin loops. If any, the effect of residual proteins on our results is minimal.

(**Fig. 18**)

A Protein depletion quantified by BFP imaging following IAA treatment

B Biochemical fractionation

4. The authors concluded that “transcriptional condensates” are not generally required for “short-term maintenance of E-P interactions” (Line 493), by drawing on previous literature (Ref #57, 58 and 88), without providing any new evidence in the current work. This statement therefore needs to be revised to better reflect this fact.

Response: We made the hypothesis model based on the large number of studies available on transcriptional condensates and gene regulation. Also, this topic is only examined in the discussion section. We now included a note that this model is based on previous literature rather than on our study.

5. It is interesting that cohesin loss results in reduced population of immobile and presumably chromatin bound YY1, but this evidence alone is insufficient to lead to the model that cohesin facilitates target search by transcription factors.

Response: We probed the nuclear dynamics of two additional transcription factors, KLF4 and SOX2, by spaSPT and found that their bound fractions were reduced by ~20% 3 hours after cohesin degradation (**Extended Data Fig. 9g**). Furthermore, while our paper was under review an independent study from Gordon Hager’s group (Rinaldi et al., 2022) found precisely the same thing: for the TF glucocorticoid receptor, cohesin depletion leads to decreased chromatin binding. Thus, while we agree with the reviewer that our model was too strong a leap based on a single TF (YY1) in the original submission, since we now have evidence for 4 TFs (YY1, KLF4, SOX2, GR) in two different cell types (mESCs, human cancer cells) and from two different labs, we now believe that the evidence is strong enough to invoke a role for cohesin in facilitating TF binding.

Minor comments:

1. With multiple loop callers, authors show that micro-C can identify more fine range E-P and P-P interactions compared to Hi-C. What percent of the E-P, P-P loops called by previous Hi-C data are also detected by Micro-C and vice versa? What is the overlap? What about with HiChIP /Plac-Seq data? This analysis is important to allow the readers to assess the specificity and sensitivity of the data.

Response: Consistent with a previous report (Ardakany et al., 2020), Micro-C recapitulates at least 80% of finer-scale loops detected by Hi-C (Bonev et al, 3.3B reads) with either Mustache or Hiccups loop callers (res=1kb) (**Fig. 19**, left panel). Micro-C also recapitulates >70% of HiChIP/PLAC-seq contacts for CTCF, KLF4, and OCT4 (*q-value* < 0.01), ~40% for SMC1A and H3K4me3, and ~18%

for H3K27ac (q -value < 0.01) (**Fig. 19**, right panel). We suspect that the lower overlap ratio might be due to non-specific enrichment in HiChIP/PLAC-seq that does not occur as significant contacts in Micro-C/Hi-C but this needs further investigation. Together, our Micro-C analysis shows high specificity and sensitivity to detect chromatin loops.

(Fig. 19)

2. Titles for the scatterplots missing (e.g., Ext Fig 4c)

Response: Corrected.

3. Extended Fig 4d. A subset of downregulated loop anchors in WAPL depletion cells seem to show more bias towards enhancer and promoter regions. Do these regions show any differences in Rad21 binding?

Response: Chromatin loops that are sensitive to WAPL depletion show higher Cohesin occupancy at the loop anchor in the normal condition but only the up-regulated loops exhibit an increase in Cohesin occupancy after WAPL depletion (**Fig. 20**).

(Fig. 20)

4. Upon depletion of YY1 itself, there is a modest decrease of Rad21 binding. Which Rad21-bound regions show reduction upon YY1 depletion, what are their characteristics? Do they correspond to loop anchors, or show any change in loop strength?

Response: Lost RAD21 peaks are primarily found in insulators, active promoters, and strong enhancers (**Fig. 21A-B**), where protein binding motifs for CTCF, HAND2, TCF3, and ZNFs are

significantly enriched in DNA sequences (**Fig. 21C**). In addition, over 60% of these RAD21 peaks are at loop anchors, but their loss does not have a major impact on chromatin loops (**Fig. 21D**).

(Fig. 21)

Reviewer #4:

In this paper, Hsieh et al. leverage high resolution Micro-C and nascent RNA (mNET-seq) data to understand the roles played by CTCF, cohesin, and YY1 in the maintenance of enhancer-promoter interactions. The authors use mESC cell lines in which CTCF, WAPL, cohesin, and YY1 are tagged for rapid (3h) degradation. The authors argue that the marginal effect of CTCF and cohesin on overall gene expression level, as observed previously in the literature, is explained by cis-regulatory loops between enhancers and promoters being relatively insensitive to the depletion of these architectural proteins. The authors then show through similar experiments that the depletion of YY1, previously suggested as an important factor for enhancer-promoter loops, also had only marginal effects on enhancer-promoter loops and transcription. Finally, using a combination of confocal microscopy and single molecule imaging, the authors characterize the dynamics of YY1 binding to chromatin and show that the depletion of RAD21, but not CTCF significantly reduces YY1-chromatin binding by increasing its searching time. This paper examines an important aspect of the relationship between chromatin folding and transcription. It is a well written paper; experiments are well justified and expertly executed. The results are timely. However, while the data is comprehensive and solid, I do have some major concerns about data analysis and interpretation that need to be addressed prior to publication.

Major comments:

(1)* The authors argue that 80-90% of loop interactions between enhancers and promoters (or promoters and promoters) are ‘unaltered’ by depletion of cohesin or CTCF. I am not convinced that the author’s data supports this conclusion. Setting aside the ~20% of enhancer-promoter interactions that the author’s analysis turns up as significantly reduced, there does appear to be a small reduction in contact frequency more broadly. Carefully examining the scatterplots in Fig. 3C shows that the bulk of enhancer promoter contacts are comfortably below the 0, 1 line, indicating lower contact frequency in AID than control cells. If this shift is not indeed explained by normalization issues (see points below), I think the paper would improve if the authors provided a more nuanced presentation of these results, acknowledging the shift that is clearly present in their data.

Response: We revised the section with a quantitative description as follows “*To further validate this, we specifically quantified the strength of loops that are anchored by E-P and P-P 5 kb to 2 Mb apart. Remarkably, acute depletion of CTCF and cohesin has only a limited impact on the E-P and P-P loops, with ~80% of E-P contacts and 90% of P-P contacts remaining unaltered (Fig. 3d). Despite being less drastic than for cohesin loops (Fig. 2f), E-P interactions appear to be slightly weakened globally, deviating from the midpoint line (i.e., the average contact intensity of unchanged loops decreased by ~2.4% and ~8.0% upon CTCF and cohesin depletion, respectively). This trend is in accordance with the mild reduction of E-P interaction strength observed upon cohesin depletion by tiled Micro-C at selected loci in mESCs (Aljahani et al., 2022). On the other hand, P-P loops are largely insensitive to CTCF or cohesin depletion (Fig. 3d). Similarly, a pile-up analysis shows that when CTCF or cohesin is depleted, the contact intensity is decreased by ~22.2% or ~29.1% for E-P loops but only ~4.1% or ~8.4% for P-P loops (Fig. 3e and Extended Data Fig. 6c-d).*”

(2)* All of the Micro-C analysis in this manuscript compares contact frequency between different conditions with very large, global differences in contact frequency. As with any other sequencing assay, contacts that are lost from (say) CTCF-CTCF loop anchors or TADs upon CTCF or Rad21 depletion will be redistributed to other loci. In 1D data (e.g., mNET-seq/ ChIP-seq/ etc.), spike-in controls can be used to normalize between different samples in absolute terms, allowing the discovery of truly global changes. While I recognize there is no easy analog of a spike-in for a Micro-C experiment, I would like the authors to discuss the importance of normalization, and the ways in which normalization issues could impact the author’s interpretations.

Response: As described above in response to Reviewer #2, the 3C-based assays involve a series of biochemical reactions that may introduce noise to the data analysis and the final results. Such noise must be eliminated before interpreting the data. Early normalization methods for Hi-C data focused on explicit factors causing noise. Factors like sequence uniqueness, GC content, uneven distribution of restriction sites across the genome may introduce biases into the 3C-based analysis (Yaffe and Tanay, 2011). As a solution to these issues, the Mirny and Aiden labs proposed using matrix balancing methods to handle all noise sources “implicitly” (Imakaev et al., 2012; Rao et al., 2014). Two

fundamental assumptions underlie the methods: 1) visibility across all genomic regions should be equal, and 2) all Hi-C biases are one-dimensional and factorizable. A balanced matrix should allow equal visibility to any genomic locus regardless of system biases, that is, a normalized matrix whose rows and columns sum to the same amount. In Micro-C/Hi-C analysis, three primary matrix balancing algorithms are commonly used, including "the square root of vanilla coverage (VC)", "Knight-Ruiz (KR)", and "Iterative Correction and Eigenvector decomposition (ICE)". The idea of VC is similar to coverage normalization as it divides each matrix element by its row sum and column sum to eliminate different sequencing coverage of each locus. ICE and KR repeat the VC process until all rows and cols sum to the same value and converge. Results from KR and ICE are typically nearly identical and highly reproducible. Additionally, we often performed distance normalization (e.g., observed/expected) to remove the bias caused by 1D distance-dependent interactions in 3C-based data.

To test whether these normalization approaches can address global changes in genome organization and correct systemic biases, we computationally injected 3x more long-range interactions (> 2Mb) and inter-chromosomal interactions into wild-type Micro-C data (**Fig. 22**, top panel). In this case, the artificially injected reads generated an extreme case of reads being redistributed to longer ranges (**Fig. 22**, top panel). Importantly, the results of compartment strength, TADs, cohesin loops, and E-P/P-P loops are not affected by the global changes caused by artificial reads (**Fig. 22**, bottom panels). We therefore conclude that matrix balancing and distance normalization can reveal the bona fide chromatin structures and differential chromatin interactions regardless of systemic biases and global changes.

(Fig. 22)

(3)* MNase, depending on the digestion conditions, does not cut uniformly across the entire genome, but rather it has specific biases for accessible regions. Any change in the distribution of MNase cut frequency near promoter/ enhancer regions between wild-type and CTCF, YY1, or Rad21 depletion could alter the contact frequency observed in contact maps. The authors should evaluate whether there are differences in MNase cutting efficiency between different conditions by examining the 1D signal in their Micro-C libraries.

Response: Please see response to Reviewer #1 above (**point 1.1**). Additionally, MNase-seq analysis suggests that CTCF is critical for nucleosome positioning around the CTCF binding motifs but does not appear to drastically change overall nucleosome occupancies (MNase cutting efficiency) after CTCF depletion (**Fig. 23**, left) (Owens et al., 2019). We analyzed averaged Micro-C reads per bin (10 kb) and found that CTCF, RAD21, and YY1 depletion do not change the read distribution. Overall, our analysis suggests that MNase digestion efficiency largely remains within a similar range under both untreated and depletion conditions.

(Fig. 23)

(4)* It generally is not clear whether the authors measure transcription after removing the pause peak, focusing on expression in the gene body (which most directly correlates with mRNA), or whether paused Pol II is included. As the different stages of transcription initiation and pause might have different relationships with contact frequency, it is crucial to distinguish clearly between them in their presentation.

Response: We only include the mNET-seq signal at the gene body for differential expression gene analysis. We added a detailed description in the Methods and the Figure legend.

(5)* Data showing that RAD21 serves as a “platform” for YY1, or possibly other transcription factors, is extremely weak. I am convinced by the data showing that it increases search times (although I will defer to an expert on the rigour of the microscopy experiments). How do the authors feel about an alternative model in which the act of loop extrusion acts to stir the nucleoplasm, resulting in higher diffusion and faster search times? There are likely other indirect explanations as well, and I would encourage the authors to consider these mechanisms before jumping right to IDR-IDR interactions. Related to this point - unless the authors can directly show that protein-protein interactions are necessary for the effect of RAD21 depletion on YY1 search time, I would encourage them to drop the use of the word “platform” and other terms that imply a direct interaction.

Response: We probed the nuclear dynamics of two additional transcription factors, KLF4 and SOX2, by spaSPT and found that their bound fractions were reduced by ~20% 3 hours after cohesin degradation (**Extended Data Fig. 9g**). Furthermore, while our paper was under review an independent study from Gordon Hager’s group (Rinaldi et al., 2022) found precisely the same thing: for the TF glucocorticoid receptor, cohesin depletion leads to decreased chromatin binding. Thus, while we agree with the reviewer that our model was too strong a leap based on a single TF (YY1) in the original submission, since we now have evidence for 4 TFs (YY1, KLF4, SOX2, GR) in two

different cell types (mESCs, human cancer cells) and from two different labs, we now believe that the evidence is strong enough to invoke a role for cohesin in facilitating TF binding.

We do agree with the reviewer that “platform” is too mechanistically suggestive without more mechanistic evidence and we have therefore removed the word “platform” from our paper.

Finally, the reviewer suggests that “*How do the authors feel about an alternative model in which the act of loop extrusion acts to stir the nucleoplasm, resulting in higher diffusion and faster search times?*”. However, loop extrusion has the opposite effect – it makes chromatin diffusion slower. Indeed, two recent studies (Gabriele et al., 2022; Mach et al., 2022) have both shown that cohesin depletion leads to a ~2-4-fold increase in chromatin diffusion dynamics. However, faster chromatin motion is extremely unlikely to noticeably accelerate the TF target search.

To understand why consider the Smoluchowski equation (Smoluchowski, Versuch einer mathematischen Theorie der Koagulationskinetik kolloider Losungen, 1916) for diffusion-limited bimolecular rate constants:

$$k_{ON} = 4\pi(D_{TF} + D_{DNA})(R_{TF} + R_{DNA})b$$

Where D is the diffusion coefficient, R is the size of the TF or DNA binding site, and b is the cross section. Using typical diffusion coefficients of $D_{TF} = 3 \frac{\mu\text{m}^2}{\text{s}}$ and $D_{DNA} = 0.01 \frac{\mu\text{m}^2}{\text{s}}$, we can calculate how much TF search would be expected to be speed up by a 4-fold increase in chromatin diffusion (as observed during cohesin depletion):

$$\frac{k_{ON,\Delta\text{RAD21}}}{k_{ON,UT}} = \frac{4\pi(D_{TF} + D_{DNA})(R_{TF} + R_{DNA})b}{4\pi(D_{TF} + D_{DNA})(R_{TF} + R_{DNA})b} = \frac{D_{TF} + D_{DNA}}{D_{TF} + D_{DNA}} = \frac{3 + 0.04 \frac{\mu\text{m}^2}{\text{s}}}{3 + 0.01 \frac{\mu\text{m}^2}{\text{s}}} = 1.01$$

In other words, the TF search time would be expected to increase by 1% due to cohesin depletion based on chemical kinetics. Instead, we observe the opposite – the 2.2-fold slower TF search for YY1. Thus, both the magnitude (1% increase vs. 54% decrease) and the direction (chemical kinetic predicts faster search, we observe much slower search) are inconsistent with the model proposed by the reviewer. In conclusion, we feel confident in ruling out changes in chromatin diffusion due to cohesin depletion as a possible explanation for the reduction in TF search kinetics.

Minor comments:

* In figure 1d, the axes should be better described. Additionally, in P-P contacts, clarify whether the data points represent an average of the two promoters.

Response: Corrected.

* Throughout the manuscript, add axes description to the APA heatmaps explaining, for example, which of the rows/columns represent promoters/enhancers.

Response: Annotations are updated.

* Please add transcriptional orientation (i.e. what is the direction of the transcription unit) to the APA heatmaps.

Response: Promoters used for APA analysis consist of both directions.

* A clearer explanation on how anchors were chosen and defined as promoters and enhancers would also help readers interpret the data. What dataset was used to define TSSs or whether it was accomplished by the MNase data should be clarified.

Response: We classified chromatin/genome features into 12 states by using ChromHMM (Ernst and Kellis, 2012) (**Extended Data Fig. 1d**) with Gro-seq, ATAC-seq, and ChIP-seq data for histone marks and transcription factors. We have a detailed description in the Methods “*Definition of chromatin states and structure observed by Micro-C*”.

* Please use the acronym polyadenylation cleavage site (PAS), rather than TES, in figure 1C. I assume TES stands for transcription end site, but presumably the point used is the end of a gene annotation. Transcription continues past the end of the PAS, as is clearly shown in the authors' plots.

Response: Corrected.

* Some interesting differences are shown between P-P and E-P pairs (Fig. 1D and 3H) but are ignored in the text. Are these associated with the different levels in initiation and pausing?

Response: We thank the reviewer for the interesting observation. Our analysis shows that nascent transcription around the anchors of P-P and E-P loops exhibits comparable activity, while the signals around enhancers and cohesin binding sites are much weaker (**Fig. 24**, left). The distinct correlations of E-P and P-P with loop strength (**Fig. 1D** in the manuscript) and the response to CTCF/cohesin depletion (**Fig. 3H** in the manuscript) are most likely to correlate with 3D genome organization rather than with transcription activity at the promoter. Additionally, we found that, in contrast to gene body signals, promoter-proximal signals do not correlate with productive transcription. Therefore, we do not intend to focus on it in the manuscript (**Fig. 24**, right).

(Fig. 24)

* Figure 3b shows that loops from up/ down/ unchanged groups are enriched in functional elements involved in transcriptional regulation. Taken along with the pattern shown in 3a, this further suggest that the effect of CTCF and RAD21 depletions on cis-regulatory elements looping is underestimated by the authors.

Response: We have revised the description of this section (see above).

Reference

- Aljahani, A., Hua, P., Karpinska, M.A., Quililan, K., Davies, J.O.J., and Oudelaar, A.M. (2022). Analysis of sub-kilobase chromatin topology reveals nano-scale regulatory interactions with variable dependence on cohesin and CTCF. *Nat Commun* *13*, 2139. <https://doi.org/10.1038/s41467-022-29696-5>.
- Ardakany, A.R., Gezer, H.T., Lonardi, S., and Ay, F. (2020). Mustache: multi-scale detection of chromatin loops from Hi-C and Micro-C maps using scale-space representation. *Genome Biol* *21*, 256. <https://doi.org/10.1186/s13059-020-02167-0>.
- Arnold, M., Bressin, A., Jasnovidova, O., Meierhofer, D., and Mayer, A. (2021). A BRD4-mediated elongation control point primes transcribing RNA polymerase II for 3'-processing and termination. *Mol Cell* *81*, 3589-3603.e13. <https://doi.org/10.1016/j.molcel.2021.06.026>.
- Bintu, B., Mateo, L.J., Su, J.-H., Sinnott-Armstrong, N.A., Parker, M., Kinrot, S., Yamaya, K., Boettiger, A.N., and Zhuang, X. (2018). Super-resolution chromatin tracing reveals domains and cooperative interactions in single cells. *Science* *362*, eaau1783. <https://doi.org/10.1126/science.aau1783>.
- Bonev, B., Cohen, N.M., Szabo, Q., Fritsch, L., Papadopoulos, G.L., Lubling, Y., Xu, X., Lv, X., Hugnot, J.-P., Tanay, A., et al. (2017). Multiscale 3D Genome Rewiring during Mouse Neural Development. *Cell* *171*, 557-572.e24. <https://doi.org/10.1016/j.cell.2017.09.043>.
- Brandão, H.B., Gabriele, M., and Hansen, A.S. (2021). Tracking and interpreting long-range chromatin interactions with super-resolution live-cell imaging. *Curr Opin Cell Biol* *70*, 18–26. <https://doi.org/10.1016/j.ceb.2020.11.002>.
- Cattoglio, C., Pustova, I., Walther, N., Ho, J.J., Hantsche-Grininger, M., Inouye, C.J., Hossain, M.J., Dailey, G.M., Ellenberg, J., Darzacq, X., et al. (2019). Determining cellular CTCF and cohesin abundances to constrain 3D genome models. *Elife* *8*, e40164. <https://doi.org/10.7554/elife.40164>.
- Cattoni, D.I., Gizzi, A.M.C., Georgieva, M., Stefano, M.D., Valeri, A., Chamousset, D., Houbron, C., Déjardin, S., Fiche, J.-B., González, I., et al. (2017). Single-cell absolute contact probability detection reveals chromosomes are organized by multiple low-frequency yet specific interactions. *Nat Commun* *8*, 1753. <https://doi.org/10.1038/s41467-017-01962-x>.
- Chakraborty, S., Kopitchinski, N., Eraso, A., Awasthi, P., Chari, R., and Rocha, P.P. (2022). High affinity enhancer-promoter interactions can bypass CTCF/cohesin-mediated insulation and contribute to phenotypic robustness. *Biorxiv* 2021.12.30.474562. <https://doi.org/10.1101/2021.12.30.474562>.
- Chereji, R.V., Bryson, T.D., and Henikoff, S. (2019). Quantitative MNase-seq accurately maps nucleosome occupancy levels. *Genome Biol* *20*, 198. <https://doi.org/10.1186/s13059-019-1815-z>.
- Ernst, J., and Kellis, M. (2012). ChromHMM: automating chromatin-state discovery and characterization. *Nat Methods* *9*, 215–216. <https://doi.org/10.1038/nmeth.1906>.
- Filion, G.J., Bemmell, J.G. van, Braunschweig, U., Talhout, W., Kind, J., Ward, L.D., Brugman, W., Castro, I.J. de, Kerkhoven, R.M., Bussemaker, H.J., et al. (2010). Systematic Protein Location Mapping Reveals Five Principal Chromatin Types in Drosophila Cells. *Cell* *143*, 212–224. <https://doi.org/10.1016/j.cell.2010.09.009>.

- Finn, E.H., Pegoraro, G., Brandão, H.B., Valton, A.-L., Oomen, M.E., Dekker, J., Mirny, L., and Misteli, T. (2019). Extensive Heterogeneity and Intrinsic Variation in Spatial Genome Organization. *Cell* 176, 1502-1515.e10. <https://doi.org/10.1016/j.cell.2019.01.020>.
- Gabriele, M., Brandão, H.B., Grosse-Holz, S., Jha, A., Dailey, G.M., Cattoglio, C., Hsieh, T.-H.S., Mirny, L., Zechner, C., and Hansen, A.S. (2021). Dynamics of CTCF and cohesin mediated chromatin looping revealed by live-cell imaging. *Biorxiv* 2021.12.12.472242. <https://doi.org/10.1101/2021.12.12.472242>.
- Gabriele, M., Brandão, H.B., Grosse-Holz, S., Jha, A., Dailey, G.M., Cattoglio, C., Hsieh, T.-H.S., Mirny, L., Zechner, C., and Hansen, A.S. (2022). Dynamics of CTCF- and cohesin-mediated chromatin looping revealed by live-cell imaging. *Science* 376, 496–501. <https://doi.org/10.1126/science.abn6583>.
- Gasperini, M., Hill, A.J., McFaline-Figueroa, J.L., Martin, B., Kim, S., Zhang, M.D., Jackson, D., Leith, A., Schreiber, J., Noble, W.S., et al. (2019). A Genome-wide Framework for Mapping Gene Regulation via Cellular Genetic Screens. *Cell* 176, 377-390.e19. <https://doi.org/10.1016/j.cell.2018.11.029>.
- Giorgetti, L., Galupa, R., Nora, E.P., Piolot, T., Lam, F., Dekker, J., Tiana, G., and Heard, E. (2014). Predictive Polymer Modeling Reveals Coupled Fluctuations in Chromosome Conformation and Transcription. *Cell* 157, 950–963. <https://doi.org/10.1016/j.cell.2014.03.025>.
- Holzmann, J., Politi, A.Z., Nagasaka, K., Hantsche-Grininger, M., Walther, N., Koch, B., Fuchs, J., Dürnberger, G., Tang, W., Ladurner, R., et al. (2019). Absolute quantification of cohesin, CTCF and their regulators in human cells. *Elife* 8, e46269. <https://doi.org/10.7554/elife.46269>.
- Hsieh, T.-H.S., Cattoglio, C., Slobodyanyuk, E., Hansen, A.S., Rando, O.J., Tjian, R., and Darzacq, X. (2020). Resolving the 3D Landscape of Transcription-Linked Mammalian Chromatin Folding. *Mol Cell* 78, 539-553.e8. <https://doi.org/10.1016/j.molcel.2020.03.002>.
- Hua, P., Badat, M., Hanssen, L.L.P., Hentges, L.D., Crump, N., Downes, D.J., Jeziorska, D.M., Oudelaar, A.M., Schwessinger, R., Taylor, S., et al. (2021). Defining genome architecture at base-pair resolution. *Nature* 1–5. <https://doi.org/10.1038/s41586-021-03639-4>.
- Krietenstein, N., Abraham, S., Venev, S.V., Abdennur, N., Gibcus, J., Hsieh, T.-H.S., Parsi, K.M., Yang, L., Maehr, R., Mirny, L.A., et al. (2020). Ultrastructural Details of Mammalian Chromosome Architecture. *Mol Cell* 78, 554-565.e7. <https://doi.org/10.1016/j.molcel.2020.03.003>.
- Kubo, N., Ishii, H., Xiong, X., Bianco, S., Meitinger, F., Hu, R., Hocker, J.D., Conte, M., Gorkin, D., Yu, M., et al. (2021). Promoter-proximal CTCF binding promotes distal enhancer-dependent gene activation. *Nat Struct Mol Biol* 28, 152–161. <https://doi.org/10.1038/s41594-020-00539-5>.
- Li, S., Prasanna, X., Salo, V.T., Vattulainen, I., and Ikonen, E. (2019). An efficient auxin-inducible degron system with low basal degradation in human cells. *Nat Methods* 16, 866–869. <https://doi.org/10.1038/s41592-019-0512-x>.
- Liu, N.Q., Maresca, M., Brand, T. van den, Braccioli, L., Schijns, M.M.G.A., Teunissen, H., Bruneau, B.G., Nora, E.P., and Wit, E. de (2021). WAPL maintains a cohesin loading cycle to preserve cell-type-specific distal gene regulation. *Nat Genet* 53, 100–109. <https://doi.org/10.1038/s41588-020-00744-4>.
- Lu, L., Liu, X., Huang, W.-K., Giusti-Rodríguez, P., Cui, J., Zhang, S., Xu, W., Wen, Z., Ma, S., Rosen, J.D., et al. (2020). Robust Hi-C Maps of Enhancer-Promoter Interactions Reveal the Function of Non-coding Genome in Neural Development and Diseases. *Mol Cell* <https://doi.org/10.1016/j.molcel.2020.06.007>.

- Luan, J., Xiang, G., Gómez-García, P.A., Tome, J.M., Zhang, Z., Vermunt, M.W., Zhang, H., Huang, A., Keller, C.A., Giardine, B.M., et al. (2021). Distinct properties and functions of CTCF revealed by a rapidly inducible degron system. *Cell Reports* 34, 108783. <https://doi.org/10.1016/j.celrep.2021.108783>.
- Luppino, J.M., Park, D.S., Nguyen, S.C., Lan, Y., Xu, Z., Yunker, R., and Joyce, E.F. (2020). Cohesin promotes stochastic domain intermingling to ensure proper regulation of boundary-proximal genes. *Nat Genet* 52, 840–848. <https://doi.org/10.1038/s41588-020-0647-9>.
- Mach, P., Kos, P.I., Zhan, Y., Cramard, J., Gaudin, S., Tünnermann, J., Marchi, E., Eglinger, J., Zuin, J., Kryzhanovska, M., et al. (2022). Live-cell imaging and physical modeling reveal control of chromosome folding dynamics by cohesin and CTCF. *Biorxiv* 2022.03.03.482826. <https://doi.org/10.1101/2022.03.03.482826>.
- Mateo, L.J., Murphy, S.E., Hafner, A., Cinquini, I.S., Walker, C.A., and Boettiger, A.N. (2019). Visualizing DNA folding and RNA in embryos at single-cell resolution. *Nature* 568, 49–54. <https://doi.org/10.1038/s41586-019-1035-4>.
- Matthey-Doret, C., Baudry, L., Breuer, A., Montagne, R., Guiguelmoni, N., Scolari, V., Jean, E., Campeas, A., Chanut, P.-H., Oriol, E., et al. (2020a). Chromosight: A computer vision program for pattern detection in chromosome contact maps. *Biorxiv* 2020.03.08.981910. <https://doi.org/10.1101/2020.03.08.981910>.
- Matthey-Doret, C., Baudry, L., Breuer, A., Montagne, R., Guiguelmoni, N., Scolari, V., Jean, E., Campeas, A., Chanut, P.H., Oriol, E., et al. (2020b). Computer vision for pattern detection in chromosome contact maps. *Nat Commun* 11, 5795. <https://doi.org/10.1038/s41467-020-19562-7>.
- Moorthy, S.D., Davidson, S., Shchuka, V.M., Singh, G., Malek-Gilani, N., Langroudi, L., Martchenko, A., So, V., Macpherson, N.N., and Mitchell, J.A. (2017). Enhancers and super-enhancers have an equivalent regulatory role in embryonic stem cells through regulation of single or multiple genes. *Genome Res* 27, 246–258. <https://doi.org/10.1101/gr.210930.116>.
- Nagano, T., Lubling, Y., Stevens, T.J., Schoenfelder, S., Yaffe, E., Dean, W., Laue, E.D., Tanay, A., and Fraser, P. (2013). Single-cell Hi-C reveals cell-to-cell variability in chromosome structure. *Nature* 502, 59–64. <https://doi.org/10.1038/nature12593>.
- Oksuz, B.A., Yang, L., Abraham, S., Venev, S.V., Krietenstein, N., Parsi, K.M., Ozadam, H., Oomen, M.E., Nand, A., Mao, H., et al. (2021). Systematic evaluation of chromosome conformation capture assays. *Nat Methods* 18, 1046–1055. <https://doi.org/10.1038/s41592-021-01248-7>.
- Owens, N., Papadopoulou, T., Festuccia, N., Tachtsidi, A., Gonzalez, I., Dubois, A., Vandormael-Pournin, S., Nora, E.P., Bruneau, B.G., Cohen-Tannoudji, M., et al. (2019). CTCF confers local nucleosome resiliency after DNA replication and during mitosis. *Elife* 8, e47898. <https://doi.org/10.7554/elife.47898>.
- Rao, S.S.P., Huang, S.-C., Hilaire, B.G.S., Engreitz, J.M., Perez, E.M., Kieffer-Kwon, K.-R., Sanborn, A.L., Johnstone, S.E., Bascom, G.D., Bochkov, I.D., et al. (2017). Cohesin Loss Eliminates All Loop Domains. *Cell* 171, 305–320.e24. <https://doi.org/10.1016/j.cell.2017.09.026>.
- Rinaldi, L., Fettweis, G., Kim, S., Garcia, D.A., Fujiwara, S., Johnson, T.A., Tettey, T.T., Ozbun, L., Pegoraro, G., Puglia, M., et al. (2022). The glucocorticoid receptor associates with the cohesin loader NIPBL to promote long-range gene regulation. *Sci Adv* 8, eabj8360. <https://doi.org/10.1126/sciadv.abj8360>.
- Rodríguez-Carballo, E., Lopez-Delisle, L., Willemin, A., Beccari, L., Gitto, S., Mascrez, B., and Duboule, D. (2020). Chromatin topology and the timing of enhancer function at the HoxD locus. *Proc National Acad Sci* 117, 31231–31241. <https://doi.org/10.1073/pnas.2015083117>.

Schwartz, U., Németh, A., Diermeier, S., Exler, J.H., Hansch, S., Maldonado, R., Heizinger, L., Merkl, R., and Längst, G. (2018). Characterizing the nuclease accessibility of DNA in human cells to map higher order structures of chromatin. *Nucleic Acids Res* 47, 1239–1254. <https://doi.org/10.1093/nar/gky1203>.

Stevens, T.J., Lando, D., Basu, S., Atkinson, L.P., Cao, Y., Lee, S.F., Leeb, M., Wohlfahrt, K.J., Boucher, W., O’Shaughnessy-Kirwan, A., et al. (2017). 3D structures of individual mammalian genomes studied by single-cell Hi-C. *Nature* 544, 59–64. <https://doi.org/10.1038/nature21429>.

Szabo, Q., Donjon, A., Jerković, I., Papadopoulos, G.L., Cheutin, T., Bonev, B., Nora, E.P., Bruneau, B.G., Bantignies, F., and Cavalli, G. (2020). Regulation of single-cell genome organization into TADs and chromatin nanodomains. *Nat Genet* 52, 1151–1157. <https://doi.org/10.1038/s41588-020-00716-8>.

Thiecke, M.J., Wutz, G., Muhar, M., Tang, W., Bevan, S., Malysheva, V., Stocsits, R., Neumann, T., Zuber, J., Fraser, P., et al. (2020). Cohesin-Dependent and -Independent Mechanisms Mediate Chromosomal Contacts between Promoters and Enhancers. *Cell Reports* 32, 107929. <https://doi.org/10.1016/j.celrep.2020.107929>.

Xu, B., Wang, H., Wright, S., Hyle, J., Zhang, Y., Shao, Y., Niu, M., Fan, Y., Rosikiewicz, W., Djekidel, M.N., et al. (2021). Acute depletion of CTCF rewires genome-wide chromatin accessibility. *Genome Biol* 22, 244. <https://doi.org/10.1186/s13059-021-02466-0>.

Decision Letter, first revision:

Dear Dr. Tjian,

Your Article, entitled "Enhancer-promoter interactions and transcription are largely maintained upon acute loss of CTCF, cohesin, WAPL, or YY1", has now been seen by 3 of the original referees. Unfortunately, reviewer #4 was unable to submit a timely report. We have now decided to proceed with a decision.

You will see from the reviewers' comments below that while they find your work improved overall, some important points are raised. We are interested in the possibility of publishing your study in Nature Genetics, but would like to consider your response to these serious concerns in the form of a revised manuscript before we make a final decision on publication.

Reviewer #1 acknowledges that you have done an impressive effort during the revision but they raise a potentially important technical issue with matrix balancing, which may affect all of the Micro-C analysis. We encourage you to address this point carefully since it may impact many of the biological conclusions.

Reviewer #2 doesn't have any major technical concerns but would like you to substantially tone down the language used throughout, taking into account important methodological caveats.

Reviewer #3 is mostly satisfied and recommends eventual publication. They have two remaining suggestions, which seem doable.

We therefore invite you to revise your manuscript taking into account all reviewer and editor comments. Please highlight all changes in the manuscript text file. At this stage we will need you to upload a copy of the manuscript in MS Word .docx or similar editable format.

We are committed to providing a fair and constructive peer-review process. Do not hesitate to contact me if there are specific requests from the reviewers that you believe are technically impossible or unlikely to yield a meaningful outcome. I would be happy to discuss the reviewers' comments in detail.

*2) If you have not done so already please begin to revise your manuscript so that it conforms to our Article format instructions, available

http://www.nature.com/ng/authors/article_types/index.html here

*3) Include a revised version of any required Reporting Summary:

[redacted]

We hope to receive your revised manuscript within ~3 months. If you cannot send it within this time, please let us know.

Sincerely,

Tiago

Tiago Faial, PhD
Senior Editor
Nature Genetics
<https://orcid.org/0000-0003-0864-1200>

Reviewers' Comments:

Reviewer #1:

Remarks to the Author:

The revised manuscript displays an impressive amount of effort and attention to detail on the part of the authors.

However, new information in the reply points to a potentially important technical issue with matrix balancing. The impact on downstream analysis and interpretation are unclear, but may affect the validity of the conclusions regarding the persistence of E-P loops upon CTCF/RAD21 degradation. Resolving this issue in the computational analysis is necessary before any of the claims of this manuscript that involve micro-C data can be evaluated further. At this stage it is impossible to evaluate the parts of the manuscript that rely on micro-C data - conclusions might very well be correct, or they may be misled by improper data correction. Before moving further it is imperative that authors address these technical issues.

As explained in detail below, it is unclear if the loops that remain upon degradation of key architectural factors can arise due to incomplete coverage correction. We understand the authors use the same procedure as in their previous mouse micro-C paper. However, given that loop calling is so central to the argumentation of the manuscript, it is absolutely critical that authors revisit the data processing, and ensure that the results are a solid foundation for future work in the field.

1) As authors know, matrix balancing aims to remove multiplicative biases, resulting in corrected maps where the marginal sum over rows or columns is a constant value (e.g. Imakev et al., 2012, Fig 1b,c). For the matrix in EDa(i), this does not appear to be the case. If uniform coverage across the genome had been achieved, dots should fall in a vertical line in the panel for balanced coverage. Instead the reported coverage varies between 100 and 500 for the balanced. This means that biases in micro-C matrices have not been properly corrected, which may affect a number of downstream analyses, and dot/loop-calling in particular.

Without more details about the computational analysis, it is unclear why the authors' approach did not correctly correct the micro-C matrices. It is imperative that authors resolve this issue and provide evidence that micro-C matrices were properly corrected before any downstream analysis.

A potential avenue to address this could be a more stringent filter on bin coverage before balancing. Filtering at the bin level is known to be required for convergence in lower-resolution Hi-C data. The high resolutions considered for Micro-C data here might benefit from a more stringent filtering strategy. It is unclear which samples or resolutions this issue might impact, but obtaining matrices with coverage that is uniform (e.g. to within $1e-03$) is crucial for downstream analysis.

2) ED1a (ii) is puzzling in light of the strong positive correlation between coverage and ATAC-seq signal in the raw data, where the opposite relationship is shown. This could potentially relate to the non-uniformity in coverage in currently-analyzed maps.

3) The reanalysis in ED1b shows a strong bias for loci with multiple loops to have 1.5x higher coverage, and this issue appears at all resolutions. This could arise from incomplete coverage correction, and has implications for the analysis and interpretation of the number of loops that remain or change after perturbing architectural factors.

4) the authors do not provide the map of pairwise correlations between treated replicates for RNA-seq

and mNET-seq as requested, only for the untreated samples. This is crucial for showing that the depletions cluster with those of the same type, and away from all the untreated samples. The authors could also show this via PCA to illustrate the absence of batch effects confounding the analysis.

The incorporation of new mNET-seq and RNA-seq data with dBET6 BRD inhibition greatly supports the initial argument that acute depletion of CTCF/RAD21/WAPL has less effect on transcription than direct transcriptional co-activators. The added time-course data demonstrate that defects eventually become detectable with time (at least for CTCF and RAD21), and is a very strong addition to the transcription part of the manuscript.

Reviewer #2:

Remarks to the Author:

The authors have adequately addressed most of my concerns – many of which were shared between my reviewing colleagues. Overall, this paper adds important findings to the field, albeit many are negative results, combining acute inhibition of architectural proteins with advanced Micro-C technology to measure effects of smaller E-P interactions typically missed with conventional approaches.

My remaining issue is the use of a single time-point (3 hours) in a single cell line to define **maintenance** of both chromatin interactions and gene expression effects. However, this reviewer does note that in many cases the authors do a nice job comparing their work on acute inhibition to others when the data don't match. For example, the addition of the qualifying statement at the end of Line 369-370: "This result suggests that the maintenance of most E-P loops and their regulatory functions in general do not require the presence of YY1, at least within a 3-hour depletion window."

However, this point seems lost and inconsistent with the conclusion at the end of the same paragraph in which the authors state "that these results not consistent with the previously proposed model of YY1 acting as a general master structural regulator of E-P interactions in mESCs [70]."

Is it not possible to be essential for establishment (which is not tested here) and still be considered a general master of structural regulation? In short, comparative statements like these are difficult to reconcile since identical conditions were not tested and several timepoints were not compared in this study. General conclusions should be made with caution given these caveats and the already complicated relationship between chromatin folding and gene expression in the field.

I think this issue can easily be addressed with a softening of conclusions throughout the results section of the manuscript, especially when disputing other data that are not directly compared in this study. I would also appreciate an additional qualifying statement in the discussion pertaining to the incomplete nature of degron-induced depletion. It remains possible the difference in levels (even if small by current measurements) between 3 hours and 6-24 hours is alone sufficient to induce structural and expression changes. Perhaps this could be tested in arrested cells.

Separately, I had a difficult time following the logic from the second part of the paper pertaining to YY1. On the one hand, the authors state that acute loss of either cohesin or YY1 had little effect, if any, on E-P interactions and gene expression. Then, in the second half of the manuscript, the authors argue that cohesin has a significant role in TF binding to chromatin that may be relevant to gene

regulation in general. Is the thought here that extended depletions would lead to a bigger effect on TF binding or that extended time with altered TF binding would lead to changes in gene expression? Obviously, neither model is tested here but perhaps this could be made clearer in the discussion.

Reviewer #3:

Remarks to the Author:

I thank the authors for their extensive response to my previous critiques and appreciate the response from the authors to many of my concerns. The observations presented in the current work, arguing minimal effects on chromatin interactions and transcriptional response upon removing most of the chromatin structural proteins, are important to be reported, since they will help to clarify the role (or lack of) of CTCF, cohesin, and YY1 in the E-P and P-P chromatin interactions, and transcriptional regulation in mammalian cells. I would recommend acceptance of the manuscript provided that the authors can address several remaining issues as detailed below:

1. In their model, they indicate loss of chromatin memory (shown by loss of E-P interactions) at later time points (>1 cell division) without showing any interaction data for the later time points. I would recommend showing it via performing it by a Micro-C at later time point of degradation.
2. Do the regions where Rad21 depletion reduced YY1 binding show any changes in looping / interactions? It would be of interest to see whether this suggested role of Rad21 in TF binding influence chromatin interactions at the regions where Rad21 depletion reduces binding of YY1 or other TFs.

Author Rebuttal, second revision:

Reviewers' Comments:

Reviewer #1:

Remarks to the Author:

The revised manuscript displays an impressive amount of effort and attention to detail on the part of the authors.

However, new information in the reply points to a potentially important technical issue with matrix balancing. The impact on downstream analysis and interpretation are unclear, but may affect the validity of the conclusions regarding the persistence of E-P loops upon CTCF/RAD21 degradation. Resolving this issue in the computational analysis is necessary before any of the claims of this manuscript that involve micro-C data can be evaluated further. At this stage it is impossible to evaluate the parts of the manuscript that rely on micro-C data - conclusions might very well be correct, or they may be misled by improper data correction. Before moving further it is imperative that authors address these technical issues.

As explained in detail below, it is unclear if the loops that remain upon degradation of key architectural factors can arise due to incomplete coverage correction. We understand the authors use the same procedure as in their previous mouse micro-C paper. However, given that loop calling is so central to the argumentation of the manuscript, it is absolutely critical that authors revisit the data processing, and ensure that the results are a solid foundation for future work in the field.

1) As authors know, matrix balancing aims to remove multiplicative biases, resulting in corrected maps where the marginal sum over rows or columns is a constant value (e.g. Imakev et al., 2012, Fig 1b,c). For the matrix in EDa(i), this does not appear to be the case. If uniform coverage across the genome had been achieved, dots should fall in a vertical line in the panel for balanced coverage. Instead the reported coverage varies between 100 and 500 for the balanced. This means that biases in micro-C matrices have not been properly corrected, which may affect a number of downstream analyses, and dot/loop-calling in particular.

Without more details about the computational analysis, it is unclear why the authors' approach did not correctly correct the micro-C matrices. It is imperative that authors resolve this issue and provide evidence that micro-C matrices were properly corrected before any downstream analysis.

A potential avenue to address this could be a more stringent filter on bin coverage before balancing. Filtering at the bin level is known to be required for convergence in lower-resolution Hi-C data. The high resolutions considered for Micro-C data here might benefit from a more stringent filtering strategy. It is unclear which samples or resolutions this issue might impact, but obtaining matrices with coverage that is uniform (e.g. to within $1e-03$) is crucial for downstream analysis.

2) ED1a (ii) is puzzling in light of the strong positive correlation between coverage and ATAC-seq signal in the raw data, where the opposite relationship is shown. This could potentially relate to the non-uniformity in coverage in currently-analyzed maps.

3) The reanalysis in ED1b shows a strong bias for loci with multiple loops to have 1.5x higher coverage, and this issue appears at all resolutions. This could arise from incomplete coverage correction, and has implications for the analysis and interpretation of the number of loops that remain or change after perturbing architectural factors.

Response: We thank the reviewer for pointing out the issue in this analysis. We have fixed a bug in the code we used to generate the **Extended Data Figure 1a**. We note that this error affected only the supplementary figure, and the rest of the analysis in the manuscript was unaffected. The balanced Micro-C data are now correctly plotted, showing uniform coverage across the genome regardless of ATAC-seq signals (**Extended Data Figure 1a(i)**). While the raw Micro-C data shows an anti-correlation to ATAC-seq around ATAC-seq peaks, this trend disappears after matrix balance (**Extended Data Figure 1a(ii)**). Also, when plotted using raw Micro-C data in **Extended Data Figure 1b**, we previously demonstrated that the loop calling in the high-resolution data is not explicitly affected by the 1D coverage. We have now included the balanced data that demonstrates the number of loops is not dependent on the coverage. Together, we confirmed that Micro-C data were normalized/balanced correctly throughout the manuscript, and that loop calling was not biased in favor of high chromatin accessibility regions. A snapshot of the revised figures is shown below.

4) the authors do not provide the map of pairwise correlations between treated replicates for RNA-seq and mNET-seq as requested, only for the untreated samples. This is crucial for showing that the depletions cluster with those of the same type, and away from all the untreated samples. The authors could also show this via PCA to illustrate the absence of batch effects confounding the analysis.

The incorporation of new mNET-seq and RNA-seq data with dBET6 BRD inhibition greatly supports the initial argument that acute depletion of CTCF/RAD21/WAPL has less effect on transcription than direct transcriptional co-activators. The added time-course data demonstrate that defects eventually become detectable with time (at least for CTCF and RAD21), and is a very strong addition to the transcription part of the manuscript.

Response: We thank the reviewer for pointing out the missing comparison in the transcription analysis. Consistent with differential expression analysis identifying only a few significantly altered genes after 3-hour depletion, Principal component analysis (PCA) of RNA-seq and mNET-seq revealed that all the clonal cell lines derived from the C59 parental clone (see Methods for details; control (C59), Δ CTCF (C58), Δ RAD21 (F1), Δ WAPL (C40) clones) with or

without IAA treatment are clustered together, except for cells depleted of RAD21 after 12 and 24 hours (DEGs > 1000). Additionally, we found that YY1-depleted cells are clustered separately from C59-derived cells and are more similar to wild-type JM8.N4 cells. The new results are in **Extended Data Figure 5f** and a snapshot is shown below.

Reviewer #2:

Remarks to the Author:

The authors have adequately addressed most of my concerns – many of which were shared between my reviewing colleagues. Overall, this paper adds important findings to the field, albeit many are negative results, combining acute inhibition of architectural proteins with advanced Micro-C technology to measure effects of smaller E-P interactions typically missed with conventional approaches.

My remaining issue is the use of a single time-point (3 hours) in a single cell line to define maintenance of both chromatin interactions and gene expression effects. However, this reviewer does note that in many cases the authors do a nice job comparing their work on acute inhibition to others when the data don't match. For example, the addition of the qualifying statement at the end of Line 369-370: "This result suggests that the maintenance of most E-P loops and their regulatory functions in general do not require the presence of YY1, at least within a 3-hour depletion window."

However, this point seems lost and inconsistent with the conclusion at the end of the same paragraph in which the authors state "that these results not consistent with the previously proposed model of YY1

acting as a general master structural regulator of E-P interactions in mESCs [70].”

Is it not possible to be essential for establishment (which is not tested here) and still be considered a general master of structural regulation? In short, comparative statements like these are difficult to reconcile since identical conditions were not tested and several timepoints were not compared in this study. General conclusions should be made with caution given these caveats and the already complicated relationship between chromatin folding and gene expression in the field.

I think this issue can easily be addressed with a softening of conclusions throughout the results section of the manuscript, especially when disputing other data that are not directly compared in this study. I would also appreciate an additional qualifying statement in the discussion pertaining to the incomplete nature of degron-induced depletion. It remains possible the difference in levels (even if small by current measurements) between 3 hours and 6-24 hours is alone sufficient to induce structural and expression changes. Perhaps this could be tested in arrested cells.

Response: We rephrased our statements when comparing our results to other studies and reiterated throughout the text that these results came from a specific cell type (mESCs) with a short degradation time.

Concerning the incomplete nature of the degron-induced depletion that was raised by another reviewer in the first round of revision, we provide below our previous response to argue against those residual proteins are sufficient to maintain E-P/P-P interactions.

*“We performed biochemical fractionation and flow cytometry analysis to determine the fraction of proteins remaining on chromatin after 3 hours of depletion (**Fig. 18**; please see (Gabriele et al., 2021), Fig. S4 and **Extended Data Fig. 2** and 7 for full details). In brief, our results showed that CTCF and RAD21 are nearly completely degraded 3 hours after depletion and that only ~8% and 13% of residual WAPL and YY1 remain associated with chromatin, respectively. Furthermore, we have previously measured the average number of CTCF, RAD21, and YY1 molecules in a single cell (Cattoglio et al., 2019; Holzmann et al., 2019). Our analysis suggests that if our degradation reaches ~95% depletion for CTCF and RAD21 (due to the detection limit of western blot) and ~87% depletion for YY1, at most only ~787 CTCF, ~722 RAD21, and ~700 YY1 proteins would remain bound to chromatin (per chromosome). A few hundreds of residual architectural proteins are very unlikely sufficient to maintain >50,000 E-P loops and transcription. Also, since only hundreds of proteins remain after depletion and all CTCF and cohesin loops are nearly completely abolished, it is hard to conceive that the residual proteins will specifically maintain E-P interactions but not CTCF/cohesin loops. If any, the effect of residual proteins on our results is minimal.”*

(Fig. 18)

Separately, I had a difficult time following the logic from the second part of the paper pertaining to YY1. On the one hand, the authors state that acute loss of either cohesin or YY1 had little effect, if any, on E-P interactions and gene expression. Then, in the second half of the manuscript, the authors argue that cohesin has a significant role in TF binding to chromatin that may be relevant to gene regulation in general. Is the thought here that extended depletions would lead to a bigger effect on TF binding or that extended time with altered TF binding would lead to changes in gene expression? Obviously, neither model is tested here but perhaps this could be made clearer in the discussion.

Response: Recruitment of transcription factors (TFs) and co-factors to cis-regulatory elements is a critical step of transcription initiation/establishment. We found that cohesin (or cohesin-mediated chromatin structure) is important to facilitate the TF target search (at least for 3 TFs tested here). Rinaldi et al. (2022) independently confirmed this mechanism for the glucocorticoid receptor. Thus, although cohesin is not required for the short-term maintenance of 3D genome and transcription, it is possible that cohesin or cohesin-mediated chromatin structure could regulate transcription in other ways. In our proposed model, the persistent chromatin marks and other transcription proteins may keep sustaining transcription for a while. During this period, there is no apparent change in gene expression (e.g., 3 hours in this study). Nevertheless, after cohesin loss, TFs take a longer time to find their targets. A longer TF searching process may result in less efficient transcription activation and eventually lead to changes in gene expression. We have now added a sentence to the discussion to articulate such hypothesis.

Reviewer #3:

Remarks to the Author:

I thank the authors for their extensive response to my previous critiques and appreciate the response from the authors to many of my concerns. The observations presented in the current work, arguing minimal effects on chromatin interactions and transcriptional response upon removing most of the chromatin structural proteins, are important to be reported, since they will help to clarify the role (or lack of) of CTCF, cohesin, and YY1 in the E-P and P-P chromatin interactions, and transcriptional regulation in mammalian cells. I would recommend acceptance of the manuscript provided that the authors can address several remaining issues as detailed below:

1. In their model, they indicate loss of chromatin memory (shown by loss of E-P interactions) at later time points (>1 cell division) without showing any interaction data for the later time points. I would recommend showing it via performing it by a Micro-C at later time point of degradation.

Response: The major goal of this study is to understand the immediate effects of CTCF, cohesin, WAPL, and YY1 on genome organization. In our analysis, we showed that >90% of the target proteins are depleted from cell ~3 hours – which is possibly the shortest time allowing us to test it and eliminate unidentified secondary effects that may arise after long-term loss of loop extrusion factors (i.e., cohesin depletion leads to widespread cell death and cell cycle arrest after 3 hours). Furthermore, evaluation of enhancer-promoter interactions after a longer degradation time has been reported in other studies (e.g., Kubo et al., 48 & 96 hours by HiChIP/PLAC-seq; Aljahani et al., 48 hours for CTCF and 6 hours for RAD21 by Tiled Micro-Capture-C (TMCC)). While they observed slightly stronger effects on E-P interactions with a longer degradation time, the signals are still largely maintained. Therefore, we argue that performing an additional Micro-C experiment with a longer degradation time course is outside the scope of the current manuscript and would not add novelty or clarity to our conclusions.

2. Do the regions where Rad21 depletion reduced YY1 binding show any changes in looping / interactions? It would be of interest to see whether this suggested role of Rad21 in TF binding influence chromatin interactions at the regions where Rad21 depletion reduces binding of YY1 or other TFs.

Response: We thank the reviewer for suggesting this additional analysis. In the figure below we highlighted the loops in which anchors are associated with the downregulated YY1 peaks after RAD21 depletion (replot of **Figure 3b**. YY1-down loops are shown in pink and the density distribution is shown in blue). We found that these loops remain largely unchanged after RAD21 depletion, similar to loops anchored by unaffected YY1 binding sites. Thus, cohesin-sensitive YY1 peaks are not specifically required for the maintenance of focal chromatin interactions. This is in line with our observation that YY1 is not generally required for the maintenance of E-P and P-P interactions.

Reviewer #4:

None

Decision Letter: Second Revision

Dear Dr. Tjian,

Thank you for submitting your revised manuscript "Enhancer-promoter interactions and transcription are largely maintained upon acute loss of CTCF, cohesin, WAPL, or YY1" (NG-A58009R2). It has now been seen by reviewer #1 and their comments are below. The reviewer finds that the paper has improved in revision, and therefore we'll be happy in principle to publish it in Nature Genetics, pending minor revisions to satisfy some of the referee's final requests and to comply with our editorial and formatting guidelines.

Thank you again for your interest in Nature Genetics. Please do not hesitate to contact me if you have any questions.

Congratulations!

Sincerely,

Tiago

Tiago Faial, PhD
Senior Editor
Nature Genetics
<https://orcid.org/0000-0003-0864-1200>

Reviewer #1 (Remarks to the Author):

The new analyses provided by Hsieh et al. address my previous reservations. Together with the new transcriptomic data the manuscript is much stronger and supports the statement that tampering with loop extrusion is initially less impactful on transcription than tampering with transcription co-factors. I encourage timely publication at this stage.

It would be very helpful if authors could include the figure about micro-C coverage at anchors and add a short sentence explaining the caveat about coverage at anchors. This is because authors' statement that "loop calling was not biased in favor of high chromatin accessibility region" appears incorrect – the number of loops called has a strong correlation with the initial coverage.

Authors now report that nearly 1000 genes are dysregulated after 12h of RAD21 depletion. While I understand this may very well be contributed by indirect effects, authors do not directly address

whether these changes simply reflect altered cell cycle / viability or if genes become dysregulated because of loss loop extrusion. I suggest authors simply update the title of the manuscript to reflect this – for example “Enhancer-promoter interactions and transcription are *INITIALLY/FIRST largely maintained upon acute loss of CTCF, cohesin, WAPL, or YY1”. This would also better reflect the “Time-buffering” model depicted in fig. 7.

Author Rebuttal, Second Revision:

Reviewer #1 (Remarks to the Author):

The new analyses provided by Hsieh et al. address my previous reservations. Together with the new transcriptomic data the manuscript is much stronger and supports the statement that tampering with loop extrusion is initially less impactful on transcription than tampering with transcription co-factors. I encourage timely publication at this stage.

It would be very helpful if authors could include the figure about micro-C coverage at anchors and add a short sentence explaining the caveat about coverage at anchors. This is because authors’ statement that “loop calling was not biased in favor of high chromatin accessibility region” appears incorrect – the number of loops called has a strong correlation with the initial coverage.

Response: We thank the reviewer’s suggestion. However, despite a correlation between “Micro-C raw coverage” and the number of loops per locus, we note that Micro-C loops were identified using “ICE normalized data”, so any bias in coverage should have been eliminated before loop calling (i.e., each region has equal coverage). To further confirm that Micro-C loops are not biased toward regions with high chromatin accessibility and high Micro-C coverage, we conducted an additional analysis focusing on ATAC-seq peaks within vs. outside Micro-C loop anchors. We found that:

- Only ~27% of all Micro-C loop anchors overlap with ATAC-seq peaks (**Fig 1A**).
- Vice versa, ~28% of all ATAC-seq peaks correspond to Micro-C loops anchors (**Fig 1A**).
- Micro-C loop anchors have a high ATAC-seq signal, but high ATAC-seq peaks do not necessarily form loops (**Fig 1B**).
- Loop anchors have a high Micro-C raw coverage, but regions with high Micro-C raw coverage do not necessarily form loops (**Fig 1C**).
- More importantly, we used ICE balanced data for loop calling in this study. The ICE matrix balancing normalized Micro-C coverage and showed no difference between ATAC-seq peaks within vs. outside Micro-C loop anchors (**Fig 1D**). Thus, the statement “*loop calling was not biased in favor of high chromatin accessibility region*” is accurate. We argue that the correlation of raw Micro-C coverage (MNase accessibility) and loop number is also likely to reflect the actual TFs and loop extrusion factors involved in accessing the genome and mediating chromatin interactions, which are not trivial to disentangle molecularly and biochemically from MNase accessibility.

Fig 1.

A.

B.

C.

D.

Authors now report that nearly 1000 genes are dysregulated after 12h of RAD21 depletion. While I understand this may very well be contributed by indirect effects, authors do not directly address whether these changes simply reflect altered cell cycle / viability or if genes become dysregulated because of loss loop extrusion. I suggest authors simply update the title of the manuscript to reflect this – for example “Enhancer-promoter interactions and transcription are *INITIALLY/FIRST largely maintained upon acute loss of CTCF, cohesin, WAPL, or YY1”. This would also better reflect the “Time-buffering” model depicted in fig. 7.

Response: As the editor suggested, we kept the manuscript title unchanged.

Final Decision Letter:

Dear Dr. Tjian,

I am delighted to say that your manuscript "Enhancer-promoter interactions and transcription are largely maintained upon acute loss of CTCF, cohesin, WAPL, or YY1" has been accepted for publication in an upcoming issue of Nature Genetics.

Your paper will be published online after we receive your corrections and will appear in print in the next available issue. You can find out your date of online publication by contacting the Nature Press Office (press@nature.com) after sending your e-proof corrections. Now is the time to inform your Public Relations or Press Office about your paper, as they might be interested in promoting its publication. This will allow them time to prepare an accurate and satisfactory press release. Include your manuscript tracking number (NG-A58009R3) and the name of the journal, which they will need when they contact our Press Office.

Please note that *Nature Genetics* is a Transformative Journal (TJ). Authors may publish their research with us through the traditional subscription access route or make their paper immediately open access through payment of an article-processing charge (APC). Authors will not be required to make a final decision about access to their article until it has been accepted. [Find out more about Transformative Journals](https://www.springernature.com/gp/open-research/transformative-journals)

Authors may need to take specific actions to achieve [compliance](https://www.springernature.com/gp/open-research/funding/policy-compliance-faq) with funder and institutional open access mandates. If your research is supported by a funder that requires immediate open access (e.g. according to [a](https://www.springernature.com/gp/open-research/funding/policy-compliance-faq)

[Plan S principles](https://www.springernature.com/gp/open-research/plan-s-compliance)) then you should select the gold OA route, and we will direct you to the compliant route where possible. For authors selecting the subscription publication route, the journal's standard licensing terms will need to be accepted, including <https://www.nature.com/nature-portfolio/editorial-policies/self-archiving-and-license-to-publish>. Those licensing terms will supersede any other terms that the author or any third party may assert apply to any version of the manuscript.

Please note that Nature Portfolio offers an immediate open access option only for papers that were first submitted after 1 January, 2021.

Sincerely,

Tiago

Tiago Faial, PhD
Chief Editor
Nature Genetics
<https://orcid.org/0000-0003-0864-1200>

Click here if you would like to recommend Nature Genetics to your librarian
<http://www.nature.com/subscriptions/recommend.html#forms>

** Visit the Springer Nature Editorial and Publishing website at http://editorial-jobs.springernature.com?utm_source=ejp_NGen_email&utm_medium=ejp_NGen_email&utm_campaign=ejp_NGen for more information about our career opportunities. If you have any questions please click [here](mailto:editorial.publishing.jobs@springernature.com).**